# ATP6V0A1-dependent cholesterol absorption in colorectal cancer cells triggers immunosuppressive signaling to inactivate memory CD8+ T cells

Tu-Xiong Huang [1,8], Hui-Si Huang[1,8], Shao-Wei Dong[2,3], Jia-Yan Chen[1], Bin Zhang [2], Hua-Hui Li[4], Tian-Tian Zhang[1], Qiang Xie[1], Qiao-Yun Long[2], Yang Yang[1], Lin-Yuan Huang[1], Pan Zhao[2], Jiong Bi[5], Xi-Feng Lu[6], Fan Pan [4], Chang Zou [2,7] ✉ & Li Fu [1] ✉

Obesity shapes anti-tumor immunity through lipid metabolism; however, the mechanisms underlying how colorectal cancer (CRC) cells utilize lipids to suppress anti-tumor immunity remain unclear. Here, we show that tumor cell-intrinsic ATP6V0A1 drives exogenous cholesterol-induced immunosuppression in CRC. ATP6V0A1 facilitates cholesterol absorption in CRC cells through RAB guanine nucleotide exchange factor 1 (RABGEF1)-dependent endosome maturation, leading to cholesterol accumulation within the endoplasmic reticulum and elevated production of 24-hydroxycholesterol (24-OHC). ATP6V0A1-induced 24-OHC upregulates TGF-β1 by activating the liver X receptor (LXR) signaling. Subsequently, the release of TGF-β1 into the tumor microenvironment by CRC cells activates the SMAD3 pathway in memory CD8+ T cells, ultimately suppressing their anti-tumor activities. Moreover, we identify daclatasvir, a clinically used anti-hepatitis C virus (HCV) drug, as an ATP6V0A1 inhibitor that can effectively enhance the memory CD8+ T cell activity and suppress tumor growth in CRC. These findings shed light on the potential for ATP6V0A1-targeted immunotherapy in CRC.

Obesity has become a global health concern. One of the major risks associated with obesity is the increased likelihood of developing various types of cancer, including colorectal cancer (CRC). There is growing evidence that obesity can impact the immune system's response to tumors via the lipid metabolic reprogramming[1,2]. Obesity suppresses anti-tumor immunity by promoting fatty acid (FA) metabolism in CRC tumor cells. High-fat diet (HFD)-induced obesity downregulates prolyl-4-hydroxylase 3 (PHD3) expression and thus promotes FA uptake and oxidation in CRC cells, depriving the tumor microenvironment (TME) of FAs and, thus, attenuating the

[1]Guangdong Provincial Key Laboratory of Regional Immunity and Diseases, Department of Pharmacology and International Cancer Center, Shenzhen University Medical School, Shenzhen 518060 Guangdong, China. [2]Department of Clinical Medical Research Center, The First Affiliated Hospital of Southern University of Science and Technology (Shenzhen People's Hospital), Shenzhen 518000 Guangdong, China. [3]Department of Hematology and Oncology, Shenzhen Children's Hospital, Shenzhen 518038 Guangdong, China. [4]Shenzhen Institute of Advanced Technology (SIAT), Chinese Academy of Sciences (CAS), Shenzhen 518055 Guangdong, China. [5]The First Affiliated Hospital, Sun Yat-sen University, Guangzhou 510080 Guangdong, China. [6]Clinical Research Center, The First Affiliated Hospital of Shantou University Medical College, Shantou 515041 Guangdong, China. [7]School of Life and Health Sciences, The Chinese University of Hong Kong, Shenzhen 518000 Guangdong, China. [8]These authors contributed equally: Tu-Xiong Huang, Hui-Si Huang. ✉e-mail: zouchang@cuhk.edu.cn; gracelfu@szu.edu.cn

FA-supported anti-tumor activities of CD8+ T cells[1]. Apart from FA, cholesterol in the TME also plays an essential role in regulating anti-tumor immunity. Appropriate levels of cholesterol in the TME are known to be essential to maintain the anti-tumor function of CD8+ T cells[3] while conversely, excess cholesterol may lead to T cell exhaustion[2]. On the other hand, cholesterol produced by tumor cells drives their immunosuppressive reprogramming. The enhanced synthesis of endogenous cholesterol in CRC cells may promote tumor immune evasion through the activation of YAP signaling[4] or the production of proprotein convertase subtilisin/kexin type 9 (PCSK9)[3]. Despite the above advances, the roles and mechanisms of CRC cells utilizing TME-derived cholesterol to drive metabolic changes that facilitate immune escape have yet to be investigated in depth.

Vacuolar-type ATPase (V-ATPase) is a highly evolutionarily conserved proton pump that is essential for the maintenance of vesicle pH and cholesterol homeostasis in numerous cell types[5]. It is a protein complex that can take numerous assembling forms, being composed of 14 subunits, each of which is an independently translated protein with multiple subtypes[5,6]. Individual subunits of V-ATPase complex have unique regulatory functions[7–9]. Some V-ATPase subunit subtypes, including ATP6V0A2, ATP6V0A3, ATP6V1C1, and ATP6V1E1, are highly expressed in a variety of tumors, including melanoma and breast cancer, where they promote metastasis and drug resistance by regulating TME-derived pH and cellular mTOR pathway[7,8,10,11]. The N terminal fragment of ATP6V0A2 released by tumor cells regulates the immunosuppressive phenotype of macrophages and neutrophils in the TME of breast and cervical tumors[9,12]. Despite the above advances, the role and mechanisms of tumor cell-intrinsic V-ATPase subunits regulating immune escape are still not well understood. Moreover, in CRC, differing effects on prognosis have been reported to result from altered ATP6V0A1 mRNA levels[13], but the biological functions of V-ATPase subunits, including ATP6V0A1, remain unknown.

Here, we explored the role of tumor cell-intrinsic V-ATPase subunits in regulating anti-tumor immune activity via lipid metabolism in CRC. Initially, the bulk RNA-sequencing data of The Cancer Genome Atlas-Colon Adenocarcinoma (TCGA-COAD) cohort was analyzed to investigate the correlations between V-ATPase subunits and lipid metabolism or immune activities. We found that the V-ATPase subunit ATP6V0A1 was positively correlated with the lipid metabolism scores and inversely correlated with the immune activity in CRC. Moreover, the correlation of ATP6V0A1 and immune activity was observed in the CRC samples with high lipid metabolism but not in those with low lipid metabolism. Importantly, we revealed tumor cell-intrinsic ATP6V0A1 as a novel immunosuppressive factor that promotes RABGEF1-dependent cholesterol absorption in CRC cells, consequently initiating paracrine TGF-β1/SMAD3 signaling to deactivate memory CD8+ T cells. In addition, we explored potential strategies for immunotherapy targeting ATP6V0A1. Our findings may offer new insights for studying the roles of CRC cells utilizing TME-derived cholesterol to drive immunosuppression, thus providing new directions for CRC immunotherapy.

## Results

### ATP6V0A1 contributes to an immunosuppressive TME through exogenous lipids in CRC

A high-fat diet (HFD) is known to induce an immunosuppressive TME in CRC, but the mechanism remains unclear. V-ATPase subunits are essential for lipid homeostasis in various cell types[5]. To investigate the potential roles of V-ATPase subunits in obesity-suppressed anti-tumor immunity in CRC, we stratified 471 TCGA-COAD samples into two clusters with high and low lipid metabolism according to their transcriptomic levels of lipid metabolism pathways (Supplementary Fig. 1A). The expression levels of 24 human V-ATPase subunits were compared between these two clusters. As shown in Supplementary Fig. 1B, the expression of 11 V-ATPase subunits, including ATP6V0A1,

ATP6V0C, ATP6V0D1, ATP6V0D2, ATP6V0E1, ATP6V0E2, ATP6V1B1, ATP6V1D, ATP6V1E1, ATP6V1F, and ATP6AP1, were significantly increased in the CRCs with high lipid metabolism. In parallel, the TCGA-COAD samples were divided into three groups according to immune scores (Supplementary Fig. 2A). Among the 11 V-ATPase subunits correlated with high lipid metabolism, ATP6V0A1 showed the strongest inverse correlation with immune activity. This was evident in COAD samples, as ATP6V0A1 was the only subunit that consistently decreased as the immune scores progressively increased from low to high levels (Supplementary Fig. 2B). Importantly, ATP6V0A1 expression was found to be positively associated with decreased immune activity in CRCs with high lipid metabolism, while no correlation was observed in CRCs with low lipid metabolism (Supplementary Fig. 3). On the other hand, high lipid metabolism was correlated with lower immune activity in CRCs with high ATP6V0A1 expression, while the immune activity was comparable between subpopulations with high and low lipid metabolism separately in low-ATP6V0A1 CRCs (Supplementary Fig. 4). Increased lipid metabolism in tumor cells is consistently associated with elevated lipid levels in the TME[1]. The above data suggested that ATP6V0A1 might be essential for exogenous lipid-suppressed anti-tumor immunity in CRC. To test this hypothesis, we investigated the roles of ATP6V0A1 in exogenous lipid-induced CRC immune evasion using an MC38 tumor model in mice with obesity (Fig. 1A–D). As shown in Fig. 1, HFD promoted the growth of MC38 tumors (Fig. 1E, F) and suppressed the tumor-killing activity of their TILs (Fig. 1G, H). Notably, the depletion of ATP6V0A1 significantly attenuated the alterations induced by the HFD (Fig. 1E–H), indicating that ATP6V0A1 plays a crucial role in regulating CRC immune-evasion through the utilization of exogenous lipid in the TME.

### Tumor cell-intrinsic ATP6V0A1 promotes the growth of CRC in an immune-dependent manner

We initially evaluated the expression level of ATP6V0A1 in CRC tumor cells. As shown in Supplementary Fig. 5A, B, ATP6V0A1 protein levels were significantly higher in 6 human CRC cell lines (CACO2, HCT-8, HCT-116, HT-29, RKO, and SW620) than in normal human colon epithelial NCM460 cells. ATP6V0A1 was also overexpressed in 2 murine CRC cell lines (MC38 and CT26) compared with normal murine colon (Supplementary Fig. C). We then stably knocked down *Atp6v0a1/ATP6V0A1* in mouse (MC38 and CT26) and human (HCT8) CRC cell lines, and also created an Atp6v0a1-overexpressing clone in MC38 cells (Supplementary Fig. 5D, E). Subsequently, we employed these modified cells to investigate the impact of tumor cell-intrinsic ATP6V0A1 on suppressing anti-tumor immunity in syngeneic and xenograft mouse models of CRC (Supplementary Fig. 6).

CCK8 assays revealed that the growth of MC38 cells in vitro was not affected by either *Atp6v0a1* knockdown or *Atp6v0a1* overexpression (Fig. 2A). Interestingly, *Atp6v0a1* knockdown in MC38 cells significantly suppressed tumor growth (Fig. 2B, C) and restored the tumor-killing activity of TILs (Fig. 2D) in immunocompetent C57BL/6 J mice, whereas *Atp6v0a1* overexpression had the opposite effect (Fig. 2B–D). The alterations in tumor growth caused by either ATP6V0A1 depletion or overexpression were completely abolished or significantly reduced in immunodeficient Rag2−/−Il2rg−/− or NOD/SCID mice (Fig. 2E–H). These data demonstrated that ATP6V0A1 enhances MC38 tumor growth by inhibiting the anti-tumor immune response. Similar to the results observed in MC38 cells, knockdown of *Atp6v0a1* in CT26 cells did not suppress their growth rate in vitro (Supplementary Fig. 7A). Moreover, depletion of ATP6V0A1 almost completely inhibited CT26 tumor growth in immunocompetent BALB/c mice, while only slightly retarded tumor growth in immunodeficient BRG or NOD/SCID mice (Supplementary Fig. 7B–D). Notably, at the respective termination time points, tumor weights were comparable for control CT26 tumors in BALB/c mice and immunodeficient mice but significantly lower for CT26-sh*v0a1* tumors in BALB/c mice than in

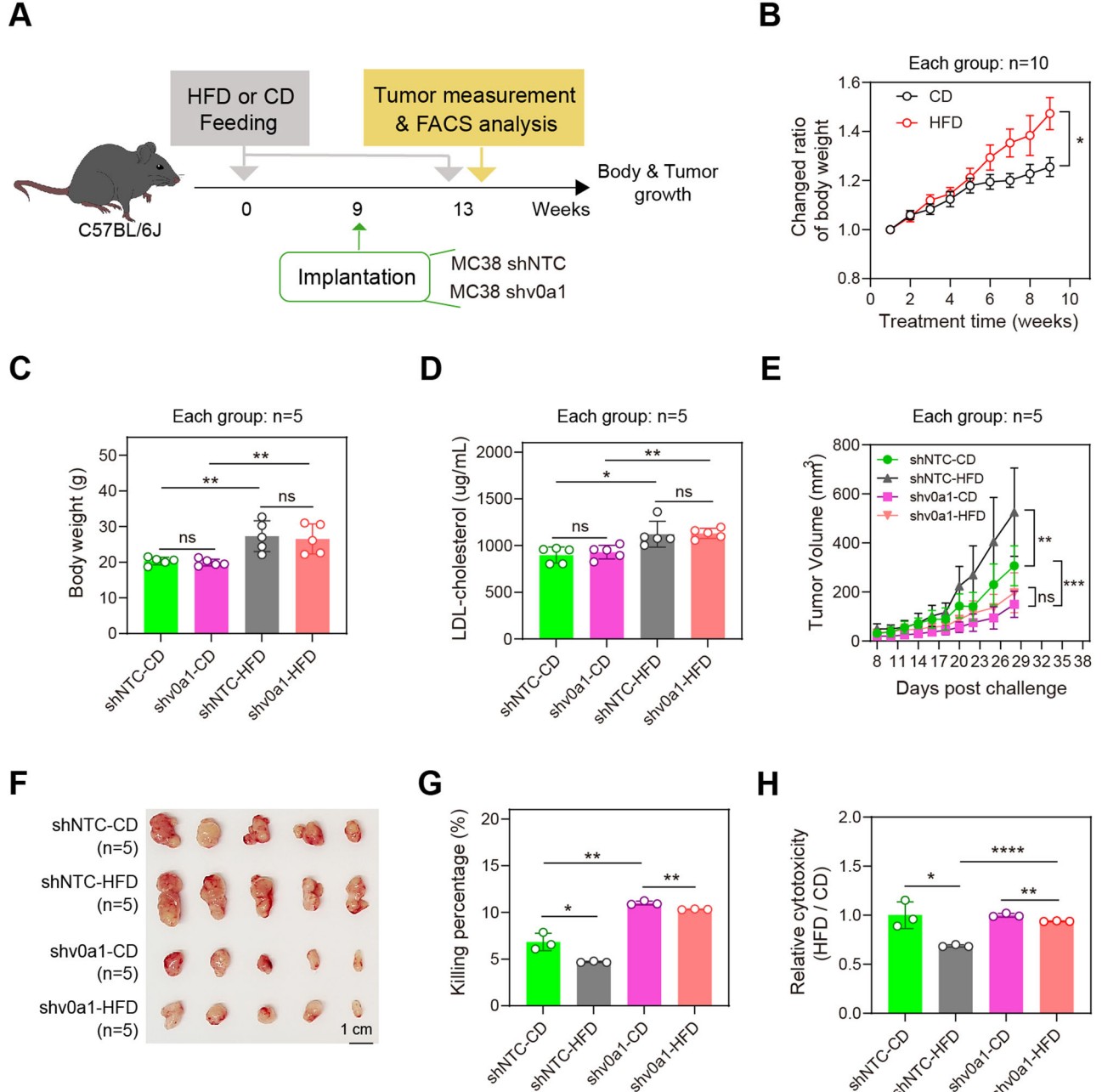

**Fig. 1 | ATP6V0A1 is required for HFD-induced suppression of anti-tumor immunity. A** Schematic diagram showing an animal model to investigate the significance of CRC cell-derived ATP6V0A1 in the suppression of anti-tumor immunity induced by elevated level of exogenous lipids. **B–D** C57BL/6 J mice were fed with high-fat diet (HFD) or control diet (CD) for 9 weeks prior to tumor cell implantation, and the body weight was measured (**B**); using a body-weight randomization grouping approach, HFD mice and CD mice were separately divided into two groups with similar body weights (**C**) and comparable serum levels of LDL-cholesterol (**D**). **E–H** Control MC38 (shNTC) cells or *ATP6v0a1*-knockdown (shv0a1) MC38 cells were subcutaneously injected to HFD mice and CD mice as shown in (**A**). Tumor volumes were monitored using calipers, and average tumor growth curves were plotted (**E**); photographs of the tumors are shown in (**F**). Tumor-infiltrating lymphocytes (TILs) were isolated and co-cultured with CFSE-labeled MC38 cells, and the cell mixture was analyzed by flow cytometry for the proportion of tumor cell death to assess the killing activities of TILs (**G**). The relative cytotoxicity of TILs between CD-treated and HFD-treated MC38-shNTC tumors or between CD-treated and HFD-treated MC38-shv0a1 tumors (**H**) were assessed by calculating the ratio of tumor cell-death proportion (**G**) between the cell mixture containing TILs from these tumors. For all experiments, data are shown as means ± s.e.m; *$p < 0.05$, **$p < 0.01$, ***$p < 0.001$, ****$p < 0.0001$. Statistical significance was determined using ordinary two-way ANOVA (in **B**, **E**) or unpaired two-sided Student's t-test (in **C**, **D**, **G**, **H**). $n = 10$ (**B**), 5 (**C–F**), or 3 (**G**, **H**) mice in each group; Data representative three independent experiments (**B–H**). Source data and exact *p*-value are provided as a Source Data file.

immunodeficient mice, showing that *Atp6v0a1* interference exhibits greater efficacy in suppressing the growth of CT26 tumors in immunocompetent mice than in immunodeficient mice (Supplementary Fig. 7B–D). These findings suggested that immune responses are essential for regulating CT26 tumor growth by ATP6V0A1, although other factors may also be involved.

To investigate the role of ATP6V0A1 in anti-tumor immunity in the context of human immune and colon cancer cells, we established a subcutaneous human colon cancer model in immunodeficient mice engrafted with human-derived immune cells (Supplementary Fig. 6B, C). Analysis of cell proliferation/tumor growth showed that *ATP6V0A1* knockdown in human HCT8 colon

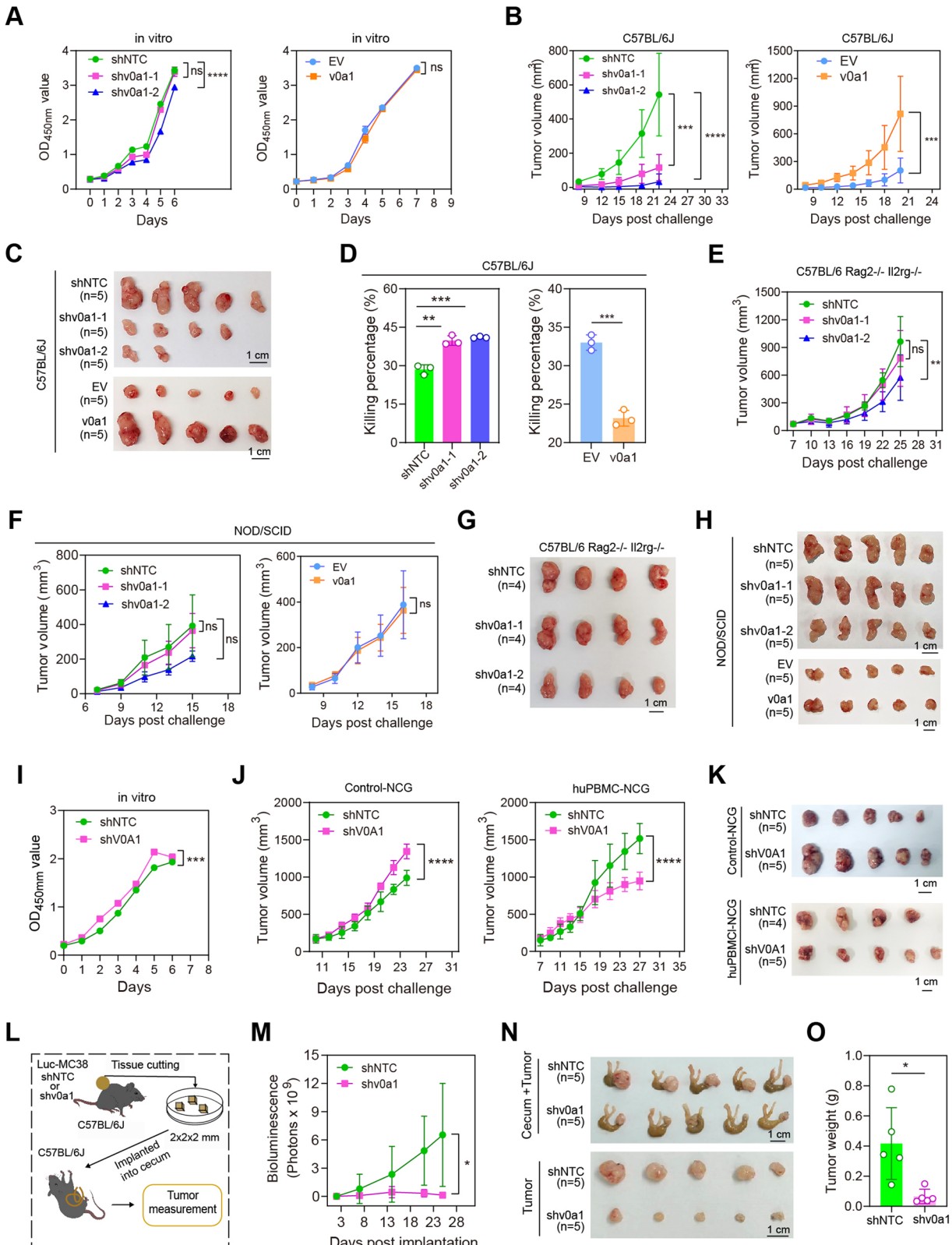

cancer cells slightly (but significantly) increased their proliferation in vitro (Fig. 2I) and in immune-deficient NCG mice (Fig. 2J, K) but significantly suppressed their growth in immune-reconstituted huPBMC-NCG mice (Fig. 2J, K). These results suggested that ATP6V0A1 also promoted immune evasion in human colon cancers. Next, to test whether tumor cell-intrinsic ATP6V0A1 could promote the development of orthotopic CRCs, we established a cecal MC38 tumor model (Fig. 2L). Similar to the subcutaneous tumor model, knockdown of *Atp6v0a1* in MC38 cells also significantly suppressed in vivo tumor growth in the cecal model (Fig. 2M–O). Collectively, these results indicate that tumor cell-intrinsic ATP6V0A1 suppresses anti-tumor immune responses and thus promotes tumor progression across different colon cancer models.

**Fig. 2 | ATP6V0A1 promotes the growth of colorectal cancers in an immune-dependent manner. A** The effect of suppressing or overexpressing Atp6v0a1 on the proliferation rates of MC38 cells in vitro was analyzed by CCK8 assay. $n = 3$ independent experiments. **B–D** C57BL/6 J mice were subcutaneously injected with Atp6v0a1-suppressing or Atp6v0a1-overexpressing MC38 cells as shown in Supplementary Fig. 6A. Tumor volumes were monitored using calipers, and average tumor growth curves were plotted (**B**); photographs of the tumors are shown in (**C**). $n = 5$ mice per group. TILs isolated from the tumors described in (**B**) and (**C**) were co-cultured with CFSE-labeled MC38 cells, and the cell mixture was analyzed by flow cytometry for the proportion of tumor cell death to assess the killing activities of TILs (**D**). $n = 3$ mice in each group; Data representative three independent experiments. **E–H** Atp6v0a1-suppressing or Atp6v0a1-overexpressing MC38 cells were subcutaneously injected into C57BL/6 Rag2$^{-/-}$Il2rg$^{-/-}$ mice (**E, G**) or NOD/SCID mice (**F, H**) as shown in supplementary Fig. 6A. Average tumor growth curves were plotted (**E, F**); photographs of the tumors are shown in (**G, H**). $n = 4$ (**E, G**) or 5 (**F, H**) mice per group. **I** The proliferation of control HCT-8 (shNTC) cells and *ATP6V0A1*-knockdown (shV0A1) HCT-8 cells in vitro was analyzed by CCK8 assay. $n = 3$ independent experiments. **J, K** HCT-8 shNTC cells or HCT-8 shV0A1 cells were subcutaneously injected into huPBMC-NCG mice and control NCG mice as shown in Supplementary Fig. 6B. Average tumor growth curves were plotted (**J**); photographs of the tumors are shown in (**K**). $n = 4$ (shNTC group in huPBMC-NCG mice model) or 5 mice (other groups) in each group. **L–O** Cecal MC38 tumors were established as illustrated in (**L**). Tumor growth was monitored via bioluminescence detection, and average tumor growth curves were plotted ($n = 5$ mice per group; **M**). The mice were sacrificed on day 25; photographs of the tumors in the cecum (**N**, upper) and the isolated tumors (**N**, lower) are shown. Tumor weights for the indicated groups are compared in (**O**). For all experiments, data are shown as means ± s.e.m; *$p < 0.05$, **$p < 0.01$, ***$p < 0.001$, ****$p < 0.0001$. Statistical significance was determined using ordinary two-way ANOVA (**A, B, E, F, I, J, M**) or unpaired two-sided Student's $t$-test (**D, O**). Source data and exact p-value are provided as a Source Data file.

## Tumor cell-intrinsic ATP6V0A1 suppresses memory CD8$^+$ T cells activity in CRC

To determine the type of immune cells contributing to ATP6V0A1-mediated immune evasion, we used single-cell transcriptome sequencing (scRNA-seq) to analyze the ATP6V0A1-edited immune microenvironment in CRC. Holistic immune profiles in *Atp6v0a1* knockdown (shv0a1, $n = 5$) and control MC38 tumors (shNTC, $n = 5$) were compared (Supplementary Figs. 8, 9) as a strategy described in methods. The results showed that *Atp6v0a1* interference in MC38 cells increased the percentages of naïve T, effector T, and memory-like T-2 cells in the total T cell population while decreasing the percentages of exhausted T, Treg, and memory-like T-1 cells (Supplementary Fig. 9A, B). Moreover, the percentages of granulocytes, monocytes, and dendritic cells within the myeloid population were increased in *Atp6v0a1* knockdown-derived tumors, whereas the percentage of macrophages was decreased (Supplementary Fig. 9C, D). Interestingly, memory-like T-2 cells were detected in shv0a1 but not shNTC MC38 tumors (Supplementary Fig. 9A, B). Of all the immune cell subclusters, memory-like T-2 cells exhibited the most significantly increased cell count ratio in *shv0a1* tumors vs. control tumors (Supplementary Fig. 9E). In addition, memory-like T-2 cells expressed more effector markers than memory-like T-1 cells (Supplementary Fig. 8D). Consistent with these findings, the expression levels of TCR signaling molecules and T-cell activation markers in total memory-like T cells were higher in shv0a1 tumors than in shNTC tumors (Supplementary Fig. 9F). These data suggest that MC38-derived ATP6V0A1 mainly regulates anti-tumor immunity by suppressing the effectiveness of memory T cells within the CRC TME.

Memory-like T-2 cells preferentially express CD8 rather than CD4 (Supplementary Fig. 8D). Therefore, we initially examined the role of CD8$^+$ T cells in ATP6V0A1-regulated immune evasion in the MC38 allograft model (Fig. 3A). Depletion of CD8$^+$ T cells via administration of monoclonal anti-CD8α (Clone 2.43) overcame the suppression/acceleration of MC38 tumor growth induced by the knock-down/over-expression of MC38-derived *Atp6v0a1* (Fig. 3B, C). This finding suggested that MC38-derived ATP6V0A1 promoted tumor immune evasion mainly by suppressing CD8$^+$ T-cell activity. Next, we used a flow cytometry (FC) strategy (Fig. 3D) to confirm whether ATP6V0A1 selectively affected the activity of memory CD8$^+$ T cells in murine colon cancers. For these assays, we used CD44 as a protein marker of murine memory CD8$^+$ T cells[14]. FC analysis also showed that BCL-2, another specific marker of memory-like T cells identified in the scRNA-seq analysis (Supplementary Fig. 8D), was most frequently detected in CD44$^+$CD8$^+$ rather than CD44$^-$CD8$^+$ T cells (Fig. 3E); this finding indicated that CD44$^+$CD8$^+$ T cells may represent the T-cell populations annotated as memory-like T cells in our scRNA-seq analysis. FC analysis of effector cytokines showed that knockdown of *Atp6v0a1* in both MC38 and CT26 cells significantly enhanced the levels of perforin,

GzmB, and IFN-γ in CD44$^+$CD8$^+$ T cells derived from the corresponding subcutaneous tumors (Fig. 3F, G). By contrast, the levels of these cytokines in CD44$^-$CD8$^+$ T cells were not consistently enhanced in MC38 tumors and CT26 tumors with *Atp6v0a1* knockdown (Fig. 3F, G). Consistent with these findings, *ATP6V0A1* knockdown in HCT8 cells significantly enhanced the effectiveness of memory CD45RO$^+$CD8$^+$ T cells but not that of CD45RO$^-$CD8$^+$ T cells in hPBMC-NCG mouse-derived HCT8 tumors (Fig. 3H). Importantly, the levels of cytotoxic cytokines in CD44$^+$CD8$^+$ T cells were also significantly enhanced in the orthotopic (cecal) MC38 tumor model as a result of *Atp6v0a1* knockdown (Fig. 3I). Similar to the subcutaneous MC38 tumor model, knockdown of tumor cell-intrinsic *Atp6v0a1* did not consistently enhance the level of the cytotoxic cytokines in CD44$^-$CD8$^+$ T cells derived from the cecal tumor model (Fig. 3I).

Collectively, these data suggest that ATP6V0A1 expressed in CRC tumor cells promotes immune evasion mainly by suppressing the effectiveness of memory CD8$^+$ T cells.

## TGF-β1/SMAD3 paracrine axis links tumor cell-intrinsic ATP6V0A1 to the suppression of memory CD8$^+$ T cells activity

To explore the molecular mechanisms underlying the cross-talk between ATP6V0A1 and memory CD8$^+$T cells, we first used the scRNA-seq data to identify genes that were differentially expressed in memory-like T cells from MC38-shNTC and MC38-shv0a1 tumors. Transcriptomic comparison showed that following *Atp6v0a1* interference in MC38 cells, the level of *Smad3* mRNA was significantly reduced and the level of *Id2* mRNA was enhanced in memory-like T cells, relative to the levels in other T-cell subpopulations (Fig. 4A). *Id2* has been reported to be repressed by SMAD3 signaling[15]. Moreover, western blotting showed decreases in total SMAD3 and phosphorylated SMAD3 (p-SMAD3) protein levels and an increase in ID2 protein levels in CD8$^+$T cells from MC38-shv0a1 tumors, compared with levels in MC38-shNTC tumors (Fig. 4B and Supplementary Fig. 10A). SMAD3 signaling is often activated by the TGF-β/TGF-βR axis[16]. Interestingly, *Tgfbr2* mRNA was more highly expressed in memory-like T-2 cells than in other T cell subclusters (Fig. 4C). Consistent with this finding, TGF-βRII protein levels in wild-type MC38 tumors were higher in CD44$^+$CD8$^+$ T cells than in CD44$^-$CD8$^+$ T cells (Fig. 4D). Meanwhile, *Tgfb1* mRNA levels and TGF-β1 protein levels were significantly lower in *Atp6v0a1*-suppressing MC38 cells than in control cells (Fig. 4E and Supplementary Fig. 10B); Overexpression of *Atp6v0a1* can restore the decreased TGF-β1 levels induced by *Atp6v0a1* knockdown (Fig. 4F). IHC analysis confirmed that intra-tumoral TGF-β1 levels were lower in MC38-shv0a1 tumors than in MC38-shNTC tumors (Supplementary Fig. 10C, D). The above data indicated that tumor-intrinsic ATP6V0A1 may regulate memory CD8$^+$ T cells via paracrine TGF-β1/SMAD3-signaling. Importantly, treatment of MC38 tumor-derived TILs with the SMAD3 inhibitor (E)-SIS3 significantly enhanced the expression of

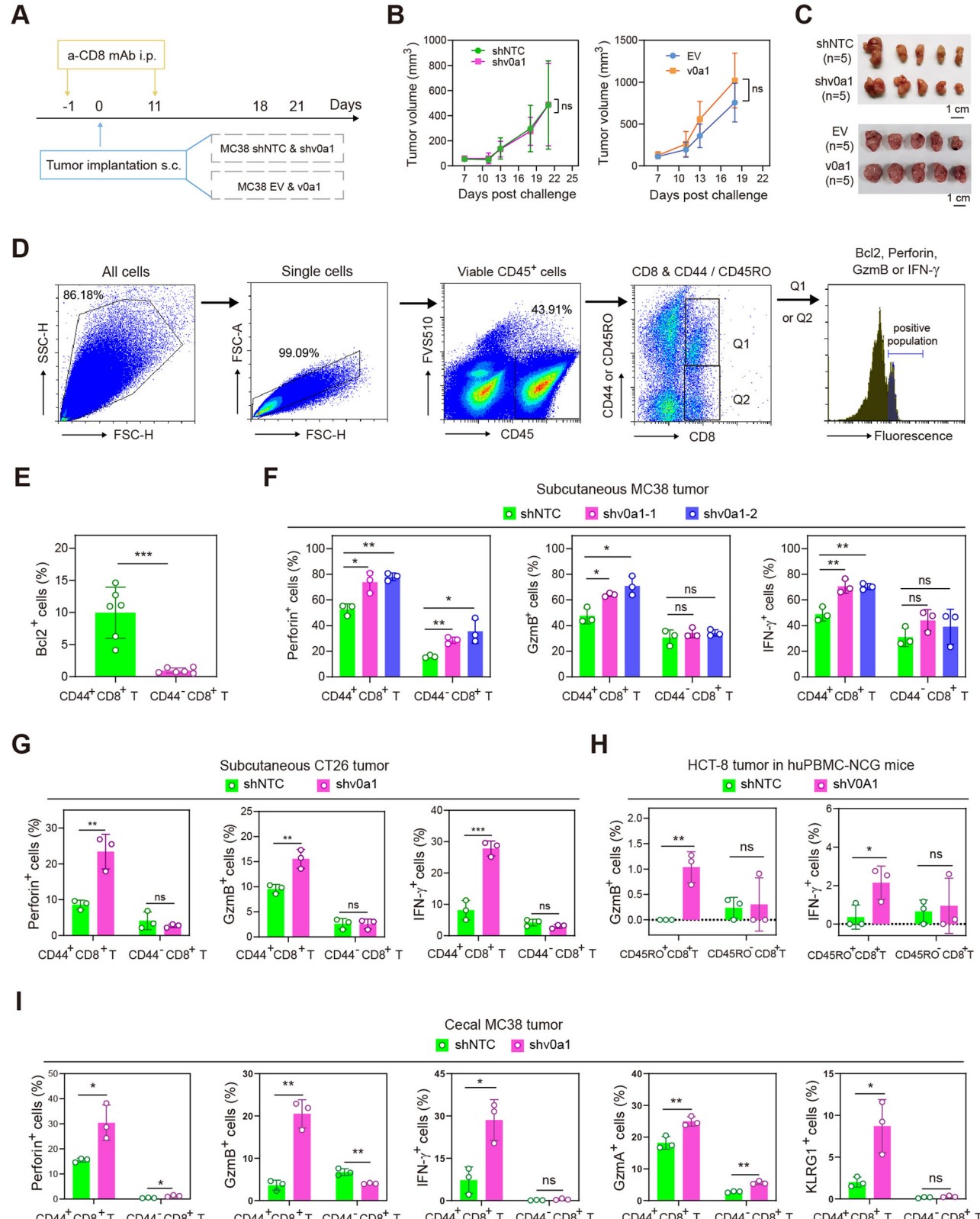

effector cytokines in CD44⁺CD8⁺ T cells (Supplementary Fig. 11 and Fig. 4G), confirming the critical role of SMAD3 activation in suppressing memory CD8⁺ T cell function. Additionally, pretreatment of MC38 cell culture medium with anti-TGF-β1 resulted in significantly increased expression of effector cytokines in CD44⁺CD8⁺ T cells but not in CD44⁻CD8⁺ T cells in the in vitro culture of CD8⁺ T cells with MC38 cell culture medium (Fig. 4H). The expression of effector cytokines was also significantly higher in CD44⁺CD8⁺ T cells treated with conditioned medium (CM) from *Atp6v0a1* knockdown MC38 cells but not in those treated with CM from control MC38 cells, while pretreatment of the CM with recombinant mouse TGF-β1 eliminated this enhancement (Fig. 4I). These data indicated the important role of TGF-β1/SMAD3-signaling in the suppression of memory CD8⁺ T cell function by MC38-derived Atp6v0a1. We went on to further confirm the contribution of

**Fig. 3 | Memory CD8⁺ T cell effectiveness is important for the regulation of CRC growth by tumor-derived ATP6V0A1. A–C** CD8⁺ T cells were depleted from C57BL/6 mice in the indicated groups by injection of anti-CD8α mAb (clone 2.43) at the time points shown (**A**). Average tumor growth curves were plotted (**B**); photographs of the tumors are shown in (**C**). $n = 5$ mice per group. **D** Flow cytometry (FC) strategy for gating CD44⁺CD8⁺ and CD44⁻CD8⁺ T cells and detecting effector cytokines. **E** BCL2 protein levels were compared in CD44⁺CD8⁺ and CD44⁻CD8⁺ T cells by FC. $n = 6$; Data are pooled from 2 independent experiments. **F–I** The effectiveness of tumor-infiltrating CD44⁺CD8⁺ and CD44⁻CD8⁺ T cells

or CD45RO⁺CD8⁺ and CD45RO⁻CD8⁺ T cells was analyzed via FC in subcutaneous MC38 tumors from C57BL/6 J mice (**F**), subcutaneous CT26 tumors from BALB/c mice (**G**), subcutaneous HCT-8 tumors from huPBMC-NCG mice (**H**), and cecal MC38 tumors from C57BL/6 J mice (**I**). $n = 3$ mice per group; Data representative of 3 independent experiments. For all experiments, data are shown as means ± s.e.m; *$p < 0.05$, **$p < 0.01$, ***$p < 0.001$. Statistical significance was determined using ordinary two-way ANOVA (**B**) or unpaired two-sided Student's $t$-test (**E–I**). Source data and exact $p$-value are provided as a Source Data file.

TGF-β1 to Atp6v0a1-mediated regulation of tumor immune evasion in vivo. Experiments conducted in the syngeneic mouse tumor model showed that anti-TGF-β1 treatment suppressed the enhancement of MC38-tumor growth promoted by *Atp6v0a1* overexpression (Fig. 4J–L). By contrast, injection of recombinant mouse TGF-β1 protein into MC38 tumors rescued the growth suppression caused by *Atp6v0a1* knockdown (Fig. 4M–O). Importantly, TGF-β1 supplementation also inhibited the enhancement of CD44⁺CD8⁺ T cell activity mediated by *Atp6v0a1* knockdown (Fig. 4O). Moreover, the restoration of Tgfb1 expression in MC38 tumor cells (Supplementary Fig. 12A) had a comparable impact to exogenous TGF-β1 in mitigating the tumor suppression (Supplementary Fig. 12B–D) and CD44⁺CD8⁺ T cell activation (Supplementary Fig. 12E) induced by *Atp6v0a1* depletion. It is of note that the suppression of *Tgfb1/TGFB1* in MC38 /HCT-8 cells did not significantly affect the growth of these cells (Supplementary Fig. 12F, G). Collectively, our data suggest that the TGF-β1/SMAD3 axis is critical for the regulation of the anti-tumor activities of CD44⁺CD8⁺ T cells by CRC cell-expressed ATP6V0A1.

## Enhanced level of cholesterol and its 24-OHC production in the ER mediates ATP6V0A1-induced upregulation of TGF-β1

To explore the molecular mechanisms underlying ATP6V0A-mediated TGF-β1 upregulation in CRC cells, we applied a quantitative proteomics approach to investigate the proteins associated with ATP6V0A1. As previously mentioned, lipid metabolic reprogramming is essential for the inhibition of anti-tumor immunity through ATP6V0A1. Notably, a key molecular cluster related to cholesterol metabolism signaling was significantly altered in MC38-shv0a1 cells compared to the control MC38 cells (Fig. 5A). In addition, LDLR was upregulated considerably in MC38-shv0a1 cells (Fig. 5B, C; Supplementary Fig. 13). Cholesterol metabolism is a vital aspect of lipid metabolism associated with regulating TGF-β1 expression[17]. The mechanism underlying the expression of TGF-β1 induced by cholesterol metabolism remains unclear. Therefore, we aimed to investigate whether ATP6V0A1 could control the expression of TGF-β1 through reprogramming cholesterol metabolism. LDLR upregulation is suggested to indicate a decrease in the levels of oxysterols, particularly 24-OHC, produced in the ER[18,19]. Indeed, *Atp6v0a1/ATP6V0A1* knockdown in MC38/HCT-8 cells significantly reduced the production of 24-OHC in cell culture (Fig. 5D). Restoration of *Atp6v0a1/ATP6V0A1* expression in ATP6V0A1-deficient MC38/HCT-8 cells have the potential to reverse the decreased levels of 24-OHC (Fig. 5E). 24-OHC is a typical LXR agonist that exerts its biological functions by activating LXR signaling[19]. Consistently, *ATP6V0A1* knockdown in HCT-8 cells reduced LXR activity as demonstrated by the LXR luciferase reporter assays, while 24-OHC treatment could completely rescue this decreased LXR activity (Fig. 5F). Moreover, treatment with GSK2033, an LXR inhibitor, suppressed the effects of 24-OHC on inducing LXR activation (Fig. 5F). These data suggested that ATP6V0A1 drives the activation of 24-OHC/LXR pathway. We further explored the roles of 24-OHC/LXR pathway in ATP6V0A1-induced TGF-β1 expression. Exogenous treatment with 24-OHC partially reversed the suppression of TGF-β1 levels in *Atp6v0a1*-suppressing MC38 cells, while completely reversing it in *ATP6V0A1*-suppressing HCT-8 cells (Fig. 5G). Additionally, treatment with GSK2033 reversed the impact of 24-OHC on TGF-β1 levels in ATP6V0A1 knockdown cells (Fig. 5G). Similarly,

downregulation of Nr1h3 (LXRα) or Nr1h2 (LXRβ) also inhibited the effects of 24-OHC on increasing TGF-β1 levels. Combining deficiencies in LXRα and LXRβ yielded more effective results in counteracting the effects of 24-OHC compared to having only one of the deficiencies (Supplementary Fig. 14A), suggesting that both LXRα and LXRβ are involved in the process of ATP6V0A1 inducing TGF-β1 expression via 24-OHC production. These results showed that ATP6V0A1 promotes TGF-β1 expression via the 24-OHC/LXR pathway in CRC cells. 24-OHC is produced in the ER by the oxidation of cholesterol, catalyzed by CYP46A1[19]. However, the levels of CYP46A1 were not significantly altered by *Atp6v0a1* knockdown (Fig. 5C; Supplementary Fig. 13). Interestingly, the levels of ER-derived cholesterol were decreased in both *Atp6v0a1*-suppressing MC38 cells and *ATP6V0A1*-suppressing HCT-8 cells, relative to the levels in control cells (Fig. 5H). Moreover, treating HCT-8 cells with LDL to supplement cholesterol in the ER restored the decreased 24-OHC production (Supplementary Fig. 14B) and reduced TGF-β1 expression (Fig. 5I) caused by ATP6V0A1 knockdown. Treatment with MβCD to decrease cellular cholesterol can reverse the effects induced by LDL (Supplementary Fig. 14B; Fig. 5I). Importantly, suppressing 24-OHC production via CYP46A1 knockdown (Supplementary Fig. 14B) effectively counteract the TGF-β1 expression increased by exogenous LDL treatment (Fig. 5I). Collectively, these data demonstrated that ATP6V0A1 promotes 24-OHC production mainly by enhancing cholesterol levels in the ER, thereby increasing TGF-β1 expression via the LXR pathway (Fig. 5J).

## ATP6V0A1 facilitates the transportation of exogenous cholesterol to the ER via the RABGEF1-dependent endosome maturation pathway

Having shown that ATP6V0A1 mediated the upregulation of TGF-β1 expression by reprogramming cholesterol metabolism in CRC cells, we wanted to investigate the mechanism by which ATP6V0A1 enhanced cholesterol levels in the ER. First, we showed that *Atp6v0a1* knockdown in MC38 cells did not significantly alter the level of HMGCR, the key rate-limiting enzyme for endogenous cholesterol synthesis (Supplementary Fig. 15). Meanwhile, quantitative proteomics and western blotting analysis revealed that the RABGEF1 endosome maturation pathway, which is critical for vesicular transport[20], was suppressed by *Atp6v0a1/ATP6V0A1* knockdown in MC38 cells and HCT-8 cells (Fig. 6A–C; Supplementary Fig. 16A, B). On the other hand, the RABGEF1 level was increased by Atp6v0a1 overexpression in MC38 cells (Fig. 6D and Supplementary Fig. 16C). Importantly, confocal microscopy showed that *Atp6v0a1* knockdown in MC38 cells resulted in significantly decreased RAB7a expression in endosomes and decreased VPS41 levels in RAB7a+ endosomes (Fig. 6E, F). Moreover, the reduced levels of endosomal RAB7a in *Atp6v0a1* knockdown cells were restored following *Rabgef1* overexpression in MC38 cells (Supplementary Fig. 17A, B). RAB7a and VPS41 are the markers of late endosomes, and are required for endosomes to mature into degradative lysosomal compartments[20]. These data demonstrated that ATP6V0A1 supports RABGEF1-dependent endosome maturation. Therefore, ATP6V0A1 may enhance ER cholesterol levels via the regulation of cholesterol transport. As well as being reduced in the ER (Fig. 5H), cholesterol levels were also decreased in endosomes and lysosomes in *Atp6v0a1*-suppressing MC38 cells and *ATP6V0A1*-

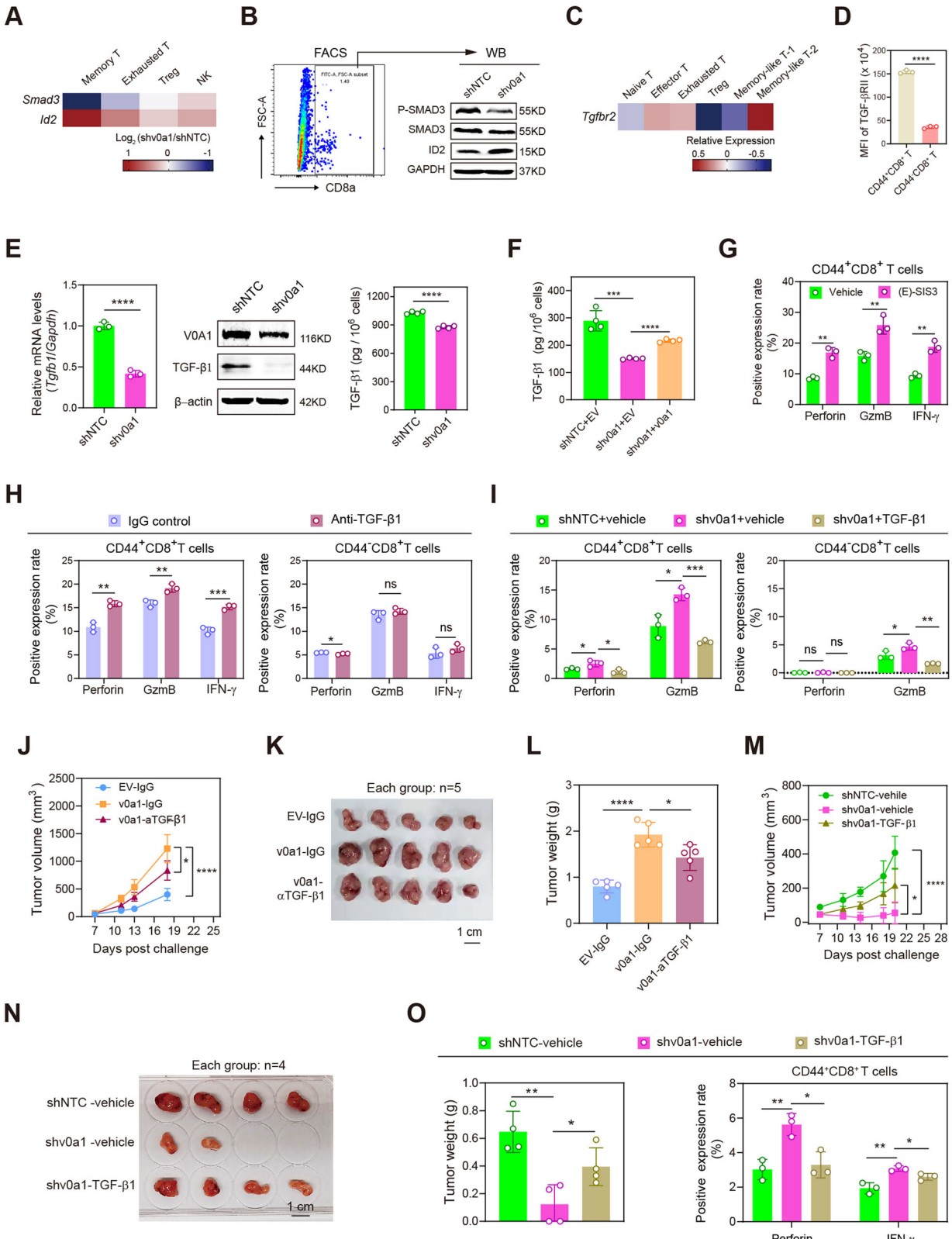

suppressing HCT-8 cells (Supplementary Fig. 17C, D), indicating that ATP6V0A1 may support the absorption of exogenous cholesterol and thus their entering into ER. To explore this possibility, we treated HCT-8-shNTC and HCT-8 shV0A1 cells with exogenous Dil-LDL. Interestingly, *ATP6V0A1* knockdown reduced the levels of exogenous Dil-LDL entering both RAB7a+ late endosomes (Fig. 6G, H) and LAMP1+ lysosomes (Fig. 6I, J) in HCT-8 cells and enhanced the localization of

Dil-LDL within RAB35+ recycling endosomes (Supplementary Fig. 17E, F). Consistent with these findings, in HCT-8 cells treated with Dil-LDL for different lengths of time, *ATP6V0A1* knockdown-induced reductions in intracellular Dil-LDL levels were detected at later time points (6 h and 8 h), but not at an earlier timepoint (4 h; Supplementary Fig. 18A, B). Moreover, the level of recycling of Dil-LDL to the culture medium in HCT-8 cells was increased by *ATP6V0A1* knockdown

**Fig. 4 | Tumor cell-intrinsic ATP6V0A1 suppresses memory CD8+ T cells via the TGF-β1/SMAD3 axis. A** Atp6v0a1 knockdown-induced expression changes in *Smad3* and *Id2* were analyzed in the indicated T-cell subpopulations based on the scRNA-seq data (Supplementary Fig. 8). **B** Western blotting detecting the indicated protein levels in CD8+ T cells isolated from MC38-shNTC and MC38-shv0a1 tumors. The samples derive from the same experiment but different gels for SMAD3, ID2, GAPDH, another for p-SMAD3 were processed in parallel. **C** scRNA-seq data comparing the expression of *Tgfbr2* in different T-cell subpopulations. **D** FC-analysis comparing TGF-βRII level between CD44+CD8+ and CD44-CD8+ T cells in wild-type MC38 tumors. **E, F** The mRNA and protein levels of TGF-β1 in MC38 cells were analyzed by qPCR (**E**, left), Western blotting (**E**, medium), and ELISA (**E**, right and **F**). **G–I** Wild-type MC38 tumor-derived CD8+ T cells were treated with the following: (E)-SIS3 (**G**), MC38 cell culture medium plus control IgG or anti-TGF-β1 (**H**), conditioned medium from MC38-shNTC cells or MC38-shv0a1 cells with/without supplement of TGF-β1 protein (**I**). The activation of CD44+ CD8+ and

CD44-CD8+ T cells was determined by FC analysis. **J–L** MC38-EV, or MC38-v0a1 cells were subcutaneously injected into C57BL/6 J mice, followed by intraperitoneal injection of control IgG or anti-TGF-β1 on days 7, 10, and 13. Average tumor growth curves (**J**), photographs of the tumors (**K**), and a comparison of tumor weights on day 18 (**L**) are shown; *n* = 5 mice per group. **M–O** MC38-shNTC or MC38-shv0a1 cells were subcutaneously injected into C57BL/6 J mice; PBS or mouse TGF-β1 protein was injected intratumorally. Average tumor growth curves (**M**), photographs of the tumors (**N**), and a comparison of tumor weights on day 20 (**O**, left) are shown. TILs from day 20 tumors were analyzed for effector production in CD44+CD8+ T cells (**O**, right). *n* = 4 mice per group. For all experiments, data are shown as means ± s.e.m; *\*p* < 0.05, *\*\*p* < 0.01, *\*\*\*p* < 0.001, *\*\*\*\*p* < 0.0001. Statistical significance was determined using ordinary two-way ANOVA (in **J, M**) or unpaired two-sided Student's *t*-test (in **D, E, F, G, H, I, L, O**). *n* = 3 independent experiments for Fig. 4D–I; Three independent experiments were performed for Fig. 4B, E (medium). Source data and exact *p*-value are provided as a Source Data file.

(Supplementary Fig. 18C, D). These data demonstrated that ATP6V0A1 facilitates the subsequent transportation of exogenous LDL to the lysosomal degradation pathway, which enables the release of LDL-cholesterol after exogenous LDL enters CRC cells. The lysosomal degradation of LDL proteins is required for the transport of exogenous LDL-cholesterol to the ER[19]. Therefore, ATP6V0A1 can facilitates the increase of cholesterol levels in ER by enhancing the absorption of exogenous cholesterol.

We further explored the role of ATP6V0A1 in exogenous cholesterol-induced TGF-β1 expression. The depletion of ATP6V0A1 significantly counteracted the increased TGF-β1 level by LDL treatment (Fig. 7A), illustrating the necessity of ATP6V0A1 for exogenous cholesterol to upregulate TGF-β1 expression. RABGEF1-dependent endosome maturation is crucial for cholesterol absorption, since the inhibition of endosome maturation through *RABGEF1* knockdown hindered the transportation of exogenous LDL to late endosomes (Supplementary Fig. 19A, B). Importantly, inhibition of cholesterol absorption via treatment with U18666A, an inhibitor of cholesterol transport from lysosome to ER, or via *RABGEF1* knockdown attenuated the enhancement of TGF-β1 levels induced by exogenous LDL in HCT-8 cells (Fig. 7B). These data indicated that RABGEF1-dependent cholesterol absorption is necessary for exogenous cholesterol to induce TGF-β1 expression. Moreover, similar with ATP6V0A1 suppression, blockade of cholesterol absorption reduced TGF-β1 expression mainly by decreasing 24-OHC production (Supplementary Fig. 19C, D). We thus explore whether ATP6V0A1 regulates 24-OHC-mediated TGF-β1 expression via RABGEF1-dependent cholesterol absorption. Over-expression of Rabgef1 in ATP6V0A1-suppressing MC38 or CT26 cells or overexpression of RABGEF1 in ATP6V0A1-suppressing HCT-8 cells could restore the 24-OHC production and TGF-β1 expression reduced by *Atp6v0a1/ATP6V0A1* knockdown (Fig. 7C, D; Supplementary Fig. 20). Consistently, inhibiting cholesterol absorption with siRabgef1 mitigated the reduction in ER-cholesterol (Fig. 7E) and TGF-β1 (Fig. 7F) in Atp6v0a1-suppressing MC38 cells. Endosome maturation and acidification are required for RABGEF1-dependent cholesterol absorption. Consequently, we explored the functions of RAB7a, a key factor of endosome maturation, in ATP6V0A1-regulated TGF-β1 expression. Knockdown of Rab7 or RAB7A in MC38 or HCT-8 cells counteracted the decrease in both 24-OHC (Fig. 7G) and TGF-β1 (Fig. 7H) levels induced by ATP6V0A1 suppression, with a more pronounced effect in MC38 cells. Endosome acidification results from endosome maturation and further drives the endo-lysosomal traffic for cholesterol absorption. As expected, ATP6V0A1 suppression did not alter the cellular pH (Supplementary Fig. 21A) but enhanced the pH in RAB7a+ endosomes (Supplementary Fig. 21B, C). It is of note that ATP6V0A1 preferred to regulate the pH of RAB7a+ vesicles rather than RAB7a- vesicles (Supplementary Fig. 21B, D). Importantly, the blockade of endosome

acidification with Baf-A1 treatment eliminated the decrease in both 24-OHC and TGF-β1 levels induced by ATP6V0A1 suppression in HCT-8 cells (Supplementary Fig. 21E). Collectively, these data suggested that ATP6V0A1 facilitates 24-OHC-mediated TGF-β1 expression through the cholesterol absorption driven by RABGEF1-dependent endosome maturation.

Since the V-ATPase complex plays a role in vesicle trafficking, we next explored whether the regulation of the RABGEF1/TGF-β1 pathway by ATP6V0A1 relies on the changes in the expression level of the V-ATPase complex. As shown in Supplementary Fig. 22, upon the knockdown of ATP6V0A1, ATP6V0A3 (translated by TCIRG1) protein level was upregulated while ATP6V0A2 expression was not significantly changed (Supplementary Fig. 22A, B). ATP6V0A4 is relatively lowly expressed in both murine (Supplementary Fig. 22A) and human (Supplementary Fig. 1B) CRCs. Moreover, ATP6V0A1 suppression did not alter the protein levels of ATP6V0C and ATP6V1A (Supplementary Fig. 22A, B), which are associated with the V0 and V1 subcomplex levels of the V-ATPase, respectively. Therefore, the changes in ATP6V0A1 expression may not alter the total expression levels of the V-ATPase complex. Moreover, the suppression of other subtypes of the V0A subunit, such as ATP6V0A2 and ATP6V0A3, did not significantly affect the expression of both RABGEF1 and TGF-β1 (Supplementary Fig. 22C–F), and the level of ER-derived cholesterol (Supplementary Fig. 22G, H), suggesting that ATP6V0A1 may have distinct functions in regulating immune evasion compared to the other V0A subtypes. Collectively, ATP6V0A1-regulated RABGEF1-dependent cholesterol absorption and TGF-β1 expression do not rely on the changes in the expression level of the V-ATPase complex or its V0 subcomplex.

## ATP6V0A1 is positively correlated with RABGEF1, TGF-β1 and immunosuppressive TME in clinical CRC samples

According to the immunohistochemical (IHC) analysis of a human CRC tissue microarray (Fig. 8A, B), ATP6V0A1 exhibited significantly higher expression levels in tumor tissues compared to adjacent non-tumor tissues (paired *t*-test; *p* < 0.0001), regardless of dMMR and pMMR status. Moreover, ATP6V0A1 protein levels were significantly higher in patients with advanced-stage disease (Stages III + IV) than in patients with early-stage CRC (Stages I + II; Fig. 8C). Importantly, ATP6V0A1 protein levels were inversely correlated with the overall survival of CRC patients (Fig. 8D). Additionally, in a Tumor Immune Dysfunction and Exclusion (TIDE) analysis[21] based on the GSE38832 database, high infiltration of cytotoxic T lymphocytes (CTLs) was found to predict better survival in CRCs with low ATP6V0A1 levels (*p* = 0.0002), but not in those with high ATP6V0A1 levels (*p* = 0.3376; Fig. 8E), suggesting the roles of ATP6V0A1 in predicting T-cell dysfunction in CRC. Collectively, ATP6V0A1 is highly expressed in human CRCs and predicts inactive anti-tumor immunity and poor survival.

To study the correlations between tumor cell-intrinsic ATP6V0A1, tumor cell-derived TGF-β1, and memory CD8+ T cell effectiveness in

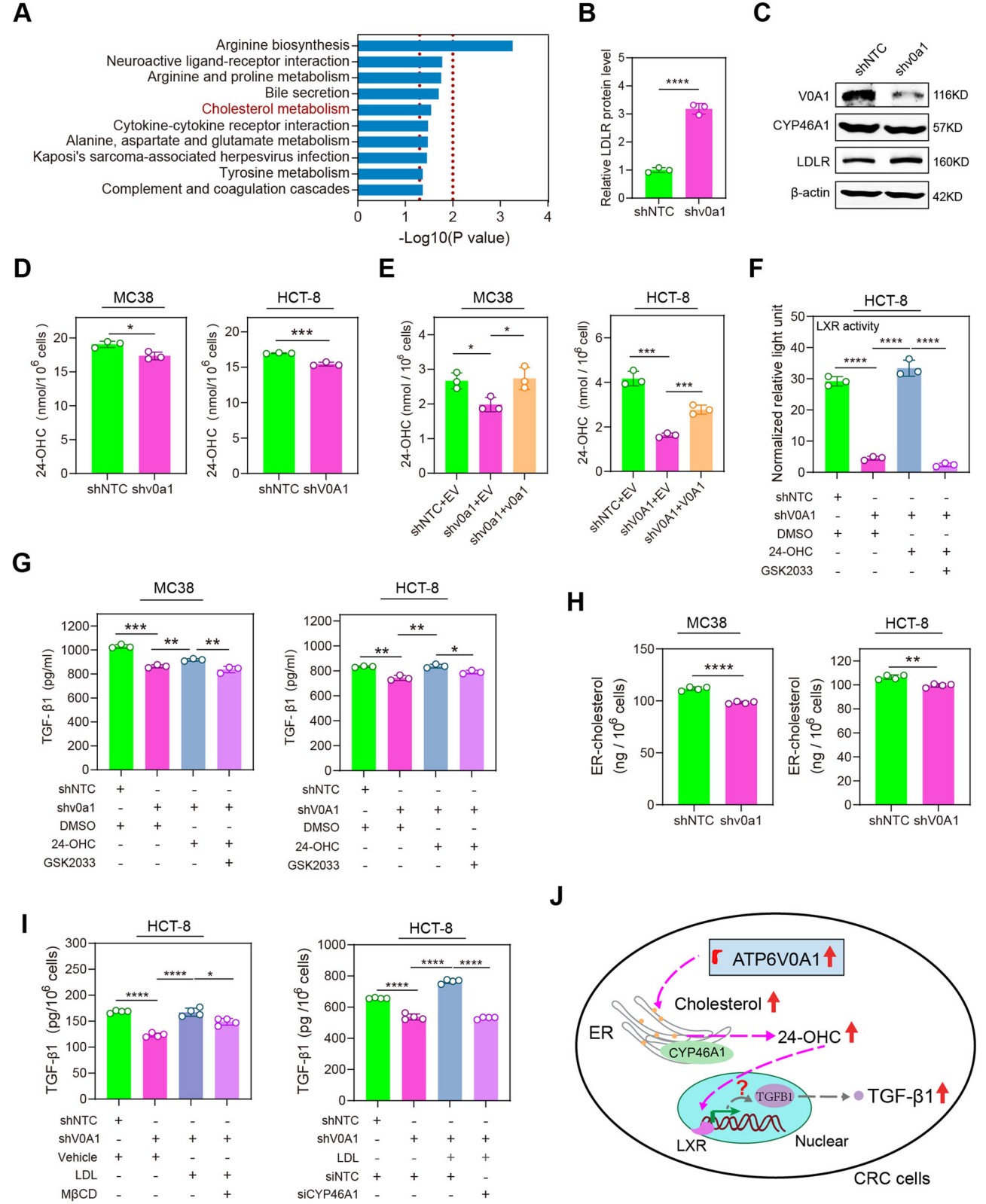

clinical CRC samples, we analyzed a scRNA-seq dataset based on 23 human CRC tumors and 10 normal samples from the GEO database[22]. Epithelial cell subpopulations were selected from the dataset using a strategy described in the methods and Supplementary Fig. 23A–D. *ATP6VOA1* and *TGFB1* were expressed in these epithelial cells and showed similar distributions in the dot plot generated from dimension reduction analysis (Supplementary Fig. 23E). Importantly, when the

epithelial cells were divided into two groups based on ATP6V0A1 expression, Pearson's Chi-squared analysis suggested that ATP6V0A1 may be the relevant factor influencing TGFB1 expression (Supplementary Table 1). Moreover, the Wilcoxon rank sum test analysis revealed significantly higher TGFB1 expression in the ATP6V0A1-positive group compared to the ATP6V0A1-negative group ($p = 4e-15$; Supplementary Fig. 23F). Next, we investigated the correlation

**Fig. 5 | Cholesterol accumulation in the ER is essential for the upregulation of TGF-β1 by ATP6V0A1 in CRC cells. A, B** *Atp6v0a1* knockdown-induced changes in protein levels in MC38 cells were analyzed using label-free protein quantitative mass spectrometry (MS); KEGG route enrichment analysis showed that changes in the cholesterol metabolism pathway were induced by *Atp6v0a1* knockdown (**A**). Relative levels of LDLR protein in MC38-shNTC and MC38-shv0a1 cells were determined by MS analysis (**B**; *n* = 3 independent experiments). **C** Western blotting was used to detect LDLR and CYP46A protein levels in MC38-shNTC and MC38-shv0a1 cells. The samples derive from the same experiment but different gels for ATP6V0A1, CYP46A1, β-actin, another for LDLR were processed in parallel. Data representative of 3 independent experiments. **D, E** Culture media from the indicated cells were analyzed for 24-OHC production by ELISA. *n* = 3 independent experiments. **F** HCT-8 (shNTC and shV0A1) cells with the indicated treatments were transfected with LXR luciferase reporter plasmids along with renilla luciferase control plasmids. The cells were subsequently assessed for LXR activities by calculating the ratio of luciferin light unit to renilla light unit (normalized relative light unit). *n* = 3 independent experiments. **G** MC38 (shNTC and shv0a1) cells and HCT-8 (shNTC and shV0A1) cells were treated with 24-OHC in the absence or presence of LXR inhibitor (GSK2033), and TGF-β1 levels in the supernatants were analyzed by ELISA. *n* = 3 independent experiments. **H** Following the isolation of ER from the indicated cells, lipids were extracted and analyzed for cholesterol levels using the Amplex™ Red cholesterol assay. *n* = 4 independent experiments. **I** HCT-8 (shNTC and shV0A1) cells were treated with LDL in the absence or presence of MβCD (**I**, left) or with LDL in the presence of siNTC or siCYP46A1 (**I**, right); TGF-β1 protein was detected in the supernatant by ELISA. *n* = 4 independent experiments. **J** Schematic diagram summarizing the results of Fig. 5A–I. For all experiments, data are shown as means ± s.e.m; *$p < 0.05$, **$p < 0.01$, ***$p < 0.001$, ****$p < 0.0001$. Statistical significance was determined using unpaired two-sided Student's *t*-test. Source data and exact *p*-value are provided as a Source Data file.

between memory CD8[+] T cell effectiveness and tumor cell-intrinsic *ATP6V0A1* or *TGFB1* in the clinical samples using a strategy described in the methods, Supplementary Fig. 23G, and Supplementary Table 2. Among the 17 tumor samples with three or more memory CD8[+] T cells, epithelial *ATP6V0A1* expression correlated positively with epithelial *TGFB1* expression (R = 0.47, $p = 0.025$), and both epithelial *ATP6V0A1* (R = −0.54, $p = 0.026$) and epithelial *TGFB1* (R = −0.49, $p = 0.046$) were inversely correlated with the cytotoxicity of memory CD8[+] T cells (evaluated as the mean value of *GZMA*, *PRF1*, and *KLRG1* expression) (Supplementary Fig. 23H–J). Notably, based on data from the TCGA database, high levels of effective memory CD8[+] T cells (indicated by the signature of CD8, EOMES, CD44, GZMA, PRF1, and KLRG1) were positively correlated with improved overall survival in CRC patients (Supplementary Fig. 23K).

To strengthen the clinical evidence supporting the aforementioned results, we performed an immunofluorescence (IF) assay to investigate the correlation between ATP6V0A1 and RABGEF1, TGF-β1, or memory CD8[+] T-cell effectiveness on the paraffin-embedded tumor sections obtained from 32 CRC patients. As shown in Fig. 8F, within the tumor tissues with ATP6V0A1 expression, both expression of RABGEF1 and TGF-β1 were relatively higher in the areas with high ATP6V0A1 rather than those with low ATP6V0A1 (Fig. 8F, upper). On the other hand, both RABGEF1 and TGF-β1 were rarely detected in the tumor tissues with minimal detection of ATP6V0A1 (Fig. 8F, lower). Moreover, significant colocalization of ATP6V0A1 and RABGEF1 or TGF-β1 was observed in the tumor tissues with high ATP6V0A1 (Fig. 8G). Importantly, among these 32 human CRC specimens, ATP6V0A1 expression was positively correlated with both RABGEF1 and TGF-β1 expression; the expression of RABGEF1 was also positively correlated with that of TGF-β1 (Fig. 8H). Additionally, a higher expression rate of IFN-γ in CD45RO[+] memory CD8[+] T cells was observed in the tumor tissues with low ATP6V0A1 rather than those with high ATP6V0A1 (Supplementary Fig. 24). ATP6V0A1 expression was inversely correlated with the expression rate of IFN-γ in CD45RO[+]CD8[+] T cells in the 32 CRC specimens (Fig. 8I). These findings provide clinical evidence supporting the inhibition of memory CD8[+] T cell efficacy by ATP6V0A1 through RABGEF1/TGF-β1 signaling in CRC.

### Targeting ATP6V0A1 restores memory CD8[+] T-cell-mediated anti-tumor immunity in CRC

Our results suggested that tumor-intrinsic ATP6V0A1 induces immunosuppressive signaling and immune evasion in CRC by decreasing the effectiveness of memory CD8[+] T cells. Therefore, we wanted to explore the therapeutic potential of ATP6V0A1 inhibition in CRC animal models. First, we used docking-based analysis of protein-inhibitor interactions to screen a group of FDA-approved small molecular inhibitors for potential inhibitors of ATP6V0A1. We found that daclatasvir (Dac), a clinical used drug to treat chronic genotype I and III hepatitis C

virus (HCV) infection[23], was able to bind to ATP6V0A1 protein (Fig. 9A and Supplementary Fig. 25A, B). We confirmed the interaction using the cellular thermal shift assay, a method of evaluating drug binding to target protein in cells and tissues by detection of changes in the thermal stability of a protein following ligand finding[24]. These assays showed that Dac could induce thermal stabilization of ATP6V0A1 protein but not a control protein (β-actin) in MC38 (Supplementary Fig. 25C, D) and HCT-8 cells (Supplementary Fig. 25E, F), confirming the binding of Dac to ATP6V0A1 in these cells. Interestingly, Dac did not significantly bind to other subtypes of the V0A subunit (Supplementary Fig. 25G), suggesting the specificity of Dac binding to ATP6V0A1. Next, we investigated the ability of Dac to disrupt ATP6V0A1-mediated cell functions in vivo (Fig. 9B). Interestingly, while Dac treatment did not alter the growth of MC38 tumors in NOD/SCID mice (Fig. 9C–E), in C57BL/6 J mice it significantly suppressed the growth of MC38 tumors (Fig. 9C–E), reduced TGF-β1 expression in tumor tissue (Fig. 9G), and activated TME-derived CD44[+]CD8[+] T cells (Fig. 9H). Importantly, Dac treatment also significantly suppressed HCT-8 tumor growth in human immune-reconstituted huPBMC-NCG mice but not in immunodeficient NCG mice (Fig. 9I–K). Moreover, Dac treatment reduced TGF-β1 expression in tumor tissues (Fig. 9M) and enhanced the activation of CD45RO[+]CD8[+] T cells within HCT8 tumors (Fig. 9N). Notably, Dac treatment induced neither body weight loss (Fig. 9F, L) nor organ toxicity (Supplementary Fig. 26) in the mice, indicating the safety of this treatment.

We further investigated the targeting specificity of Dac in treating CRC. The in vitro treatment with Dac significantly reduced RABGEF1 expression, 24-OHC production, and TGF-β1 levels in control CRC cells but not in ATP6V0A1-deficient CRC cells (Fig. 10A–D). Finally, Dac treatment suppressed the tumor growth (Fig. 10E–G) and restored the activation of CD44[+]CD8[+] T cells (Fig. 10H, I) in control MC38 tumors but not in ATP6V0A1-suppressing MC38 tumors. Collectively, these findings demonstrated that Dac enables the effective suppression of CRC by targeting ATP6V0A1.

## Discussion

As a risk factor for CRC development, obesity suppresses anti-tumor immunity in CRC. TME-derived fatty acids and cholesterol can directly regulate the anti-tumor functions of immune cells via metabolism-dependent or independent pathways[2,25]. On the other hand, the utilization and metabolism of TME-derived fatty acids in CRC cells play essential roles in immune evasion by modulating fatty acid levels in the TME[1]. Despite the above advances, the mechanisms utilized by CRC cells to suppress anti-tumor immunity via TME-derived cholesterol remain unknown. In this study, we revealed tumor cell-intrinsic ATP6V0A1 as a novel and crucial factor promoting the absorption of exogenous cholesterol via a RABGEF1-dependent endosome mature pathway in CRC cells, leading to higher ER-derived cholesterol level and 24-OHC production and upregulated TGF-β1expression.

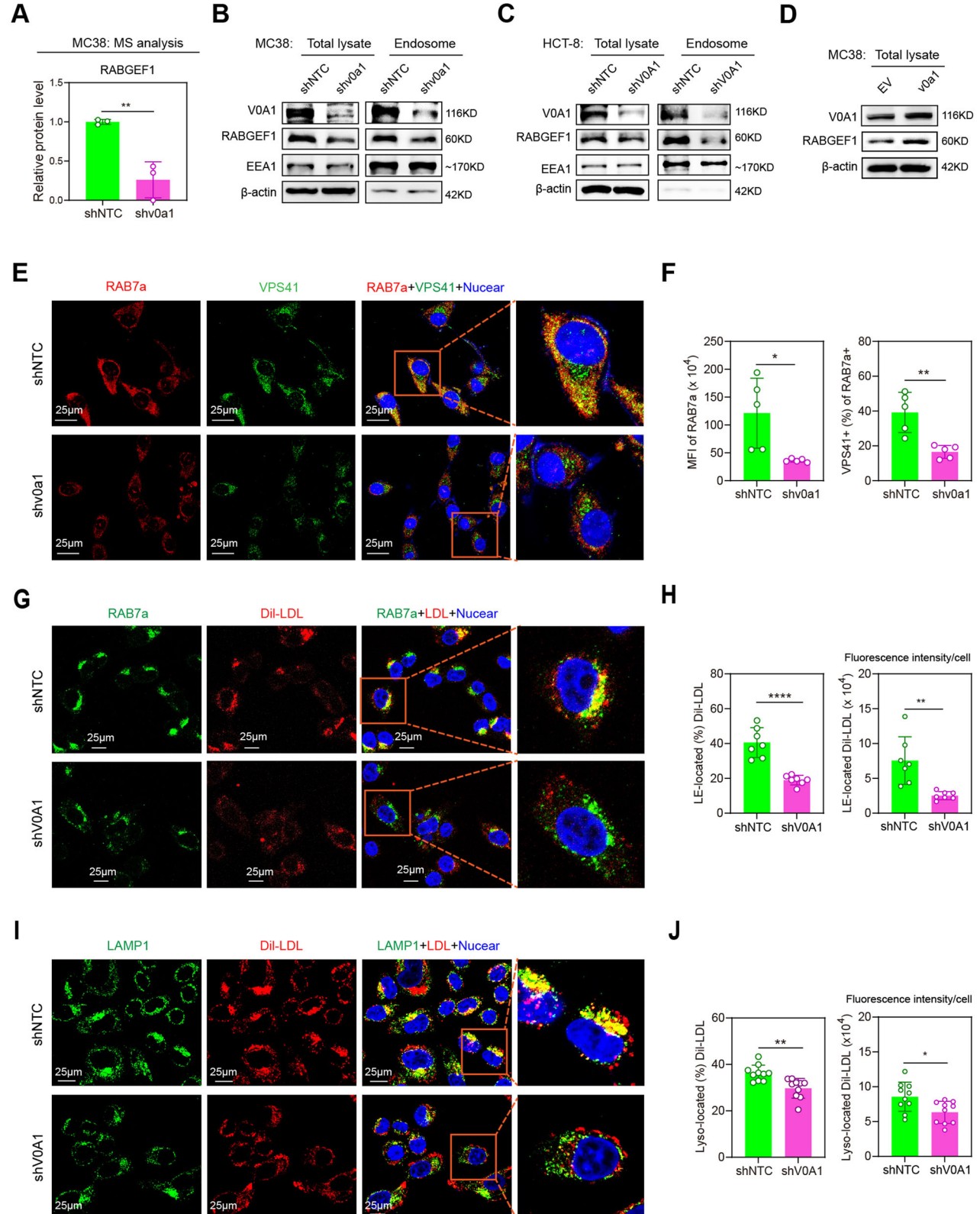

Moreover, upregulation of TGF-β1, a critical cytokine involved in anti-tumor immunity, suppressed the activation of memory CD8$^+$ T cells and promoted immune evasion in CRC mouse models. The links between cholesterol metabolic reprogramming in CRC cells and the tumor immunosuppressive microenvironment remains unclear. In this study, we highlight TGF-β1 as a new potential link between tumor cell-derived cholesterol metabolic reprogramming and the suppressive

functions of immune cells. These findings open up new avenues for studying the mechanisms by which CRC cells utilize TME-derived cholesterol to suppress anti-tumor immunity. Meanwhile, these data demonstrate new tracks for V-ATPase subunits regulating cholesterol metabolism and anti-tumor immunity.

In this study, we focused on the role of ATP6V0A1-induced TGF-β1 in the inactivation of memory CD8$^+$ T cells, as we observed the most

**Fig. 6 | ATP6V0A1 enhances the absorption of exogenous cholesterol by promoting RABGEF1-dependent endosome maturation. A** *Atp6v0a1* knockdown-induced changes in the levels of RABGEF1 protein were analyzed in MC38 cells by quantitative mass spectrometry (MS). $n = 3$ independent experiments. **B–D** Western blotting was used to detect levels of RABGEF1 protein in whole cells (total lysate) and endosomes. In (**B, C**), the samples derive from the same experiment but different gels for RABGEF1, EEA1, β-actin, another for ATP6V0A1 were processed in parallel. **E, F** RAB7a (red) and VPS41 (green) proteins in MC38-shNTC and MC38-shv0a1 cells were detected using confocal fluorescence microscopy (**E**); quantitative analysis of vesicle-derived RAB7a levels (**F**, left; $n = 5$ fields per group) and the percentages of RAB7a+ vesicles that were VPS41+ (**F**, right; $n = 5$ fields per group) was carried out using Image J. **G–J** HCT-8-shNTC and HCT-8-shV0A1 cells were treated with 50 μg/ml of human Dil-LDL for 6 h, and confocal fluorescence microscopy was used to analyze the colocalization of Dil-LDL with RAB7a+ late endosomes (LE; **G, H**) or LAMP1+ lysosomes (Lyso; **I, J**). Representative images are shown in (**G, I**). Quantitative analyses of Dil-LDL localization in endosomes (**H**; $n = 7$ fields per group) and lysosomes (**J**; $n = 10$ fields per group) were carried out by measuring the ratio (**H, J**, left) or the fluorescence intensity (**H, J**, right) of Dil-LDL located in these vesicles with Image J. For all experiments, data are shown as means ± s.e.m; $*p < 0.05$, $**p < 0.01$, $****p < 0.0001$. Statistical significance was determined using unpaired two-sided Student's $t$-test. MFI mean fluorescence intensity = fluorescence intensity/cell. Three independent experiments were performed for Fig. 6B–J. Source data and exact $p$-value are provided as a Source Data file.

significant changes in activation and SMAD3 signaling within this specific population of immune cells. Consistently, TGF-βRII, an essential receptor for TGF-β1, exhibited pronounced expression in memory-like T-2 cells as opposed to other subsets of CD8⁺ T cells within MC38 tumors. Our data suggests that TGF-βRII may act as a receptor checkpoint in memory CD8⁺ T cells. However, TGF-β1 would also contribute to regulating Treg differentiation and activating suppressive myeloid cells[26,27]. Moreover, TGF-β1 may also play essential roles in CRC metastasis through an autocrine pathway. In future investigations, we plan to explore the influence of ATP6V0A1-induced TGF-β1 on tumor cells and other immune cells.

The enhancement of cholesterol absorption is critical for CRC cells utilizing exogenous cholesterol to trigger immunosuppressive signaling. Endo-lysosomal systems play vital roles in cholesterol absorption[28], but the mechanism by which CRC cells support the endo-lysosomal pathway to enhance cholesterol absorption remains unclear. Our study revealed ATP6V0A1 as a novel regulator that promotes endosome maturation in CRC cells. Previous studies showed that V-ATPase subunits are involved in the vesicle traffic[29], but the underlying mechanism is unclear. Our study revealed that the RABGEF1-dependent pathway is critical for ATP6V0A1-induced endosome maturation, indicating a new function of V-ATPase subunits regulating vesicle traffic. However, further investigation is required to fully understand how ATP6V0A1 regulates RABGEF1 expression. Moreover, it remains to be elucidated what promotes the expression level of ATP6V0A1 in CRC cells. Since ATP6V0A1 is elevated in the CRC with a high level of lipid metabolism compared to those with a lower lipid metabolism, exogenous lipids may be a factor contributing to the increased expression of ATP6V0A1 in CRC cells. The expression of ATP6V0A1 could also be elevated by upstream molecular signaling pathways governed by other critical genes, which merits further investigation in future studies.

Our findings reveal a novel function of ATP6V0A1 in facilitating tumor progression by inhibiting anti-tumor immune response. Furthermore, we provide the first evidence that the V-ATPase-driven immune evasion is governed by the cholesterol metabolism pathway in colorectal cancer (CRC) cells. The functions of ATP6V0A1 in regulating RABGEF1-dependent cholesterol absorption and TGF-β1 expression may not necessarily rely on the overall levels of the V-ATPase complex or its V0 subcomplex, since the changes in ATP6V0A1 expression did not alter the total levels of the V-ATPase complex and its V0 subcomplex. Moreover, ATP6V0A1 had distinct functions in regulating RABGEF1/TGF-β1 pathway compared to the other V0A subtypes. V-ATPases play vital roles in regulating cellular and vesicular pH, and the V-ATPase complexes with different V0A subtypes would exert different functions in regulating pH[30]. ATP6V0A1 suppression did not alter the cellular pH, but enhanced the pH in RAB7a⁺ endosomes. We cannot exclude the possibility that the pH regulation of RAB7a⁺ vesicles may be involved in ATP6V0A1-mediated cholesterol absorption and TGF-β1 expression in the present study. In fact, ATP6V0A1 suppression can reduce 24-OHC production and TGF-β1 expression in the HCT-8 cells treated with control vehicles but not in those treated with Baf-A1, an

inhibitor of V-ATPase activity, indicating that the acidification function of V-ATPase is involved in the pathway of ATP6V0A1 regulating 24-OHC and TGF-β1. However, it is of note that the endosome acidification would be the result of endosome maturation[31]. While promoting endosome maturation, the RABGEF1 pathway can recruit VPS41 and RILP to endosomes and thus strengthen the linkage of V0-subcomplex and V1-subcomplex via the interaction of RILP and ATP6V1G1[20,31]. Therefore, the process of ATP6V0A1 regulating endosome maturation via a RABGEF1-dependent pathway may be not dependent on the acidification activity of V-ATPases. Moreover, ATP6V0A1 may regulate the acidification of endosomes via the RABGEF1/VPS41/RILP pathway rather than the changes in the level of the V-ATPase complex or its V0 subcomplex. We plan to further investigate the crosstalk between the RABGEF1 pathway, endosome maturation, and vesicle pH in the context of ATP6V0A1-induced cholesterol absorption.

In the present study, we offer a novel immunotherapeutic strategy for CRC through the targeted inhibition of ATP6V0A1. A screen of small molecular inhibitors revealed that Dac, a drug approved by the FDA to treat chronic HCV infection[23,32], was able to bind ATP6V0A1 protein and suppress the growth of murine and human CRCs in an immune-dependent manner. To the best of our knowledge, this is the first report of the use of Dac to treat cancer. As Dac is an FDA-approved clinical drug, it would be safe for patients if repurposed for cancer treatment. Indeed, we found no significant toxic side effects when using Dac to treat CRCs in the animal models in this study. Importantly, the efficacy of Dac in attenuating CRC is attributed to its selective targeting of ATP6V0A1. Depletion of ATP6V0A1 diminishes the ability of Dac to reduce the levels of 24-OHC and TGF-β1 in CRC cells, thereby impeding its capacity to suppress CRC growth and restore anti-tumor activity in in vivo models. Notably, in CRC, the therapeutic effect of anti-TGF-β1 may not replace that of targeting ATP6V0A1 since anti-TGF-β1 treatment only partially weakened the promoting effect of *Atp6v0a1* overexpression on MC38 tumor growth. Other factors would be involved in ATP6V0A1-mediated immune escape, which needs to be clarified in future studies. Moreover, targeting ATP6V0A1 may also produce distinct effects from those induced by targeting V-ATPase complexes. The relationship with the immune microenvironment in CRC varied between different V-ATPase subunits. Additionally, the suppression of ATP6V0A1 would not affect the overall level of the V-ATPase complex and exhibits distinctive roles in the regulation of immune evasion compared to other V0A subtypes. Therefore, ATP6V0A1-targeted therapy may be uniquely significant for developing effective anti-CRC immunotherapy strategies. According to the single cell type analysis by The Human Protein Atlas (www.proteinatlas.org), within colon tissue, ATP6V0A1 prefers to be expressed in epithelial cells rather than in other cell types, including immune cells. Moreover, the IHC analysis of CRC samples also revealed an elevated expression level of ATP6V0A1 in the tumor niches compared to the stromal areas. The data indicate that ATP6V0A inhibitors may selectively target tumor cells within CRC tissues. However, further investigation is needed to determine the targeting specificity of Dac in CRC treatment.

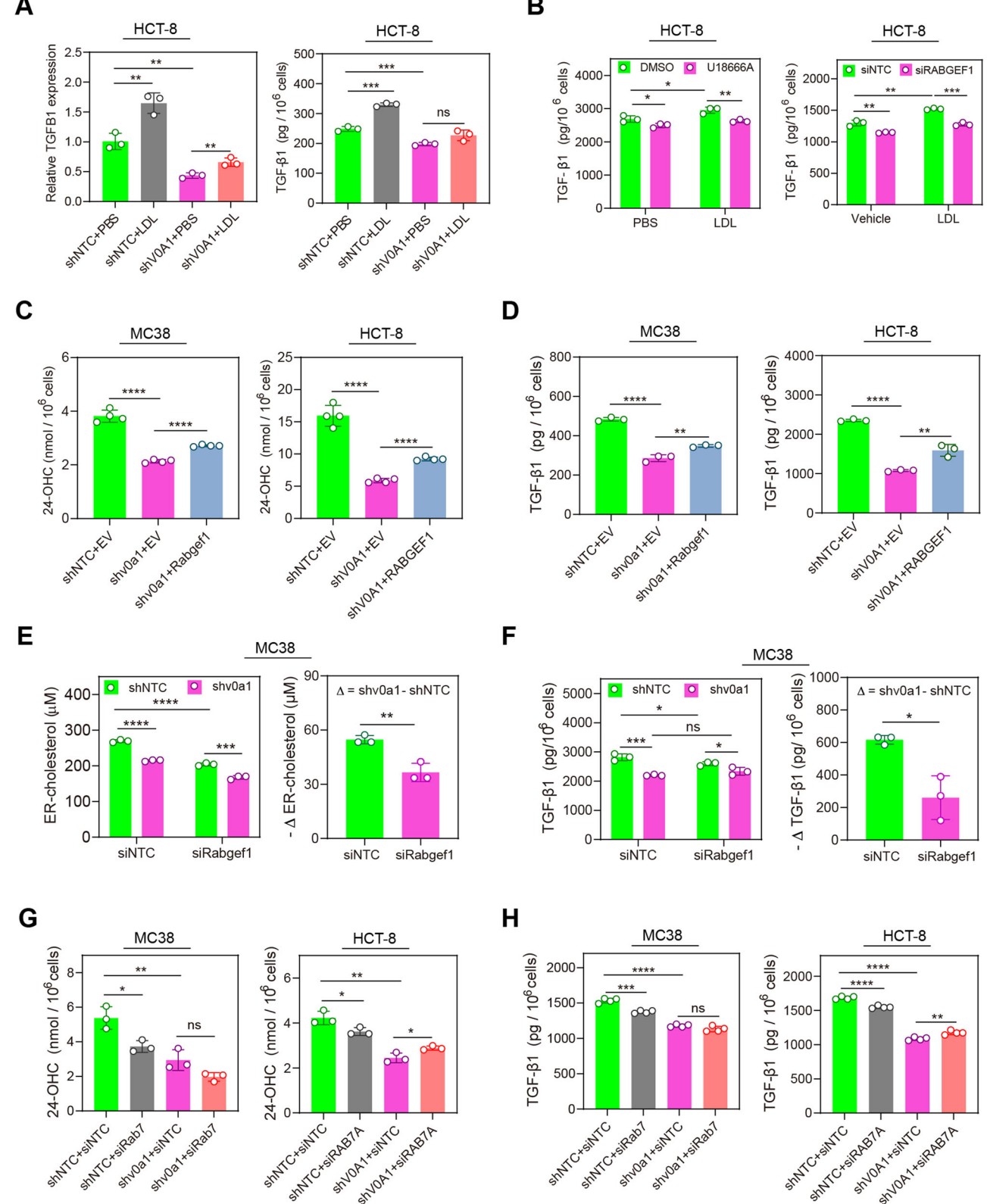

The up-regulation of ATP6V0A1 and its influence on immune evasion through the RABGEF1/24-OHC/TGF-β1 pathway is independent of the type of mismatch repair (MMR) or microsatellite instability (MSI) in CRCs. ATP6V0A1 is overexpressed in both CRC patients with deficient mismatch repair (dMMR) and with proficient mismatch repair (pMMR). Moreover, ATP6V0A1 drives ER-cholesterol-induced TGF-β1 enhancement through RABGEF1-dependent endosome maturation in both CRC cells with dMMR (MC38) and pMMR (CT26). Importantly, inhibition of ATP6V0A1 by expression interference or inhibitor treatment was shown to suppress tumor growth and restore anti-tumor immunity in both dMMR and pMMR murine CRCs. Immune checkpoint blockade demonstrates a delightful therapeutic effect in dMMR CRCs but not in pMMR CRCs. As immunotherapy for pMMR CRCs encounters

**Fig. 7 | Cholesterol absorption mediates the enhancement of ER-cholesterol and TGF-β1 induced by ATP6V0A1 in CRC cells. A** HCT-8-shNTC and HCT-8-shV0A1cells were treated with 20 μg/ml of human Dil-LDL in the culture medium containing 5% lipid-depleted fetal bovine serum, and the levels of TGF-β1 were evaluated separately in the cells and supernatant using Q-PCR and ELISA. $n = 3$ independent experiments. **B** HCT-8 cells were treated as the indication, and the level of TGF-β1 in the supernatant was detected by ELISA. $n = 3$ independent experiments. **C, D** Exogenous Rabgef1 or RABGEF1 was overexpressed in *ATP6v0a1*-suppressing MC38 cells and *ATP6V0A1*-suppressing HCT-8 cells; 24-OHC (**C**; $n = 4$ independent experiments) and TGF-β1 (**D**; $n = 3$ independent experiments) in the supernatant was measured by ELISA. **E, F** Following the isolation of ER from the control or *Atp6v0a1*-suppressing MC38 cells transfected with control or *Rabgef1*-targeted siRNAs, lipids were extracted and analyzed for cholesterol levels using the Amplex™ Red cholesterol assay (**E**). Moreover, TGF-β1 levels in the supernatants were analyzed by ELISA (**F**). $n = 3$ independent experiments. **G, H** ATP6V0A1-suppressing MC38/HCT-8 cells were transfected with control or *Rabgef1/RABGEF1*-targeted siRNAs, and 24-OHC (**G**; $n = 3$ independent experiments) or TGF-β1 (**H**; $n = 4$ independent experiments) in the supernatant was measured by ELISA. For all experiments, data are shown as means ± s.e.m; *$p < 0.05$, **$p < 0.01$, ***$p < 0.001$, ****$p < 0.0001$. Statistical significance was determined using unpaired two-sided Student's *t*-test. Source data and exact *p*-value are provided as a Source Data file.

challenges, our study may offer promising strategies for enhancing immunotherapy in this context.

## Methods

### Ethical statement
The animal study complies with the Institutional Animal Care and Use Committee (IACUC) guidelines at Shenzhen University Medical School. All human subjects provided informed consent, and Institutional Review Board approval was obtained for this study from Shenzhen University and Shenzhen People's Hospital.

### Cell lines, mice, and human specimens
MC38 and CT26 murine colon adenocarcinoma cell lines and HCT-8 human colon adenocarcinoma cell line were obtained from the American Type Culture Collection (ATCC, Manassas, VA, USA). All cell lines were negative for mycoplasma and were maintained in Dulbecco's modified Eagle medium (DMEM; Invitrogen, Carlsbad, CA, USA) containing 10% fetal bovine serum (Invitrogen), 100 U/ml penicillin, 100 mg/ml streptomycin (Invitrogen), 2 mM L-glutamine (Invitrogen), and 1 mM pyruvate acid (Invitrogen).

Four- to six-week-old female C57BL/6 J mice, BALB/c mice, and NOD.CB17-Prkdcscid/NcrCrl (NOD/SCID) mice were purchased from Charles River Laboratories (Beijing, China); C57BL/6 Rag2$^{-/-}$Il2rg$^{-/-}$ mice and BRG mice were purchased from Shanghai Model Organisms Center, Inc. (Shanghai, China); Four- to six-week-old female NOD/ShiLtJGpt-Prkdcem26Cd52Il2rgem26Cd22/Gpt (NCG) mice were purchased from Gempharmatech Co., Ltd (Nanjing, China). The mice were maintained under specific pathogen-free conditions and cohoused with five mice per incubator with a 12-h light/dark cycle. The mice were fed with sterilized food and water ad libitum. All animal procedures followed guidelines approved by the Institutional Animal Care and Use Committee at Shenzhen University Medical School with approval No. IACUC-202400061.

Human peripheral blood mononuclear cells (PBMCs) used to reconstitute human lymphocytes in NCG mice were isolated from 12 healthy volunteer donors (6 male and 6 female), and 32 paraffin-embedded human CRC tissue samples were obtained from Shenzhen People's Hospital. All human subjects provided informed consent, and Institutional Review Board approval was obtained for this study from Shenzhen University and Shenzhen People's Hospital. In addition, colon adenocarcinoma (COAD) tissue microarrays (TMAs) for three independent patient cohorts (HColA160Bc01, HColA160CS01, and HColA180Su12) were purchased from Shanghai OUTDO BIOTECH Co., Ltd. (Shanghai, China).

### In vivo experiments
Control or *Atp6v0a1* knockdown murine CRC cells (MC38, $1 \times 10^6$ cells/mouse; CT26, $5 \times 10^5$ cells/mouse) were subcutaneously injected into the right flanks of homologous immunocompetent mice (MC38, C57BL/6 J; CT26, BALB/c) and immunodeficient Rag2$^{-/-}$Il2rg$^{-/-}$ or NOD/SCID mice. Similarly, HCT-8-shNTC or HCT-8-shV0A1 cells ($3 \times 10^6$ cells/mouse) were subcutaneously injected into the right flanks of control NCG Mice and huPBMC-NCG mice (in which the immune systems had been successfully reconstituted using human PBMCs). To establish the orthotopic CRC model, subcutaneous MC38-shNTC or MC38-shv0a1 tumors were cut into pieces measuring $2 \times 2 \times 2$ mm, and implanted into the cecal mucosa as described in previous studies[33]. Tumors were monitored using calipers, and tumor volumes were calculated using the formula volume = length × width × height / 2.

In tumor treatment experiments, MC38 or MC38-shNTC/shv0a1 tumor-bearing C57BL/6 J mice or HCT-8-tumor bearing huPBMC NCG mice were randomly assigned to treatment groups after the development of tumors of around 5 mm in diameter. They were then intraperitoneally injected with 5 mg/kg or 7.5 mg/kg of Daclatasvir (Dac) on days 7 and 11.

The animal experiments were terminated before tumor weight reached 10% of the animal weight.

### Gene knockdown and overexpression
Murine MC38 and CT26 and human HCT-8 cells were purchased from ATCC and cultured in DMEM containing 10% FBS, 1% P/S (penicillin-streptomycin), 1% L-glutamine and 1% sodium pyruvate. Lentivirus was generated with a packaging mix (Sigma) and pLKO.1-Puro-shAtp6v0a1/shATP6V0A1 plasmid, pLVX-Puro-Atp6v0a1 plasmid, or the corresponding control plasmid, according to the manufacturer's protocol, and used for tumor cell transfection. The lentivirus-transfected CRC cells were treated with puromycin (Sigma) to select Atp6v0a1/ATP6V0A1 knockdown or Atp6v0a1-expressing stable clones. For Rabgef1/RABGEF1 knockdown, siRNA targeting Rabgef1/RABGEF1 was transfected into MC38/HCT-8 cells with Lipofectamine 3000 (Invitrogen). The sequences and sources of shRNA and siRNA are described in Supplementary Table 3.

### Flow cytometry analysis
For the in vitro T-cell stimulation analysis, CD8$^+$ T cells were isolated from C57BL/6 J mice-bearing MC38 wild-type tumors using a negative mouse CD8$^+$ T-cell isolation kit (STEMCELL). These isolated CD8$^+$ T cells were cultured in RPMI 1640 medium containing 10% FBS with IL-2 (10 ng/mL) and treated with the following: SMAD3 inhibitor (E)-SIS3, MC38 cell culture medium plus control IgG or anti-TGF-β1, conditioned medium (CM) from MC38-shNTC cells or MC38-shv0a1 cells with/without incubation with recombinant mouse TGF-β1 protein. Brefeldin A (eBioscience) was added to the culture in the last 4 h, and the CD8 + T cells were collected and stained with Fixable Viability Stain 510 (FVS510; BD Horizon™), fluorescein isothiocyanate (FITC)-conjugated anti-CD8a (53-6.7; eBioscience) and PerCP-Cyanine5.5-conjugated anti-CD44 (IM7; eBioscience) as cell-surface staining protocols followed by fixation, permeabilization, and intracellular staining with phycoerythrin (PE)-conjugated anti-perforin (eBioOMAK-D; eBioscience) and allophycocyanin (APC)-conjugated anti-GzmB (QA16A02; BioLegend), or with PE-conjugated anti- IFN-γ (XMG1.2; eBioscience). In another experiment, TILs were isolated from C57BL/6 J mice-bearing MC38 shNTC or shv0a1 tumors and directed to the antigen staining. TILs were stained with Fixable Viability Stain 510 (FVS510; BD Horizon™), BV421-conjugated anti-CD45 (30-F11; BD Horizon™), PerCP-Cyanine5.5-conjugated anti-CD8a (53-6.7; BD

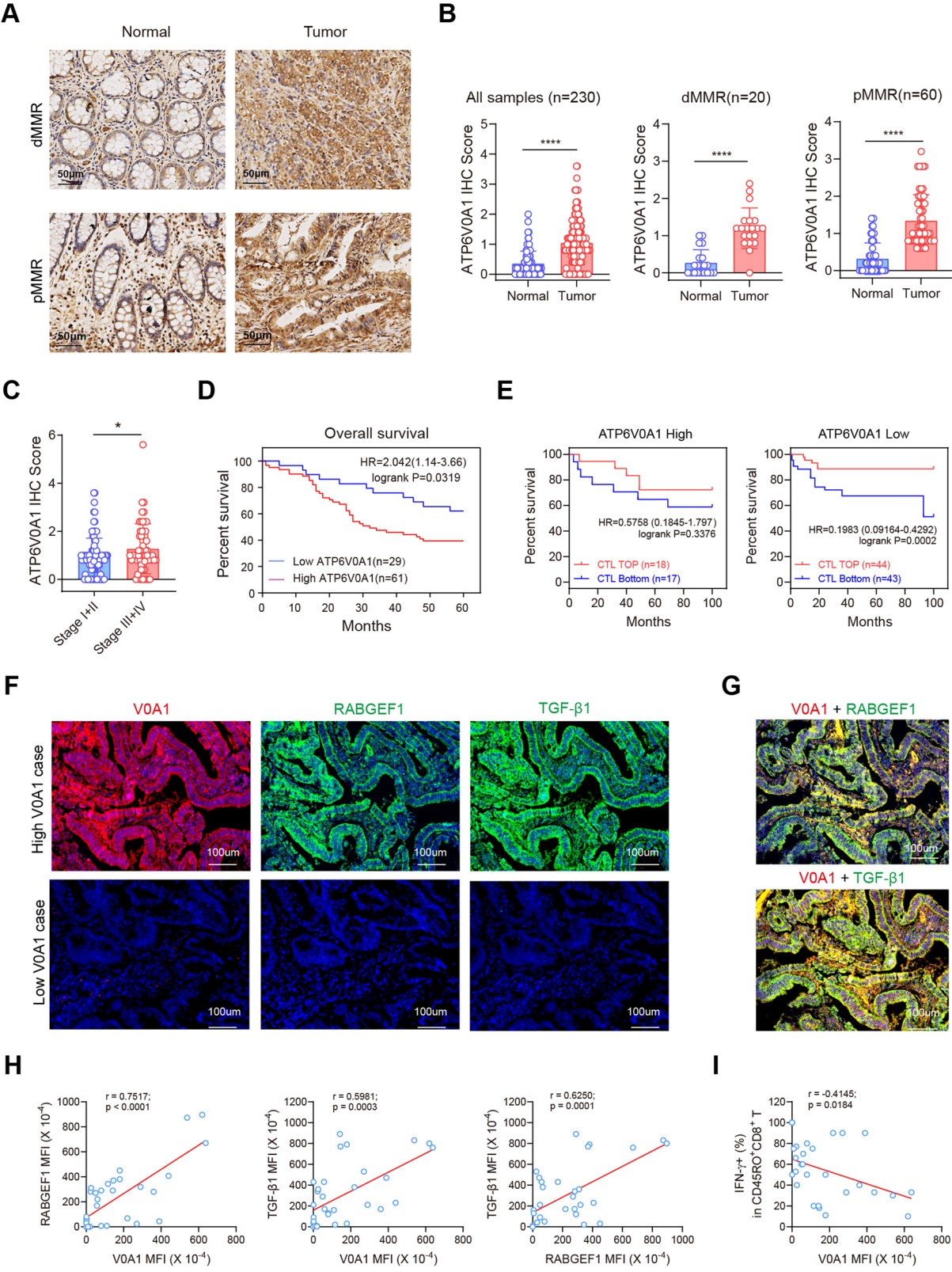

Horizon™), APC-conjugated anti-CD44 (IM7; BD Horizon™) as cell-surface staining protocols followed by fixation, permeabilization, and intracellular staining with phycoerythrin (PE)-conjugated anti-perforin (eBioOMAK-D; eBioscience), PE-conjugated anti-GzmB (QA16A02; BioLegend), or with PE-conjugated anti- IFN-γ (XMG1.2; eBioscience). For the detection of TILs from huPBMC-NCG mice-bearing HCT-8 shNTC or shV0A1 tumors, the following dyes or antibodies were used:

Fixable Viability Stain 510 (FVS510; BD Horizon™), BV421-conjugated anti-CD45 (HI30; BD Horizon™), FITC-conjugated anti-CD8a (RPA-T8; eBioscience), APC-conjugated anti-CD45RO (UCHL1; BioLegend), PerCP-Cyanine5.5-conjugated anti-IFN-γ (4 S.B3; BioLegend), PE-conjugated anti-GzmB (QA16A02; BioLegend). The stained cells were analyzed using the BD FACSAria system and FlowJo software (BD Biosciences).

**Fig. 8 | High expression of ATP6V0A1 is positively correlated with RABGEF1 and TGF-β1 and inversely correlated with anti-tumor immunity in human CRC.** **A**, **B** Representative images of ATP6V0A1 in tumor tissues and the corresponding peritumoral tissues were shown for CRC patients with dMMR or pMMR (**A**). Paired *t*-test analysis of ATP6V0A1 levels in all 230 tumor tissues, 20 tumor tissues with dMMR, or 60 tumor tissues with pMMR and their corresponding peritumoral tissues (**B**). Tumor tissues from 153 patients with stage information (**C**) and 90 patients with survival information (**D**) were analyzed to compare ATP6V0A1 expression in early and late stages and determine the correlation between ATP6V0A1 expression and patient overall survival, respectively. **E** The prognostic value of ATP6V0A1 in predicting T-cell dysfunction and patient survival in CRC was evaluated by the Tumor Immune Dysfunction and Exclusion (TIDE) analysis based on the GSE38832 database. **F**–**I** Paraffin-embedded tumor sections from 32 CRC patients were stained with antibodies against ATP6V0A1, RABGEF1, TGF-β1, and IFN-γ⁺CD45RO⁺CD8⁺ T cells, and the correlations between ATP6V0A1 and RABGEF1, TGF-β1, or IFN-γ⁺CD45RO⁺CD8⁺ T cells were analyzed. Representative immunofluorescence (IF) images for ATP6V0A1, RABGEF1, and TGF-β1 expression between high- and low-ATP6V0A1 cases were shown (**F**). Representative images showing the co-localization of ATP6V0A1 with RABGEF1 or TGF-β1 in high ATP6V0A1 cases (**G**). The correlation between ATP6V0A1 and RABGEF1 expression, ATP6V0A1 and TGF-β1 expression, or RABGEF1 and TGF-β1 expression was analyzed among 32 CRC specimens (**H**). The correlation between ATP6V0A1 expression and CD45RO⁺CD8⁺ T-cell effectiveness (IFN-γ expression rate) was analyzed among 32 CRC specimens (**I**). For all experiments, data are shown as mean ± s.e.m; *$p < 0.05$, ****$p < 0.0001$. Statistical significance was determined using paired two-sided Student's *t*-test in (**B**), unpaired two-sided Student's *t*-test in (**C**), and two-sided Correlation test (**H**, **I**). MFI mean fluorescence intensity. Source data and exact *p*-value are provided as a Source Data file.

For the cytotoxic T lymphocyte (CTL) assay, TILs were separately isolated from C57BL/6 J mice-bearing *Atp6v0a1* knockdown or over-expressing MC38 tumors and those bearing control tumors. These TILs (effector cells) were cocultured with CFSE-labeled MC38 cells (target cells) for 8 h at 37 °C to test their cytolytic activity (effector cells: target cells = 100: 1). Following 5 min of staining with one ug/ml of propidium iodide (PI; Invitrogen), the cell mixture was analyzed for the proportion of tumor cell death using the CytoFLEX system and Kaluza software (Beckman Coulter).

### scRNA-seq analysis

Following tissue digestion, single cells from control MC38 tumors (shNTC, *n* = 5; mixed in equal proportions) and Atp6v0a1 knockdown MC38 tumors (shv0a1, *n* = 5; mixed in equal proportions) were sequenced by 10x Genomics Chromium platform (Novogene Bioinformatics Technology Co., Ltd.). Raw fastq files were processed with CellRanger V3, and the raw sequencing reads were aligned to the murine genome (GRCm38/mm10, ENSEMBL). After quality control, total cells from the above two groups were retained and clustered into five major groups using the t-distributed stochastic neighbor embedding (TSNE) method (Supplementary Fig. 8A). The major groups were annotated as MC38 cells, cancer-associated fibroblasts (CAFs), endothelial cells, myeloid cells, and T and natural killer (NK) cells using the markers *Col5a1*, *Fabp4*, *Cd3e*, *Shisal2b*, *Cd68* and *Ncr1*, respectively (Supplementary Fig. 8B).

T and NK cells were further subclustered into seven subgroups (Supplementary Fig. 8C) and annotated as naïve T, effector T, exhausted T, Treg, memory-like T-1, memory-like T-2, and NK subgroup using the markers shown (Supplementary Fig. 8C, D). In detail, Cluster 1 expressed naïve T-cell markers, including *Lef1*, *Tcf7*, *Ccr7*, and *Sell*, and was defined as naïve T cells. Clusters 2, 3, and 7 expressed effector markers, including *Prf1*, *Gzmb*, *Plac8*, *Gzmk*, and *Nkg7*. Cluster 7 highly expressed *Ncr1* with expression of neither *Cd4* nor *Cd8a* and was hence named NK cells. Cluster 2 and Cluster 3 did not express *Ncr1* and were distinguished from each other by the expression of inhibitory markers, including *Lag3*, *Pdcd1*, *Tigit*, and *Ccl4*. We defined Cluster 2, with low inhibitory markers, as effector T cells and named Cluster 3, which highly expressed inhibitory markers as exhausted T cells. Cluster 4 expressed Treg markers, including *Foxp3*, *Il2r*, and *Ikzf2*, and was defined as Treg cells. Clusters 5 and 6 highly expressed *Ptprc*, *Lck*, and *Bcl2*, which can be used as memory T-cell markers in scRNA-seq analysis of murine samples[34]. Moreover, among the T cell subpopulations which preferred to express *Cd8a* but *Cd4*, Cluster 5 and Cluster 6 have the highest levels of *Il7r*, a classic marker of memory CD8⁺ T cells[35]. IL7R plays essential roles in maintaining functions of memory CD8⁺ T cells[35–37]. Therefore, Clusters 5 and 6 were separately named memory-like T-1 and T-2 cells.

Myeloid cells were clustered into four subgroups, and annotated as macrophages, granulocytes, monocytes, and dendritic cells (DCs) using the markers *Adgre1*, *Cxcl2*, *Ly6c2*, and *CD86*, respectively (Supplementary Fig. 8E, F).

For the study in clinical CRC samples, we downloaded a scRNA-seq dataset (GSE132465) based on 23 human CRC tumors and ten normal samples from the GEO database. The 63,689 cells in this dataset were clustered into epithelial, stromal, T, B, myeloid, and mast cell subpopulations (Supplementary Fig. 23A, B). As in the previous study[22], these cell subpopulations were annotated using the markers shown in Supplementary Fig. 23B. Epithelial cells were then further divided into six clusters (Supplementary Fig. 23C). Clusters 1 and 5 were found to express abundant fibroblast markers, including *COL1A2* and *TAGLN*, with (cluster 1) or without (cluster 5) epithelial markers (Supplementary Fig. 23D). These two clusters may not be pure epithelial cells. We, therefore, selected the other epithelial cell subpopulations, which expressed epithelial markers but not obvious fibroblast markers, for further investigation. CD8⁺ T cells were selected from the T-cell subpopulations annotated in the downloaded dataset[22] and evaluated for their expression of T memory markers; in line with previous studies[38,39], *EOMES*⁺*CD44*⁺*CD8A*⁺ T cells were used as memory CD8⁺ T cells in this study (Supplementary Fig. 23G). Memory CD8⁺ T cell counts were analyzed individually in the 23 CRC tumors (Supplementary Table 2). The tumor samples with three or more memory CD8⁺ T cells were selected for the subsequent analyses as the cytotoxic effector levels may have been too variable in those with less than three memory CD8⁺ T cells.

### Label-free quantitative proteomics

MC38-shNTC and MC38-shv0a1 cells (*n* = 3 biological replicates/group) were harvested to a 1.5 ml centrifuge tube as cell pellets and then lysed with DB lysis buffer (8 M Urea, 100 mM TEAB, pH 8.5). Following ultrasonication and centrifugation, the cell lysate supernatant was treated with 10 mM DTT for reduction and subsequently with sufficient iodoacetamide for alkylation. The protein samples were digested with trypsin treatment as previous study[40], and the resulting peptides were lyophilized and redissolved in Mobile phase A solution (100% water, 0.1% formic acid). After being separated using linear gradient elution, the separated peptides were analyzed by Q ExactiveTM series mass spectrometer (Thermo Fisher) from Novogene Bioinformatics Technology Co., Ltd. The raw data were searched against the Mus_musculus_uniprot_2020_7_2.fasta (86555 sequences) database by Proteome Discoverer 2.4 (Thermo). The identified peptide-spectrum matches (PSMs) and proteins were filtered to ensure a false discovery rate (FDR) of no more than 1.0%. The protein quantification data underwent statistical analysis using the T-test. Moreover, the KEGG (Kyoto Encyclopedia of Genes and Genomes) database was used to analyze the protein family and pathway, and DEPs (differentially expressed proteins) were used for the enrichment analysis of KEGG.

### Cholesterol absorption and endosome maturation analyses

For the detection of cholesterol absorption, HCT-8-shNTC and HCT-8-shV0A1cells on coverslips were treated with 50 μg/ml of human

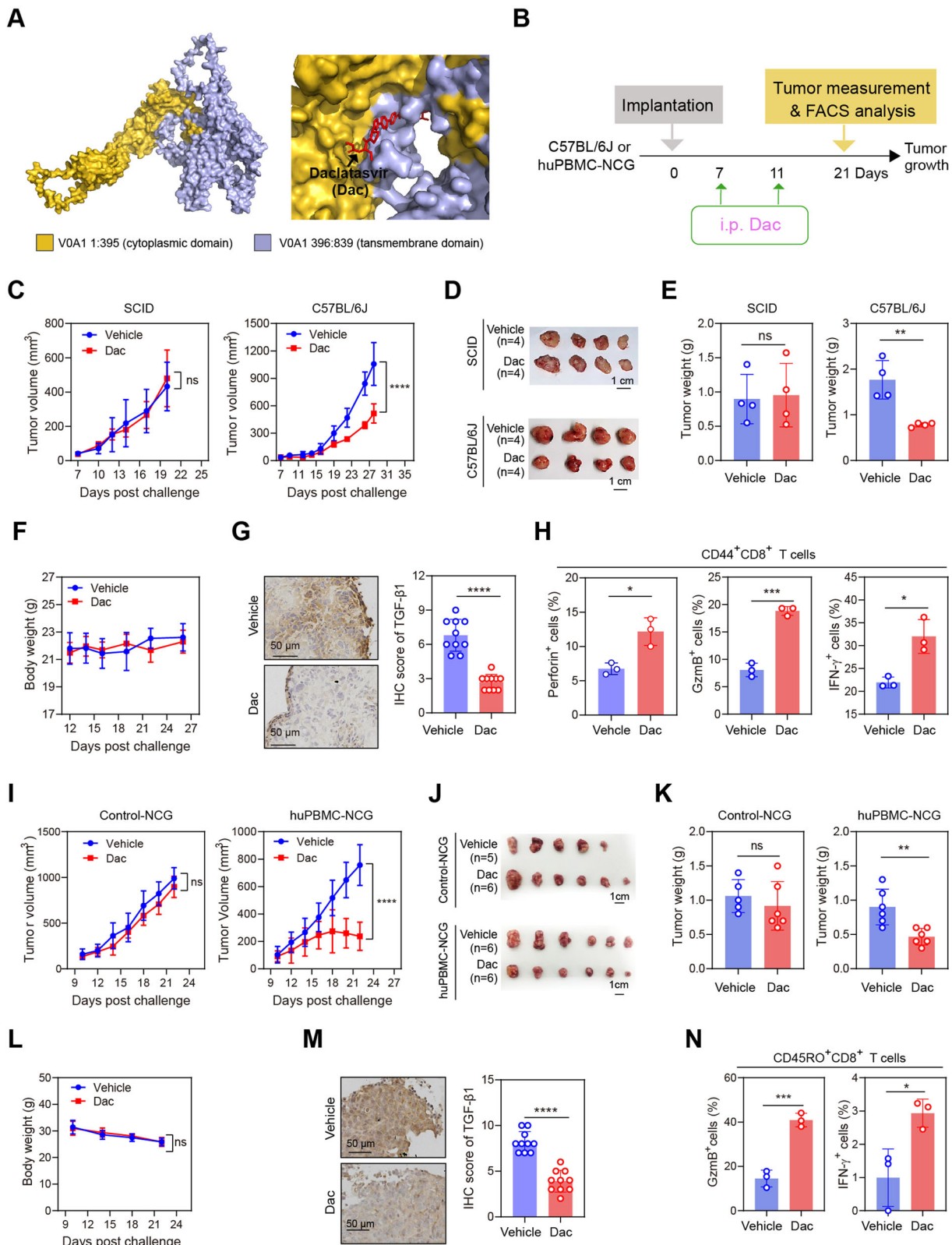

Dil-LDL (YEASEN) for 6 h followed by 10 min of fixation and 10 min of permeabilization. The cells were then separately incubated with primary antibodies specific to various vesicle markers (late endosome: mouse anti-Rab7a from Santa Cruz Biotechnology; lysosomes: mouse anti-Lamp1 from Santa Cruz Biotechnology; recycling endosome: rabbit anti-Rab35 from Proteintech Group, Inc.) followed by incubation with corresponding secondary antibody (Alexa Fluor488-conjugated anti-mouse IgG or Alexa Fluor488-conjugated anti-rabbit IgG; Invitrogen). After the counterstaining of cells with Hoechst 33342 (Invitrogen), the coverslips were subsequently mounted on slides with ProLong™ Diamond Antifade Mountant (Invitrogen). Images were acquired using the LSM880 Confocal Microscope (Zeiss) and analyzed with ZEN lite (Zeiss) and Image J (National Institutes of Health) software.

**Fig. 9 | Dac efficiently suppresses the growth of colorectal tumors in an immune-dependent manner. A** A molecular docking approach predicted Daclatasvir (Dac) as a candidate inhibitor of ATP6V0A1. **B** Schematic showing the drug intervention protocol for Dac therapy. **C–H** NOD/SCID mice and C57BL/6 J mice were subcutaneously injected with wild-type MC38 cells. Tumor-bearing NOD/SCID mice or C57BL/6 J mice were then randomized into two groups according to tumor size and treated with vehicle or Dac. Average curves for tumor growth (**C**) were plotted. Photographs of the tumors (**D**) and comparisons of tumor weights on day 20 in NOD/SCID mice and day 28 in C57BL/6 J mice (**E**) are shown. The body weights of C57BL/6 J mice treated with vehicle or Dac were measured and plotted (**F**). $n = 4$ mice per group for **C–F**. Tissue sections of MC38 tumors from C57BL/6 J mice were analyzed for TGF-β1 expression (**G**; $n = 10$ fields from two mice per group). Tumor-infiltrating CD44+CD8+ T cells from day 28 tumors in C57BL/6 J mice were analyzed for their levels of effector production (**H**; $n = 3$ mice per group). **I–N** NCG mice or huPBMC-NCG mice were subcutaneously injected with HCT-8 wild-type cells.

Tumor-bearing NCG mice or huPBMC-NCG mice were then randomized into two groups according to tumor size and treated with vehicle or Dac. $n = 5$ (Vehicle group in Control-NCG mice model) or 6 mice (other groups) in each group. Average curves for tumor growth (**I**) were plotted. Photographs of the tumors (**J**) and comparisons of tumor weights on day 22 (**K**) are shown. The body weights of huPBMC-NCG mice treated with vehicle or Dac were measured and plotted (**L**). Sections of HCT-8 tumor tissue from huPBMC-NCG mice were analyzed for TGF-β1 expression (**M**; $n = 10$ fields from two mice per group). Tumor-infiltrating CD45RO+CD8+ T cells from day 22 tumors in huPBMC-NCG mice were analyzed for their levels of effector production (**N**; $n = 3$ mice per group). For all experiments, data are representative of 3 independent experiments and shown as means ± s.e.m; *$p < 0.05$, **$p < 0.01$, ***$p < 0.001$, ****$p < 0.0001$. Statistical significance was determined using ordinary two-way ANOVA (in **C, F, I, L**) or unpaired two-sided Student's $t$-test (in **E, G, H, K, M, N**). Source data and exact $p$-value are provided as a Source Data file.

For endosome maturation analysis, MC38 shNTC or MC38 shv0a1 cells growing on coverslips were fixed with 4% paraformaldehyde (PFA) and treated with 0.25% Triton X-100 for 10 min at room temperature (RT). Following the blockade with 10% bovine serum albumin (BSA) in PBS for one hour, the cells were incubated with primary antibodies (rabbit against Rab7a from Cell Signaling Technology and mouse against VPS41 from Santa Cruz Biotechnology) overnight at 4 °C. Following washing, the cells were incubated for one hour at room temperature with corresponding secondary antibodies (Alexa Fluor594-conjugated anti-rabbit IgG and Alexa Fluor488-conjugated anti-mouse IgG; Invitrogen) and then counterstained with Hoechst 33342 (Invitrogen). The coverslips were subsequently mounted on slides with ProLong™ Diamond Antifade Mountant (Invitrogen). Images were acquired and analyzed as described above.

## Immunohistochemical (IHC) and hematoxylin/eosin (HE) analysis

The CRC tissue microarray (TMA) slides were deparaffinized and treated with 10 mM citrate buffer (pH 6.0) at 95–100 °C for 15 min. Then, the tissue cores on slides were treated with 3% $H_2O_2$ for 10 min to block the endogenous peroxidase activity. After washing three times with phosphate-buffered saline (PBS; Gibco), the slides were treated with 0.25% Triton X-100 for 10 min and then blocked in 10% bovine serum albumin (BSA) in PBS for one hour. After incubation with rabbit polyclonal anti-human ATP6V0A1 antibody (Proteintech) for one hour, the tissue cores were washed three times with PBS and then incubated with a biotinylated goat anti-rabbit secondary antibody (Beyotime) for 30 min. The slides were visualized using streptavidin-horseradish peroxidase and 3, 3'-diaminobenzidine (Beyotime) and then counterstained with hematoxylin (Sigma–Aldrich). The images were captured with Slide Scan System (SQS1000; TEKSQRAY).

The frozen sections of ATP6V0A1-suppressing MC38 tumors (*Atp6v0a1* knockdown or Dac treatment) or HCT-8 tumors (Dac treatment) were utilized for IHC analysis of ATP6V0A1 and TGF-β1 expression or TGF-β1 expression alone. Following permeabilization and blockade of non-specific binding, these sections were incubated with antibodies and visualized as the methods described above. Rabbit polyclonal anti-human TGF-β1 antibody (Proteintech) was used as the primary antibody to detect TGF-β1.

The frozen sections of normal organs, including the heart, liver, spleen, lung, and kidney, from Dac-treated mice, were stained with hematoxylin and eosin (HE) according to the manufacturer's instructions (Vector, Burlingame, CA). The images were captured with Slide Scan System (SQS1000; TEKSQRAY).

## Transcriptome analysis based on the TCGA database

Bulk RNA-sequencing data of COAD cohort containing 471 clinical CRC samples were downloaded from The Cancer Genome Atlas (TCGA) database (https://portal.gdc.cancer.gov/). The normalization method

of the fragments per kilobase of exon model per million reads mapped (FPKM) was employed to evaluate the expressional level of genes in CRC samples. Eighty-seven metabolism-associated pathways of human species were acquired from the website of the Kyoto Encyclopedia of Genes and Genomes (KEGG) pathway (https://www.genome.jp/kegg/) by the use of the R package of KEGGREST (version 1.40.0)[41]. Sixteen lipid metabolism-associated pathways, as indicated on the website of KEGG, were selected for further investigation. The R package of GSVA (Gene Set Variation Analysis)[42] using the gene-centric single sample Gene Set Enrichment Analysis (ssGSEA) method[43,44] was executed to assess the transcriptomic levels of each lipid metabolism pathway. Total lipid metabolism scores for each sample were obtained by summing up these 16 lipid metabolism-associated pathways. The mean value of lipid metabolism scores within all CRC samples was adopted as a cutoff value to dichotomize these samples into two stratifications: low-lipid and high-lipid metabolism groups. In parallel, the R package of ESTIMATE (Estimation of STromal and Immune cells in MAlignant Tumor tissues using Expression data)[45] was utilized to calculate the scores for tumor cells, immune cells and stromal cells, and the R package of GSVA (Gene Set Variation Analysis)[42] using gene-centric single sample Gene Set Enrichment Analysis (ssGSEA) method[43,44] was carried out to analyze the immune activity in each sample. Total immunity scores were obtained by summing up 29 distinct immune scores for individual CRC samples. An unsupervised machine learning algorithm of hierarchical clustering was applied to classify CRC samples into three immune profile-defined clusters: CRC tissues with low, medium, and high immune activity. Subsequently, the transcriptomic expression levels of various V-ATPase subunits were analyzed for their associations with lipid metabolism scores or immune activity. Furthermore, CRC samples with high or low lipid metabolism were divided into high and low ATP6V0A1 separately. The cutoff value for the grouping was determined by taking the average level of ATP6V0A1 expression within all samples. The association between ATP6V0A1 expression and immune activity was analyzed individually in the samples with high or low-lipid metabolism.

## Cellular thermal shift assay (CETSA)

The cell lysate CETSA experiments were performed as previously described[24,46] with some modifications. In brief, MC38 wild-type or HCT-8 wild-type cells were harvested, washed with PBS, and suspended in PBS containing a protease inhibitor cocktail (Roche). Following three freeze-thaw cycles using liquid nitrogen, the cell suspensions were centrifuged at $20,000 \times g$ for 20 min at 4 °C to separate the soluble fraction (lysate) from the cell debris. The cell lysates were diluted in the appropriate buffer and then divided into smaller aliquots, followed by the treatment with different concentrations of drug (Dac) or equal amounts of diluent. After 30 min of incubation at room temperature, the respective lysates were heated at the appropriate temperature for 3 min, then cooled for 3 min at room

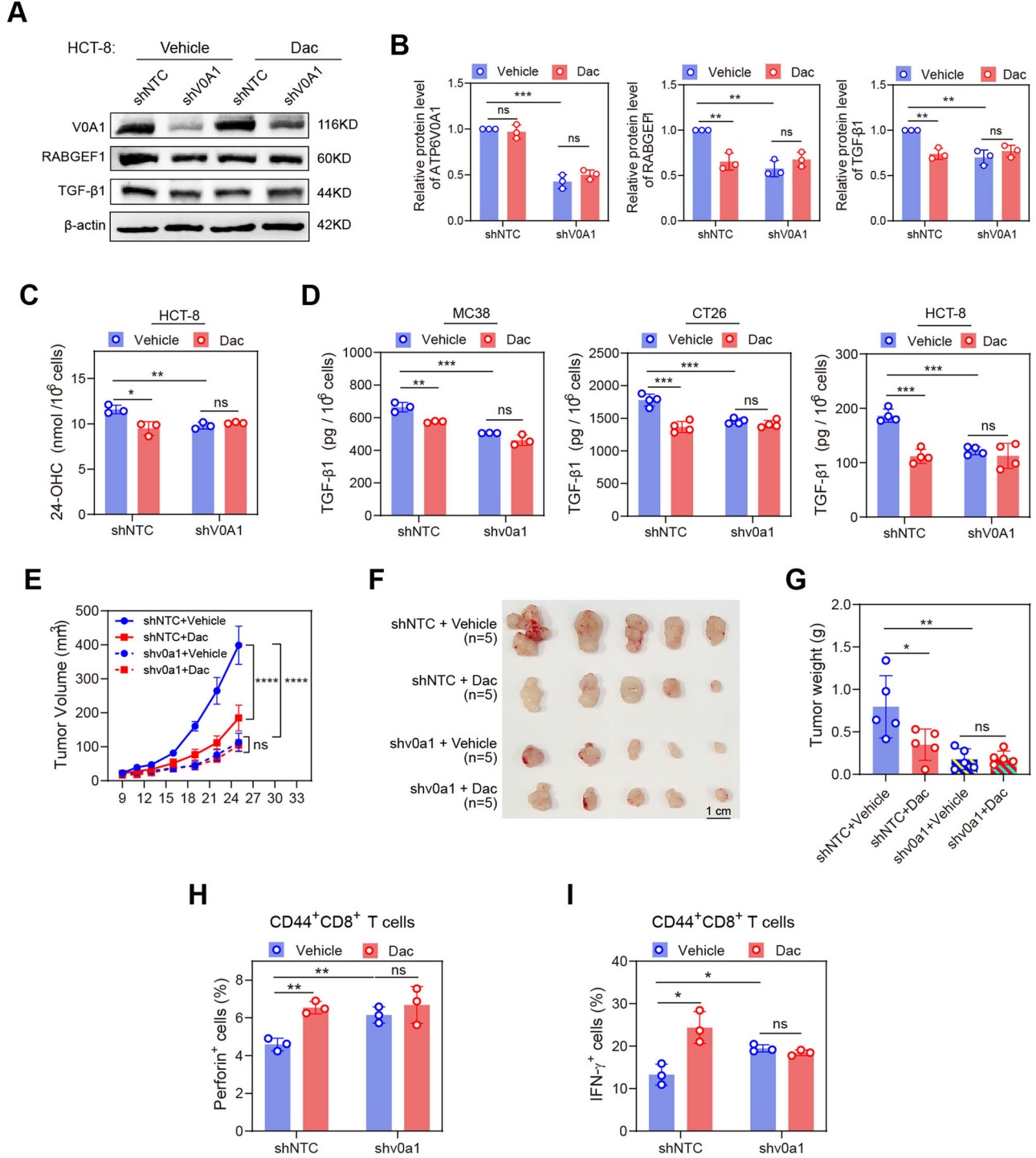

**Fig. 10 | Targeting ATP6V0A1 is required for Dac to suppress colorectal tumors.**
**A–C** ATP6V0A1-suppressing HCT-8 cells were treated with vehicle or Dac. The cell lysate was detected with the expression of ATP6V0A1, RABGEF1, and TGF-β1 using western blotting; The presentive blots were shown (**A**), and the quantification of these proteins was analyzed based on three independent experiments (**B**). The supernatant was detected by ELISA for 24-OHC level (**C**; $n = 3$ independent experiments). **D** ATP6V0A1-suppressing MC38, CT26, and HCT-8 cells treated with vehicle or Dac, and the level of TGF-β1 in the supernatant was detected by ELISA. $n = 3$ (MC38) or 4 (CT26 and HCT-8) independent experiments. **E–I** C57BL/6 J mice were subcutaneously injected with control or *Atp6v0a1*-suppressing MC38 cells.

The mice bearing control MC38 tumors or those bearing *Atp6v0a1*-suppressing MC38 tumors were separately randomized into two groups according to tumor size and treated with vehicle or Dac. $n = 5$ mice per group. Average curves for tumor growth (**E**) were plotted. Photographs of the tumors (**F**) and comparisons of tumor weights (**G**) are shown. Tumor-infiltrating CD44+CD8+ T cells from the above tumors described in **E–G** were analyzed for their levels of effector production (**H** and **I**; $n = 3$ mice per group). For all experiments, data are shown as means ± s.e.m; *$p < 0.05$, **$p < 0.01$, ***$p < 0.001$. Statistical significance was determined using ordinary two-way ANOVA (in **E**) or unpaired two-sided Student's *t*-test (in **B**, **C**, **D**, **G**, **H**, **I**). Source data and exact *p*-value are provided as a Source Data file.

temperature. The appropriate heating temperatures for detecting the changes of ATP6V0A1 or β-actin thermostability mediated by the inhibitors were determined separately in preliminary CETSA experiments. Following the centrifugation of heated lysates at $20,000 \times g$ for 20 min at 4 °C, the soluble fractions were isolated and further analyzed by Sodium dodecyl sulfate-polyacrylamide gel electrophoresis (SDS-PAGE) and Western blot.

## Statistical analysis

All quantitative data are presented as means ± standard error of the mean (s.e.m.) unless otherwise indicated. Statistical analyses were performed using unpaired two-tailed Student's $t$-tests or ordinary two-way ANOVA, using Prism statistical software (GraphPad Software, Inc., San Diego, CA, USA). A $P$-value less than 0.05 was considered significant. $*P < 0.05$, $**P < 0.01$, $***P < 0.001$, $****P < 0.0001$.

## Reporting summary

Further information on research design is available in the Nature Portfolio Reporting Summary linked to this article.

## Data availability

The scRNA-seq data used in this study are available in the NIH GEO database under accession code GSE238084. The mass spectrometry proteomics data generated in this study have been deposited in the ProteomeXchange database under accession code PXD044010. The bulk RNA-sequencing data of COAD cohort containing 471 CRC samples were downloaded from The Cancer Genome Atlas (TCGA) database. Source data are provided with this paper.

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

## Acknowledgements

This work was supported by grants from the National Natural Science Foundation of China [82173003 (to L.F.), 32370968 (to T.X.H.), 82170881 (to X.F.L.), 81870605 (to X.F.L.), and 82173016 (to J.B.)], the National Key R&D Program of China [2017YFA0503900 (to L.F.)], the Science and Technology Program of Guangdong Province [2019B030301009 (to L.F.) and 2021A1515010707 (to J.B.)], the Guangdong Basic and Applied Basic Research Foundation [2024A1515010539 (to X.F.L.)], and the Science and Technology Foundation of Shenzhen [JCYJ20200109113810154 (to L.F.)].

## Author contributions
T.X.H., H.S.H., and B.Z acquired the animal data. T.X.H., H.S.H., T.T.Z., J.Y.C., and Y. Y. performed the in vitro cell experiments. S.W.D., H.H.L., and Q.Y.L. performed the TCGA-COAD database analysis, scRNA-seq analysis and molecular docking analysis, Q.X., P.Z., J.B., L.Y.H., and F. P. acquired the tissue IHC/IF data, and X.F.L. designed and constructed the plasmids. T.X.H. and L.F. designed the study, analyzed the data, and wrote the manuscript. C.Z. and L.F. supervised the study and revised the manuscript.

## Competing interests
The authors declare no competing interests.
