## [Peer Review File · Nature Communications]

ATP6V0A1-dependent cholesterol absorption in colorectal cancer cells triggers immunosuppressive signaling to inactivate memory CD8⁺ T cellsREVIEWER COMMENTS

Reviewer #1 (Remarks to the Author):

Huang et al. identified a unique mechanism by which colorectal cancer (CRC) cells can escape from the immune system. First, authors noted the inverse correlation between the expression level of ATP6V0A1, one of V-ATPase subunits, in CRC cells and immune activity. Based on this observation, the authors demonstrated that knockdown of Atp6v0a1 in CRC cells potentially enhanced tumor killing activity of CD8+ T cells in vivo. Mechanistically, ATP6V0A1 promotes cholesterol absorption in CRC cells via RABGEF1-dependent endosome maturation. As a result, intracellular 24-OHC level increases in CRC cells due to the cholesterol accumulation in ER. 24-OHC then upregulate TGF- β 1 expression in CRC cells by activating LXR pathway. Finally, CRC-derived TGF- β 1 activates SMAD3 pathway in CD8+ T cells which in turn attenuate their tumor killing activity. Authors have further demonstrated that dactalatasvir (Dac), clinically used anti-HCV drug, binds and inhibits ATP6V0A1, and can effectively suppress tumor growth in CRC by activating CD8+ T cells. The experiments were performed to a high standard and logically, and the authors' conclusions are well supported by the experimental results. There are, however, several points that need to be clarified as follows;

1. What promote the expression level of ATP6V0A1 in tumor cells? Is it due to metabolites derived from host?
2. In the first section, the authors demonstrated the correlation between the expression level of ATP6V0A1 and lipid metabolism and anti-tumor immunity. There was no causal relationship. It is thus inappropriate to say that positive association of ATP6V0A1 expression with high lipid metabolism and decreased immune activity indicates that lipid metabolic reprogramming is required for ATP6V0A1-suppressed anti-tumor immunity (lines 99-100).
3. Supplementary Fig. 5g strongly suggest that CTL activity is more important in ATP6V0A1<low> CRC. In ATP6V0A1<high> CRC, CTL activity does not seem to be important for the overall survival, which is better than ATP6V0A1<low> CRC. This seems contradictory to the notion that ATP6V0A1 level inversely correlated with immune activity.
4. In supplementary Fig. 10, authors demonstrated that percentage of macrophages decreases in Atp6v0a1 knockdown-derived tumors. Does it affect the function of CD8+ T cells?
5. In Figure 2c, authors demonstrated that Tgfb2 is highly expressed in tumor infiltrating CD8+ T cells. Does this phenotype apply to CD8+ T cells outside of tumors as well? Also, why CD8+ T cells cannot penetrate into ATP6V0A1-sufficient tumor cells although other immune cells can?
6. Fig. 3a needs a control (shNTC-treated cells).
7. Supplementary Fig. 11 need quantification as Fig. 5.
8. Line 232: "higher" seems better than "enhanced", which seems misleading.
9. What may happen to tumor growth in vivo/in vitro if Tgf β 1 is knocked down in ATP6V0A1-sufficient tumor cells?
10. What may happen to Tgf β 1 expression in Atp6v0a1-overexpressing tumor cells?
11. In Fig.4, authors should knockdown CYP46A1 and 24-OHC respectively so as to demonstrate that they are all required for regulating Tgf β 1 expression in tumor cells.
12. What may happen to 24-OHC and ER-cholesterol levels in Atp6v0a1-overexpressing tumor cells?
13. LXR is comprised of LXR-alpha and LXR-beta. Which of them are involved in 24-OCT-LXR mediated Tgf β 1 expression in tumor cells?
14. In Fig.5, authors should knockdown Rab7a so as to demonstrate that it is required for regulating Tgf β 1 expression in tumor cells.

15. What may happen to RABGEF-1 and Rab7a expression in Atp6v0a1-overexpressing tumor cells?

16. Some of the WB analysis do not seem to support authors' conclusion. Throughout the manuscript, authors should quantify the signal intensity of bands.

17. Genetic background sometimes interferes with immune responses in mice. Is NOD/SCID mice established with B6J background? If not, authors should do so, or use immunodeficient mice such as Rag1^{-/-}, Rag2^{-/-}, or Il2rgRag2^{-/-} mice with B6J background. This also applies to the comparison between NOD/SCID and BALB/c mice.

18. Some of anti-CD8 mAb such as 53-6.7 downregulates the expression of CD8 but unable to deplete CD8⁺ cells. The authors should provide the clone name for anti-CD8 (Lyt2.1) mAb.

Minor points

MC38 are mistyped as MC8 several times; they can be found in page 8, line 129 and page 13, line 251.

Reviewer #2 (Remarks to the Author):

This is an interesting paper addressing a significant and timely topic in cancer biology. The study is based on the premise that lipid metabolism impacts anti-tumor immunity in colorectal cancer (CRC). The authors' overarching goal is to understand the mechanisms whereby cholesterol metabolism regulates the immune response in CRC. Intriguingly, the paper initially focuses on ATP6V0A1, which the authors convincingly show modulates the mechanisms of immunosurveillance in cell-lines-based models of CRC. From those relatively solid initial experiments, they jump to the idea that the regulation of TGFbeta levels by ATP6V0A1 in cancer cells is what accounts for the effect that manipulating ATP6V0A1 has in the CD8 T cell response. However, the rescue experiments are not robust since the effects of ablating TGFbeta in vivo are only partial (Figs 3I-3L). Therefore, although TGFbeta might be relevant in their model, they do not rule out other potential immunosuppressor mechanisms.

From these experiments, the authors do another (big) conceptual jump based on the data of Fig. 4A. From these, they conclude that cholesterol metabolism is affected by the inactivation of ATP6V0A1 in CRC cells. However, they totally ignore other metabolic pathways, which are even more significantly affected, such as arginine and proline metabolism. Why focus on cholesterol? What is the contribution of the other pathways to the phenotype of ATP6V0A1-deficient cells? This is a very important question because the central role of ATP6V0A1 in organelle biology suggests that it will impact many pathways, which might contribute to its role in the immune system. Their model that increased 24-OHC levels promotes TGFbeta upregulation through LXR, is very interesting but, again, needs a more thorough analysis and validation. The rescue experiments in vivo are only partial (Fig. 4G) and there is a general lack of a rigorous metabolomic approach to place the 24-OHC/LXR/TGFbeta pathway in the context of other alterations in the cholesterol (and non-cholesterol) metabolism. Without that broader approach, it is very difficult to be certain that what the authors proposed is what actually accounts for the role of ATP6V0A1 in the control of immunoevasion in CRC. In the same vein, the focus on RABGEF1 is also surprisingly underdeveloped from the mechanistic point of view. Much more rigorous mechanistic studies are needed to convincingly demonstrate that the model proposed here actually accounts for the effect of perturbing ATP6V0A1.

In addition, the study mostly uses cell lines that might or might not represent what CRC actually is. For example, MC38 is considered to have some "similarity" to the CMS1 (MSI) type of CRC, and, therefore, is representative of tumors that are "hot" from an immunological point of view. This should be considered when focusing on the concept in this paper that ATP6V0A1 is upregulated in CRC. How do the authors reconcile their data with the fact that MC38 cells are actually quite active from the immunosurveillance point of view? In which type of CRC is ATP6V0A1 really relevant? In this regard, the analysis of the human CRC specimens is partial and does not take into account a more rigorous understanding of the biology and pathology of CRC and the more contemporary classification in the field. In connection with this, the utilization of organoids with defined genomic and transcriptomics profiles better representing the different types of CRC is a must, as it is also the utilization of PDOs with also defined genetic profiles. Finally, the pharmacological approach using repurposed drugs needs many controls to ensure that the effects reported here are actually specific to cholesterol metabolism.

In summary, this is a potentially interesting and ambitious manuscript addressing many different

angles of the highly significant biological problem of immunoevasion in CRC, but all these aspects need much more controls and more rigorous validation, both functional and mechanistic to support their models and conclusions rigorously.

Reviewer #3 (Remarks to the Author):

Huang et al. investigate a novel function of the V-ATPase VoA1 subunit in immune suppression by colorectal cancer cells. Intriguingly, they observe that depletion of VoA1 selectively impairs tumor growth in immunodeficient mice. Using a range of experimental approaches, they investigate a mechanistic connection between VoA1, cholesterol absorption, lipid signaling and immunosuppression. The manuscript spans an enormous range of topics to characterize this process in cancer cells, tumors and patient samples. However, this comes at the cost of a lack of mechanistic depth and results that are sometimes of correlative nature.

Major concerns:

1. The analysis of patient data is correlative and sometimes difficult to interpret (Fig. S1 – 4). It is unclear what 'high lipid metabolism' means. For example, fatty acid degradation and synthesis seem to be upregulated; terms such as primary bile acid and cutin/suberin/wax synthesis are included – what does this mean in the context of CRC?
2. The impact of VoA1 knockdown on tumor growth in immune-deficient mice is intriguing. However, one would normally expect the use of multiple control/target shRNAs, at least in key experiments. This is important to exclude off-target effects, and because the depletion of VoA1 sometimes appears to be rather modest (Fig. S6).
3. The authors provide evidence that VoA1 is important for lysosomal cholesterol mobilization from LDL, which affects ER cholesterol levels, 24-OHC production and TGF- β signaling. This conclusion could be strengthened by demonstrating that removing extracellular lipids has similar consequences.
4. While the role of V-ATPase-mediated endosomal acidification in endosomal trafficking is well established, the authors suggest that VoA1 depletion blocks lysosomal LDL transport by decreasing Rabgef1. Is this a VoA1-specific effect, or mediated by the V-ATPase complex? Is proton-pumping activity of VoA1-containing V-ATPases required? Does depletion of other VoA subunits have similar effects? Does VoA1 deficiency have any effect on vesicle acidification?
5. The data on Daclatasvir as a potential VoA1 inhibitor are not convincing. How does Dac binding affect VoA1 – does it suppress proton pumping? Does Dac alter the thermal stability of other V-ATPase subunits, in particular other VoA subunits? The authors show that Dac reduces Tgf- β levels in cells expressing VoA1 shRNA (depletion of the protein is not shown in the blot), and conclude that Dac acts on VoA1. This is a very indirect readout of VoA1 function – a more direct demonstration of VoA1 inhibition would be essential. Moreover, it would be important to demonstrate that Dac does not affect the growth of VoA1-depleted tumors.

Reviewer #4 (Remarks to the Author):

Although lipid metabolism is known to modulate antitumor immunity, it remains unknown how the tumor immune microenvironment and lipid metabolism are related in colorectal cancer (CRC). Tu-Xiong Huang et al found that ATP6V0A1 expressed by tumor cells induced immunosuppression using exogenous cholesterol in the tumor microenvironment (TME) of CRC. ATP6V0A1 promotes cholesterol absorption by cancer cell, and cholesterol accumulation within the endoplasmic reticulum increased 24-OHC, an oxidized derivative of cholesterol and upregulated TGF- β 1 by activating the LXR signaling. Subsequently, TGF- β 1 activates the SMAD3 pathway in memory CD8+ T cells, and suppresses their anti-tumor activities. Finally, the authors demonstrated that daclatasvir, a clinically used anti-HCV drug, as an ATP6V0A1 inhibitor, improved the activity of memory CD8+ T cells and suppressed tumor growth in CRC. The concept that targeting ATP6V0A1 of CRC suppress tumor growth by activating memory CD8+T cells in the TME is interesting. However, the impact of this study is severely impaired since there

are critical problems with the experimental method and interpretation of the data, and several points were not adequately addressed.

General comments;

In supplementary fig.1 to supplementary fig.3, the authors divide TCGA dataset into lipid metabolism high group and low group, and ATP6V0A1 was selected among the V-ATPase subunits highly expressed in the lipid metabolism high group as the most inversely correlated with immune score. However, several subunits other than ATP6V0A1, for instance ATP6V0C, ATP6V0D1 and ATP6V0E1, show opposite trend in the immune score despite their increased expression in the lipid metabolism high group. Moreover, although ATP6V0A1 in CRCs mainly affects CD8+ T cell function in this study, all immune cell populations are considered as the immune score. Therefore, it is necessary to clarify how ATP6V0A1 differs in expression and function compared to other V-ATPase subunits in colorectal cancer cells, and why ATP6V0A1 is more important in lipid-mediated immunosuppression than the other subunits. In addition, the flowcytometry (FCM) analysis in Figure 2 shows not including live dead, CD45, or CD3 staining, therefore, target cells are not properly gated and the reliability of the subsequent analysis cannot be ensured.

Specific points;

1. Quantitative evaluation should be added in all Western blotting
2. Numbers on the Y-axis should start from zero (Fig.3e, 3f, 4d, 4e, 4f, 4g, 6a, 6b, 6c etc).
3. In supplementary fig.4, bulk MC38 tumors include a lot of tumor-infiltrated immune cells and stroma cells. The increased expression of ATP6V0A1 by HFD is not necessarily derived from tumor cells.
4. In supplementary fig.5, the authors evaluate the expression of ATP6V0A1 by immunohistochemistry (IHC) using specimens from CRC patients. Is high expression of ATP6V0A1 in IHC staining associated with low CD8+T cell infiltration?
5. In supplementary fig.10, single cell RNA seq is powerful to identify the specific immune cell subset. However, the authors should be careful when using scRNAseq for quantitative evaluation. FCM is advantageous for quantitative evaluation, which also available for evaluation at protein level. If there is a marker that characterizes memory like T-2 subset, it is necessary to confirm these cells by FCM or IHC and whether the number of T cells expressing that marker is increasing in the tumor.
6. Representative FCM images are necessary for Fig.2i and Fig.3f.
7. In Fig.3b, CD8+ cells were isolated from tumors. Their purity should be shown.

Mar 17, 2024

Dear Reviewers,

Re: Manuscript ID NCOMMS-23-46893-T

Title: ATP6V0A1-dependent cholesterol absorption in colorectal cancer cells triggers immunosuppressive signaling to inactivate memory CD8+ T cells

Thank you for reviewing the above-referenced manuscript submitted earlier to *Nature Communications*. We would like to take this opportunity to express our appreciation for your constructive comments and suggestions. Following the Reviewers' comments and suggestions, the manuscript has been revised accordingly. The revisions in the manuscript text were marked as red. We feel that this revised manuscript has been strengthened by the Reviewers' comments and suggestions. A point-by-point response to the Reviewers' comments and suggestions has been prepared and shown below.

I thank you again for reviewing the manuscript and look forward to hearing your favorable reply soon.

Yours sincerely,

Grace L. Fu, MD; PhD.

Professor
Director, Laboratory of Tumor Microenvironment,
International Cancer Center,
Shenzhen University Medical School,
Room 212, A1 Block, Xili Campus
1066 Xueyuan Road, Nanshan District, Shenzhen
Tel: (86) 0755-86671992; E-mail: gracelfu@szu.edu.cn

A point-by-point response to the Reviewers' comments and suggestions

Reviewer: 1

Q1: What promote the expression level of ATP6V0A1 in tumor cells? Is it due to metabolites derived from host?

●Response:

Thanks to the reviewer for the insightful comments. It is a noteworthy issue that how tumor cells upregulate ATP6V0A1. Understanding this mechanism could have important theoretical and clinical implications. However, due to limited manuscript space, in the present study, we mainly focused on the role and mechanism of ATP6V0A1 utilizing exogenous cholesterol to promote CRC immune escape after being upregulated in CRC cells, and have not yet looked for the upstream mechanism responsible for ATP6V0A1 overexpression. Regarding the regulation of ATP6V0A1 in CRC cells, we agree with reviewer's opinion that the metabolites may be a possible factor to increase ATP6V0A1 in CRC cells. In fact, by analyzing the bulk RNA sequencing data from the GSE157994 database, we found that the MC38 grafts in mice treated with a high-fat diet (HFD) exhibited elevated levels of tumor cell-intrinsic Atp6v0a1, compared to those in mice treated with a control diet (CD) (**Reply letter Figure 1A, B**), suggesting that exogenous lipids may induce Atp6v0a1 upregulation in CRC cells. Moreover, high concentrations of exogenous cholesterol can induce upregulation of Atp6v0a1 in MC38 cells (**Reply letter Figure 1C**). However, further investigation is required to explore the detailed mechanism by which exogenous lipids increase the expression of ATP6V0A1 in CRC cells. Additionally, ATP6V0A1 could also be increased by the upstream molecular signaling driven by other cancer genes. We will investigate the upstream mechanisms of ATP6V0A1 upregulation in CRC cells in future studies.

A

HFD induced Atp6v0a1 upregulation in MC38 cells

Samples [^]	CD						HFD					
	#1	#2	#3	#4	#5	#6	#9	#10	#11	#12	#13	#14
Atp6v0a1 (TPM)	19.06	20.50	26.96	21.03	28.15	18.08	22.79	31.95	32.92	21.83	35.56	29.91
Relative expression ^δ	0.85	0.92	1.21	0.94	1.26	0.81	1.02	1.43	1.48	0.98	1.59	1.34

[^] The sample ID is consistent with that in GSE157994; Six samples from each group were analyzed.

^δ The mean value in the CD group was normalized to 1.

Huang *et al* Reply letter Fig. 1

Q2: *In the first section, the authors demonstrated the correlation between the expression level of ATP6V0A1 and lipid metabolism and anti-tumor immunity. There was no causal relationship. It is thus inappropriate to say that positive association of ATP6V0A1 expression with high lipid metabolism and decreased immune activity indicates that lipid metabolic reprogramming is required for ATP6V0A1-suppressed anti-tumor immunity (lines 99-100).*

•Response:

Thanks to the reviewer for the kind suggestion. In addition to showing positive association of ATP6V0A1 expression with high lipid metabolism and decreased immune activity (**Supplementary Figs. 1 and 2**), we also found that ATP6V0A1 expression was positively associated with decreased immune activity only in the CRCs with high lipid metabolism. No correlation was observed in the CRCs with low lipid metabolism (**Supplementary Fig. 3**). Moreover, in the revised manuscript, we added new data to show that high lipid metabolism was positively associated with decreased immune activity only in the CRCs with high ATP6V0A1 expression, but not in those with low ATP6V0A1 expression (**Supplementary Fig. 4**). These data suggest that high level of lipid metabolism is important to ATP6V0A1-suppressed anti-tumor immunity and high level of ATP6V0A1 is required to reversely associate lipid metabolism level with immune activity. The overall level of lipid metabolism in tumor tissues may be

associated with lipid levels in tumor microenvironment (TME) since that high levels of exogenous lipids caused by high-fat diet (HFD) can increase the overall level of lipid metabolism in tumors (**Cell. 2020 Dec 23;183(7):1848-1866**). Therefore, the data in Supplementary Figs 1-3 may indicate the important roles of ATP6V0A1 in exogenous lipid-induced CRC immune suppression. Importantly, to test this hypothesis, we conducted an experiment to investigate the roles of HFD in inducing tumor promotion and immune suppression in control or ATP6V0A1-suppressed MC38 tumors. As shown in the newly added Fig. 1, HFD promoted tumor growth and suppressed anti-tumor immunity in control MC38 tumors, and ATP6V0A1 depletion in tumor cells blocked these changes induced by HFD. These data determined the important roles of ATP6V0A1 in exogenous lipid suppressed anti-tumor immunity of CRCs. We thus aimed to investigate the role of ATP6V0A1 in regulating anti-tumor immunity through exogenous lipids.

Q3: *Supplementary Fig. 5g strongly suggest that CTL activity is more important in ATP6V0A1<low> CRC. In ATP6V0A1<high> CRC, CTL activity does not seem to be important for the overall survival, which is better than ATP6V0A1<low> CRC. This seems contradictory to the notion that ATP6V0A1 level inversely correlated with immune activity.*

●**Response:**

Thanks for bringing up this concern. As shown in **Fig. 8E** (corresponding to Supplementary Fig. 5g in the previous version of the manuscript), the overall survival is better in ATP6V0A1<high> CRC than in ATP6V0A1<low> CRC when the infiltration of cytotoxic T lymphocytes (CTL) is insufficient (CTL bottom). Consistent with this, ATP6V0A1-suppressing HCT-8 cells grow faster than the control cells in the immunodeficient mice (**Fig. 2J**). However, when the infiltration of CTLs is sufficient, the overall survival is better in ATP6V0A1<low> CRC than in ATP6V0A1<high> CRC. This is consistent with the animal experiment in which ATP6V0A1-suppressing HCT-8 cells grow significantly slower than the control cells in the mice constructed with human PBMC (**Fig. 2J**). Although the alteration of overall survival between ATP6V0A1<low> CRC and ATP6V0A1<high> CRC in sufficient CTL-infiltration cases may not be statistically significant due to the limited number of cases, the alteration trend is consistent with the notion that ATP6V0A1 levels are inversely correlated with immune activity. With **Fig. 8E** (corresponding to Supplementary Fig. 5g in the previous version of the manuscript), we aim to show that higher infiltrations of CTLs improve survival only in ATP6V0A1<low> CRC, while no alteration is observed in ATP6V0A1<high> CRC. These data suggest that high levels of ATP6V0A1 in tumors may impair the activity of CD8+T cells infiltrated into tumors.

Q4: *In supplementary Fig. 10, authors demonstrated that percentage of*

macrophages decreases in *Atp6v0a1* knockdown-derived tumors. Does it affect the function of CD8+ T cells?

•**Response:**

Thanks to the reviewer for the comment. Although the percentage of macrophages decreases in *Atp6v0a1* knockdown-derived tumors as indicated in the scRNA-seq data, it only slightly increases (by approximately 1.3 folds) in *Atp6v0a1* knockdown-derived tumors. Importantly, the ratio of M1 to M2 macrophages does not significantly change (**Reply letter Figure 2**). Therefore, we did not delve into the role of macrophages in ATP6V0A1-regulated immunosuppression. Moreover, since the ratio of memory-like T-1 cells (a cytotoxic subtype of memory-like T cells) changes most significantly in *Atp6v0a1*-knockdown tumors compared with in control tumors as indicated in the scRNA-seq data, and the change of memory-like T cell activities can be demonstrated by FACS, we thus focus on the roles of memory-like T cell activities in ATP6V0A1-regulated immune evasion in the present study. Regarding the mechanism of ATP6V0A1 regulating memory-like T-cell activities, we found a direct pathway (paracrine TGF- β 1/SMAD3 pathway) between tumor cells and CD8+T cells that is important for this regulation. Therefore, we focus on regulating memory-like T-cell activities directly by tumor cells in the present study and will investigate the roles of other immune cells, including macrophages, in the context of ATP6V0A1-regulated immune evasion.

Huang *et al* Reply letter Fig. 2

Q5: In Figure 2c, authors demonstrated that *Tgfb2* is highly expressed in tumor infiltrating CD8+ T cells. Does this phenotype apply to CD8+ T cells outside of tumors as well? Also, why CD8+ T cells cannot penetrate into ATP6V0A1-sufficient tumor cells although other immune cells can?

•**Response:**

Thanks to the reviewer for these questions. (1) By analyzing the scRNA-seq database of CRC patient-derived PBMC, we found that TGFBR2 is highly expressed in CD8⁺ effector T cells and NKT cells among different T-cell subpopulations of PBMCs (**Reply letter Figure 3**). (2) According to our scRNA-seq data shown in Supplementary Fig. 9A and E (corresponding to Supplementary Fig. 10A and E in the last-version of the manuscript), both CD8⁺ and CD4⁺ T cells are obviously decreased in ATP6V0A1-sufficient tumors. However, as shown in the below **Reply letter Figure 4A**, both CD8⁺ and CD4⁺ T cells increased only slightly in ATP6V0A1-suppressing tumors (CD8⁺: ~1.1 folds; CD4⁺: ~1.2 folds;) by FACS assay. Moreover, ATP6V0A1 interference also did not significantly change the percentage of memory CD8⁺T cells in MC38 tumors (**Reply letter Figure 4B**). Therefore, tumor cell-derived ATP6V0A1 mainly regulates immune evasion by suppressing the activities of memory CD8⁺T cells (**Pls see newly revised Fig. 3**) rather than by inhibiting the infiltration of CD8⁺T cells.

Huang *et al.* Reply letter Figure 3

Teff: effector T cells; Tn: naïve T cells; Tcm: central memory T cells

Huang *et al* Reply letter Fig. 4

Q6: Fig. 3a needs a control (shNTC-treated cells).

•**Response:**

Thanks to the reviewer for the kind suggestion. In Fig. 3a of the previous version of the manuscript (corresponding to Fig. 4A in the revised manuscript), we showed the ratio of *Smad3* or *Id2* expression in each lymphocyte subset of ATP6V0A1-suppressing tumors to that in the corresponding subset of control (shNTC) tumors to demonstrate more intuitively the effect of V0A1 interference on the expression levels of *Smad3* and *Id2* in different lymphocyte subsets.

Q7: Supplementary Fig. 11 need quantification as Fig. 5.

•**Response:**

In accordance to reviewer's suggestion, a quantification analysis has been provided in **Supplementary Fig. 10D**. Pls note that Supplementary Fig. 10 is corresponding to Supplementary Fig. 11 in the previous version of the manuscript.

Q8: Line 232: "higher" seems better than "enhanced", which seems misleading.

•**Response:**

In accordance to reviewer's suggestion, "enhanced" has been replaced by "higher" in the revised manuscript (**Line 12 on Page 13**).

Q9: What may happen to tumor growth *in vivo/in vitro* if *Tgfβ1* is knocked down in ATP6V0A1-sufficient tumor cells?

•**Response:**

Thanks to the reviewer for the kind suggestion. To investigate the role of tumor cell-derived TGF-β1 in ATP6V0A1-promoted tumor growth *in vivo*, we constructed ATP6V0A1-suppressing MC38 cells with TGF-β1 expression recovery. As shown in **Supplementary Fig. 12**, the recovery of TGF-β1 expression can overcome the suppression of MC38 tumor growth caused by ATP6V0A1 depletion. It is worth noting that the level of TGF-β1 after recovery in ATP6V0A1-suppressing cells is comparable to that in control cells (**Supplementary Fig. 12A**). Moreover, TGF-β1 neutralizing Ab can suppress the growth of MC38 tumors enhanced by ATP6V0A1 overexpression (**Fig. 4J-L**). These data suggested TGF-β1 mediate ATP6V0A1-regulated tumor growth *in vivo*. Since ATP6V0A1 do not consistently and significantly affect the *in vitro* growth of CRC cells, we did not delve into the roles of ATP6V0A1-induced TGF-β1 in regulating the *in vitro* growth of CRC cells. In the present study, we actually focus on the role of ATP6V0A1 in regulating immune evasion. Our data suggest that ATP6V0A1-induced TGF-β1 may mainly affect the growth of CRC tumors

by influencing anti-tumor immunity. However, we do not exclude the possibility that the alteration of TGF- β 1 in CRC cells may have a direct role in the growth of CRC cells, which could depend on the extent of change in TGF- β 1. The TGF- β 1 induced by ATP6V0A1 in CRC cells may be sufficient to affect the immune response, but insufficient to significantly affect the growth of CRC cells. We will address these issues in future studies.

Q10: *What may happen to Tgf β 1 expression in Atp6v0a1-overexpressing tumor cells.*

•Response:

In accordance to reviewer's suggestion, we conducted additional experiments to analyzed the TGF- β 1 expression in Atp6v0a1-overexpressing MC38 cells (**Fig. 4F**) and CT26 cells (**Supplementary Fig. 20A-E**). To more accurately study the impact of elevated ATP6V0A1 levels on tumor cell progression, ATP6V0A1 was overexpressed in ATP6V0A1-suppressing MC38 cells, which have ATP6V0A1 levels comparable to those of normal colon cells. As shown in Fig. 4F, ATP6V0A1 overexpression enhances TGF- β 1 expression in ATP6V0A1-suppressing MC38 cells. Moreover, the reduced TGF- β 1 level by *Atp6v0a1* knockdown is restored by *Atp6v0a1* overexpression, suggesting ATP6V0A1 interference alter TGF- β 1 level by targeting ATP6V0A1 specifically.

Q11: *In Fig.4, authors should knockdown CYP46A1 and 24-OHC respectively so as to demonstrate that they are all required for regulating Tgf β 1 expression in tumor cells.*

•Response:

Thanks for the constructive suggestion. In Fig. 5G (corresponding to Fig. 4g in the previous version of the manuscript), treatment with 24-OHC, an oxidized derivative of cholesterol, can restore the TGF- β 1 expression reduced by ATP6V0A1 depletion, suggesting that ATP6V0A1 induce TGF- β 1 expression via 24-OHC production. Importantly, we conducted additional experiments to investigate the role of CYP46A1 in TGF- β 1 regulation by ATP6V0A1. CYP46A1 is an enzyme catalyzing the oxidation of cholesterol to produce 24-OHC in ER. CYP46A1 knockdown by CYP46A1-targeted siRNAs reduces the production of 24-OHC (**Supplementary Fig. 14B**). Moreover, as shown in **Fig. 5 I**, the recovery of ER-cholesterol by LDL treatment can restore the TGF- β 1 expression reduced by ATP6V0A1 depletion, and CYP46A1 knockdown eliminates this restoration, suggesting the important roles of CYP46A1-mediated 24-OHC production in the regulation of TGF- β 1 expression by ATP6V0A1 (Pls note that Fig. 5 is corresponding to Fig.4 in the previous version of the manuscript).

Q12: *What may happen to 24-OHC and ER-cholesterol levels in Atp6v0a1-*

overexpressing tumor cells?

•**Response:**

Thanks for the constructive suggestion. To more accurately study the impact of elevated ATP6V0A1 levels on the 24-OHC production of CRC cells, ATP6V0A1 was overexpressed in ATP6V0A1-suppressing MC38 cells or HCT-8 cells, which have ATP6V0A1 levels comparable to those of normal colon cells. As shown in **Fig. 5E**, ATP6V0A1 overexpression enhances 24-OHC production in both ATP6V0A1-suppressing MC38 and HCT-8 cells.

Q13: LXR is comprised of LXR-alpha and LXR-beta. Which of them are involved in 24-OHC-LXR mediated Tgfβ1 expression in tumor cells?

•**Response:**

Thanks for the constructive comment. As shown in **Supplementary Fig 14 A**, downregulation of Nr1h3 (LXRα) or Nr1h2 (LXRβ) inhibited the effects of 24-OHC on increasing TGF-β1 levels. Moreover, combining deficiencies in LXRα and LXRβ yielded more effective results in counteracting the effects of 24-OHC compared to having only one of the deficiencies. These data suggested that both LXRα and LXRβ are involved in the process of ATP6V0A1 inducing TGF-β1 expression via 24-OHC production.

Q14: In Fig.5, authors should knockdown Rab7a so as to demonstrate that it is required for regulating Tgfβ1 expression in tumor cells.

•**Response:**

In accordance to reviewer's suggestion, we added additional experiment in which *Rab7* or *RAB7A* was separately knocked down in MC38 and HCT-8 cells. Knockdown of *Rab7* or *RAB7A* in MC38 or HCT-8 cells counteracted the decrease in both 24-OHC (**Fig. 7G**) and TGF-β1 (**Fig. 7H**) levels induced by ATP6V0A1 depletion, with a more pronounced effect in MC38 cells. These data determined the important role of *RAB7A* in ATP6V0A1-regulated TGF-β1 expression.

Q15: What may happen to RABGEF-1 and Rab7a expression in Atp6v0a1-overexpressing tumor cells?

•**Response:**

Thanks for the suggestion. RABGEF-1 and *Rab7a* are in the same pathway (endosome maturation and vesicle traffic). RABGEF-1 increases the *Rab7a* level in endosomes by promoting the recruitment of *Rab7a* to endosomes, thus supporting endosome maturation (**Journal of Cell Science 2015, 128: 4126-4137**). In fact, we confirmed the important roles of RABGEF-1 in regulating

Rab7a+ vesicle levels (**Supplementary Fig. 19A-B**). Moreover, the data supplemented in the revised manuscript shows that the RABGEF1 level is increased by ATP6V0A1 overexpression in MC38 and CT26 cells (**Fig. 6D and Supplementary 20A, C**).

Q16: *Some of the WB analysis do not seem to support authors' conclusion. Throughout the manuscript, authors should quantify the signal intensity of bands.*

•**Response:**

Thanks for the suggestion. Quantitative evaluation has been added for the Western blotting data in the functional experiments throughout the manuscript (Supplementary Fig. 10A-B; Supplementary Fig. 13; Supplementary Fig. 15B; Supplementary Fig. 16; Supplementary Fig. 20B-C; Fig. 10B; Supplementary Fig. 25B, D, F).

Q17: *Genetic background sometimes interferes with immune responses in mice. Is NOD/SCID mice established with B6J background? If not, authors should do so, or use immunodeficient mice such as Rag1^{-/-}, Rag2^{-/-}, or Il2rgRag2^{-/-} mice with B6J background. This also applies to the comparison between NOD/SCID and BALB/c mice.*

•**Response:**

Thanks for the constructive suggestion. NOD/SCID mice are not established with a B6J background. Therefore, we added C57BL/6 Rag2^{-/-}Il2rg^{-/-} mice as another immunodeficient mice model for the comparison between C57BL/6 and immunodeficient mice (**Fig. 2E-H**) and added BRG mice as another immunodeficient mice model for the comparison between BALB/c and immunodeficient mice (**Supplementary Fig. 7B-D**). Similar to what is observed in NOD/SCID mice, ATP6V0A1 suppression does not significantly alter the tumor growth in Rag2^{-/-}Il2rg^{-/-} mice for the MC38 tumor model, and only slightly retards the tumor growth in BRG mice for the CT26 tumor model.

Q18: *Some of anti-CD8 mAb such as 53-6.7 downregulates the expression of CD8 but unable to deplete CD8+ cells. The authors should provide the clone name for anti-CD8 (Lyt2.1) mAb.*

•**Response:**

In accordance to reviewer's suggestion, the clone name of the anti-CD8 mAb used to deplete CD8+ T cells in the present study is 2.43. The anti-CD8 mAb clone 2.43 exhibits depleting activity when used *in vivo*, according to the user manual. We have corrected the Ab information and provided the clone number

of the Ab in the main text (**Line 2 on Page 11**) and Figure legend of Fig. 3 (**Line 3 on Page 52**) of the revised manuscript.

Minor points

MC38 are mistyped as MC8 several times; they can be found in page 8, line 129 and page 13, line 251.

•Response:

Thanks to the reviewer for pointing out the mistakes. The typos have been corrected in the revised manuscript (**Line 8 on Page 8; Line 14 on Page 14**).

Reviewer: 2

Q1: However, the rescue experiments are not robust since the effects of ablating TGFbeta in vivo are only partial (Figs 3I-3L). Therefore, although TGFbeta might be relevant in their model, they do not rule out other potential immunosuppressor mechanisms.

•Response:

Thanks to reviewer for the constructive comments. As discussed in the discussion section (**Lines 15-18 on Page 27**), other factors rather than TGF- β 1 would be involved in ATP6V0A1-mediated immune escape since the effects of altering TGF- β 1 *in vivo* are not completely responsible for that of ATP6V0A1 alterations (**Fig. 4J-O; Supplementary Fig.12B-E**). According to our current data, TGF- β 1 could be one of the most important factors in the process of ATP6V0A1-regulated immune evasion. However, due to limited manuscript space, we plan to explore other potential immunosuppressive mechanisms involving ATP6V0A1 in future studies.

Q2: From these experiments, the authors do another (big) conceptual jump based on the data of Fig. 4A. From these, they conclude that cholesterol metabolism is affected by the inactivation of ATP6V0A1 in CRC cells. However, they totally ignore other metabolic pathways, which are even more significantly affected, such as arginine and proline metabolism. Why focus on cholesterol? What is the contribution of the other pathways to the phenotype of ATP6V0A1-deficient cells? This is a very important question because the central role of ATP6V0A1 in organelle biology suggests that it will impact many pathways, which might contribute to its role in the immune system.

•Response:

Thanks for the constructive comments. To better demonstrate the link between ATP6V0A1 and lipid metabolism, we conducted an additional experiment to investigate the role of high-fat diet (HFD) in inducing tumor promotion and immune suppression in control or ATP6V0A1-suppressed MC38 tumors. As

shown in the newly added **Fig. 1 and Supplementary Figs 1-4**, tumor cell-derived ATP6V0A1 is required for enhancing exogenous lipids to induce immune suppression in CRC, suggesting that ATP6V0A1 may suppress anti-tumor immunity via lipid metabolism reprogramming induced by exogenous lipids. Therefore, in the present study, we aim to study the role of ATP6V0A1 regulating CRC immune evasion via lipid metabolism reprogramming. Based on the data of Fig. 5A (corresponding to Fig. 4a in the previous version of the manuscript), a key molecular cluster related to cholesterol metabolism signaling was significantly altered in MC38-shv0a1 cells compared to the control MC38 cells. Cholesterol metabolism is a vital aspect of lipid metabolism associated with regulating TGF- β 1 expression (Cancer Cell 2020, 38: 567-583 e511), and the mechanism underlying the expression of TGF- β 1 induced by cholesterol metabolism remains unclear. Taken together, we investigated whether ATP6V0A1 could regulate the TGF- β 1 expression through reprogramming cholesterol metabolism in the present study. We will figure out other potential mechanisms in the future studies.

Q3 : *Their model that increased 24-OHC levels promotes TGFbeta upregulation through LXR, is very interesting but, again, needs a more thorough analysis and validation. The rescue experiments in vitro are only partial (Fig. 4G) and there is a general lack of a rigorous metabolomic approach to place the 24-OHC/LXR/TGFb pathway in the context of other alterations in the cholesterol (and non-cholesterol) metabolism. Without that broader approach, it is very difficult to be certain that what the authors proposed is what actually accounts for the role of ATP6V0A1 in the control of immunoevasion in CRC.*

●**Response:**

Thanks for the constructive suggestions. In the revised manuscript, we provided a broader approach to determine the role of ER-cholesterol/24-OHC/LXR axis in ATP6V0A1-regulated TGF- β 1 expression. The level of ER-cholesterol is reduced by ATP6V0A1 suppression (**Fig. 5H**), and restoration of ER-cholesterol by LDL supplement recovered the level of 24-OHC and TGF- β 1 reduced by ATP6V0A1 knockdown (**Supplementary Fig. 14B and Fig. 5I**), indicating ATP6V0A1 may regulate TGF- β 1 expression by altering the level of ER-cholesterol. Importantly, cholesterol depletion by M β CD treatment blocked the role of LDL in restoring 24-OHC production and TGF- β 1 expression in ATP6V0A1-suppressing cells. Moreover, inhibiting 24-OHC production by CYP46A1 knockdown eliminated the restoration of TGF- β 1 expression induced by LDL treatment in ATP6V0A1-suppressing cells (**Supplementary Fig. 14B and Fig. 5I**). On the other hand, the 24-OHC supplement can restore the reduced TGF- β 1 level by ATP6V0A1 suppression. These data showed that ATP6V0A1 promotes TGF- β 1 expression via ER-cholesterol/24-OHC axis. 24-OHC is a classic endogenous agonist of LXR, we thus further explored whether ATP6V0A1 promotes TGF- β 1 expression by activating LXR. As shown in **Fig.**

5F, suppression of ATP6V0A1 significantly attenuates the activities of LXR in CRC cells, while the 24-OHC supplement can restore them. Importantly, the inhibition of LXR by an inhibitor (GSK2033) or by Nr1h3 (LXR α) or/and Nr1h2 (LXR β) knockdown can overcome the restoration of TGF- β 1 expression induced by the 24-OHC supplement in ATP6V0A1-suppressing cells (**Fig. 5G and Supplementary Fig. 14A**), suggesting that LXR activities are important to ATP6V0A1-induced TGF- β 1 expression via 24-OHC production. Collectively, all these data could be sufficient to determine the important roles of the ER-cholesterol/24-OHC/LXR axis in ATP6V0A1-regulated TGF- β 1 expression.

***Q4 :** In the same vein, the focus on RABGEF1 is also surprisingly underdeveloped from the mechanistic point of view. Much more rigorous mechanistic studies are needed to convincingly demonstrate that the model proposed here actually accounts for the effect of perturbing ATP6V0A1.*

●**Response:**

Thanks for the constructive suggestions. In the revised manuscript, we provided more evidence to support the important role of RABGEF1-dependent endosome maturation in ATP6V0A1-regulated TGF- β 1 expression. The level of RABGEF1 is reduced by ATP6V0A1 suppression (**Fig. 6A-C**), and the recovery of RABGEF1 expression restores the level of 24-OHC and TGF- β 1 reduced by ATP6V0A1 knockdown (**Fig. 7C-D**). On the other hand, RABGEF1 suppression attenuates the reduction of ER-cholesterol and TGF- β 1 expression by ATP6V0A1 depletion (**Fig. 7E-F**). These data showed that ATP6V0A1 promotes TGF- β 1 expression via RABGEF1 upregulation. Moreover, RABGEF1 is required for ATP6V0A1-induced endosome maturation (**Fig 6E-F and Supplementary Fig 19A-B**). Thus, we further study the role of endosome maturation in ATP6V0A1-regulated TGF- β 1 expression. As shown in **Fig. 7G-H**, inhibition of endosome maturation by knockdown of Rab7/RAB7A in MC38 /HCT-8 cells counteracted the decrease in both 24-OHC and TGF- β 1 levels induced by ATP6V0A1 suppression, suggesting that ATP6V0A1 regulates TGF- β 1 expression via RABGEF1-dependent endosome maturation. Regarding the mechanism in which ATP6V0A1-induced endosome maturation mediates TGF- β 1 expression via ER-cholesterol/24-OHC axis, we proposed that ATP6V0A1 promotes the absorption of exogenous cholesterol via endosome maturation to enhance ER-cholesterol and 24-OHC production, thereby upregulating TGF- β 1 expression. Indeed, ATP6V0A1 promotes cholesterol absorption (**Fig. 6G-J, Supplementary Fig. 17E-F and Supplementary Fig. 18A-D**), and RABGEF1-dependent endosome maturation is required for cholesterol absorption (**Supplementary Fig 19A-B**). Similar to ATP6V0A1 suppression, the blockade of cholesterol absorption by RABGEF1 depletion or by treatment with U18666A (an inhibitor of cholesterol transport from lysosome to ER) suppresses TGF- β 1 upregulation, which can be rescued by supplementation with exogenous LDL or 24-OHC (**Fig. 7B and Supplementary Fig 19C-D**). Moreover, the blockade

of cholesterol absorption by RABGEF1 depletion or by treatment with Baf-A1 (an inhibitor of vesicular transport) also counteract the decrease in ER-cholesterol or 24-OHC level, and TGF- β 1 expression induced by ATP6V0A1 suppression (**Fig. 7E-F and Supplementary Fig. 26E**). These data suggested that cholesterol absorption could be involved in ATP6V0A1-regulated TGF- β 1 expression through RABGEF1-dependent endosome maturation.

Q5: *In addition, the study mostly uses cell lines that might or might not represent what CRC actually is. For example, MC38 is considered to have some “similarity” to the CMS1 (MSI) type of CRC, and, therefore, is representative of tumors that are “hot” from an immunological point of view. This should be considered when focusing on the concept in this paper that ATP6V0A1 is upregulated in CRC. How do the authors reconcile their data with the fact that MC38 cells are actually quite active from the immunosurveillance point of view? In which type of CRC is ATP6V0A1 really relevant? In this regard, the analysis of the human CRC specimens is partial and does not take into account a more rigorous understanding of the biology and pathology of CRC and the more contemporary classification in the field. In connection with this, the utilization of organoids with defined genomic and transcriptomics profiles better representing the different types of CRC is a must, as it is also the utilization of PDOs with also defined genetic profiles.*

● **Response:**

Thanks for the insightful comments. It is worth noting whether the roles of ATP6V0A1 regulating immune evasion through the RABGEF1/24-OHC/TGF- β 1 signaling are limited to the CRC with deficient mismatch repair (dMMR) or high microsatellite instability (MSI-H), as this type of CRC accounts for only about 15% of the total CRC. By analyzing the CRC-TMA data with mismatch repair information (**Fig. 8A-B**), we found that ATP6V0A1 is overexpressed in both CRC patients with deficient mismatch repair (dMMR; n=20) and with proficient mismatch repair (pMMR; n=60). Moreover, according to previous reports (*Sci Rep* 2022, 12: 10999; *Cancer communications* 2023, 43(4): 435-454), we used MC38 cells as a MSI/dMMR CRC model and CT26 cells as a MSS/pMMR CRC model in the present study. As shown in **Figs. 5-7 and Supplementary Fig. 20**, ATP6V0A1 drives ER-cholesterol-induced enhancement of TGF- β 1 through RABGEF1-dependent endosome maturation in both CRC cells with dMMR (MC38) and pMMR (CT26). Importantly, the interference of ATP6V0A1 has been shown to suppress tumor growth and restore anti-tumor immunity in both dMMR and pMMR murine CRCs (**Fig. 2 and Supplementary Fig. 7**). Therefore, our findings in this study may be applicable to both dMMR and pMMR CRCs. As immune checkpoint blockade for pMMR CRCs encounters limitations and challenges, our study may offer promising strategies for enhancing immunotherapy in this context.

Q6: Finally, the pharmacological approach using repurposed drugs needs many controls to ensure that the effects reported here are actually specific to cholesterol metabolism.

●**Response:**

Thanks for the constructive suggestions. In the revised manuscript, we conducted additional experiments to determine the targeting specificity of Dac in CRC treatment. As shown in newly added **Fig. 10**, depletion of ATP6V0A1 diminishes the ability of Dac to reduce the levels of 24-OHC and TGF- β 1 in CRC cells, thereby impeding its capacity to suppress CRC growth and restore anti-tumor activity in *in vivo* models. Similar to the effects of ATP6V0A1 interference on restoring anti-tumor immunity through reprogramming cholesterol metabolism in CRC cells, Dac also reduces RABGEF1 expression and 24-OHC production by targeting ATP6V0A1 specifically. These data suggested that Dac exerts therapeutic effectiveness in CRC via the specific inhibition of ATP6V0A1, which regulates anti-tumor immunity through cholesterol metabolism reprogramming.

Reviewer: 3

Q1: The analysis of patient data is correlative and sometimes difficult to interpret (Fig. S1 – 4). It is unclear what ‘high lipid metabolism’ means. For example, fatty acid degradation and synthesis seem to be upregulated; terms such as primary bile acid and cutin/suberin/wax synthesis are included – what does this mean in the context of CRC?

●**Response:**

Thanks for the comments. Previous study has reported that high levels of exogenous lipids caused by a high-fat diet (HFD) can increase the overall level of lipid metabolism in tumors (**Cell. 2020 Dec 23;183(7):1848-1866**). Therefore, the overall level of lipid metabolism in tumor tissues could be associated with lipid levels in the tumor microenvironment (TME). According to the KEGG database, the overall lipid metabolism pathways, including fatty acid metabolism and cholesterol metabolism, such as primary bile acid synthesis, are used in this study. In addition to showing positive association of ATP6V0A1 expression with high lipid metabolism and decreased immune activity (**Supplementary Figs. 1 and 2**), we also found that ATP6V0A1 expression was positively associated with decreased immune activity only in the CRCs with high lipid metabolism. No correlation was observed in the CRCs with low lipid metabolism (**Supplementary Fig. 3**). Moreover, in the revised manuscript, we added new data to show that high lipid metabolism was positively associated with decreased immune activity only in the CRCs with high ATP6V0A1 expression, but not in those with low ATP6V0A1 expression (**Supplementary Fig. 4**). These data suggest that high level of lipid metabolism is important to

ATP6V0A1-suppressed anti-tumor immunity and high level of ATP6V0A1 is required to reversely associate lipid metabolism level with immune activity. By showing these bioinformatics data, we aimed to make a preliminary indication that ATP6V0A1 plays important roles in exogenous lipid-induced CRC immune suppression. To test this hypothesis, we conducted an additional experiment to investigate the roles of HFD-inducing tumor promotion and immune suppression in control or ATP6V0A1-suppressing MC38 tumors. As shown in the newly added **Fig. 1**, HFD promoted tumor growth and suppressed anti-tumor immunity in control MC38 tumors, and ATP6V0A1 depletion in tumor cells blocked these changes induced by HFD. These data determined the important roles of ATP6V0A1 in exogenous lipid suppressed anti-tumor immunity of CRCs. We thus aimed to investigate the role of ATP6V0A1 in regulating anti-tumor immunity through exogenous lipids.

Q2: *The impact of VoA1 knockdown on tumor growth in immune-deficient mice is intriguing. However, one would normally expect the use of multiple control/target shRNAs, at least in key experiments. This is important to exclude off-target effects, and because the depletion of VoA1 sometimes appears to be rather modest (Fig. S6).*

●**Response:**

We totally agree with the reviewer's suggestion to exclude the off-target effects of shRNAs in the key experiments. In fact, for the key *in vivo* experiments, we used two *Atp6v0a1*-targeted shRNAs with different sequences. In the revised manuscript, we have supplemented the data with another *Atp6v0a1*-targeted shRNA (**named shv0a1-2**) in the experiments determining the roles of ATP6V0A1 in promoting MC38-tumor growth in the immunocompetent and immunodeficient mice (**Fig. 2A-H**), as well as in suppressing memory CD8⁺ T-cell activities (**Fig. 3F**). The sequence of shv0a1-2 was provided in **Supplementary Table 3**. As shown in **Fig. 2A-H** and **Fig. 3F**, both *Atp6v0a1* knockdowns with shv0a1-1 and shv0a1-2 significantly suppress MC38 tumor growth in the immunocompetent mice rather than in the immunodeficient mice and increase the activities of CD44⁺CD8⁺ T cells infiltrated into MC38 tumors. Moreover, in the key mechanism studies *in vitro*, *Atpv0a1* was overexpressed in the *Atp6v0a1*-suppressing MC38 or CT26 cells with shv0a1-1 (**Fig. 4F, 5E, and Supplementary 20D-E**), and the results showed that the recovery of ATP6V0A1 expression can restore both the 24-OHC production and TGF-β1 expression reduced by ATP6V0A1 suppression. ATP6V0A1 was also overexpressed in the ATP6V0A1-suppressing HCT-8 cells with shV0A1, and similar results were obtained with the human cell model (**Fig. 5E**). Collectively, these data suggest the key functional phenotypes and mechanisms by interfering with ATP6V0A1 expression shown in this study are mainly attributable to specific effects (Pls Note: shv0a1 in the present manuscript means shv0a1-1 unless indicated otherwise).

Q3: *The authors provide evidence that VoA1 is important for lysosomal cholesterol mobilization from LDL, which affects ER cholesterol levels, 24-OHC production and TGF- β signaling. This conclusion could be strengthened by demonstrating that removing extracellular lipids has similar consequences.*

●**Response:**

In accordance to reviewer's suggestion, exogenous LDL was used to treat HCT-8 cells in the culture medium containing lipid-depleted fetal bovine serum. As shown in **Fig. 7A**, LDL treatment increases TGF- β 1 expression, which can be blocked by ATP6V0A1 depletion, suggesting that ATP6V0A1 is important for LDL inducing TGF- β 1 signaling. Importantly, the enhancement of 24-OHC production and TGF- β 1 expression induced by LDL treatment is blocked by M β CD treatment, which removes cholesterol from cells (**Supplementary Fig. 14B and Fig. 5I**). Moreover, blockade of 24-OHC production via CYP46A1 knockdown also counteracts the increase in TGF- β 1 levels induced by LDL treatment (**Supplementary Fig. 14B and Fig. 5I**). Collectively, these data strengthen the evidence that V0A1 is important for lysosomal cholesterol mobilization from LDL.

Q4 : *While the role of V-ATPase-mediated endosomal acidification in endosomal trafficking is well established, the authors suggest that VoA1 depletion blocks lysosomal LDL transport by decreasing Rabgef1. Is this a VoA1-specific effect, or mediated by the V-ATPase complex? Is proton-pumping activity of VoA1-containing V-ATPases required? Does depletion of other VoA subunits have similar effects? Does VoA1 deficiency have any effect on vesicle acidification?*

●**Response:**

Thanks for the valuable comments. It is worth noting whether the regulation of Rabgef1 and TGF- β 1 signaling is induced by the V0A1-specific effect or the V-ATPase complex. As described in the discussion section (**Page 26**), the functions of ATP6V0A1 in regulating RABGEF1-dependent cholesterol absorption and TGF- β 1 expression may not necessarily rely on the overall levels of the V-ATPase complex or its V0 subcomplex. Upon the knockdown of ATP6V0A1, ATP6V0A3 (translated by TCIRG1) protein level was upregulated while ATP6V0A2 expression was not significantly changed (**Supplementary Fig. 25A**). ATP6V0A4 is relatively lowly expressed in both murine (**Supplementary Fig. 25A**) and human (**Supplementary Fig. 1B**) CRCs. Moreover, ATP6V0A1 suppression did not alter the protein levels of ATP6V0C and ATP6V1A (**Supplementary Fig. 25A**), which are associated with the V0 and V1 subunits of the V-ATPase complex, respectively. Therefore, the changes in ATP6V0A1 expression may not alter the total levels of the V-ATPase complex. Importantly, the suppression of other subtypes of the V0A subunit, such as

ATP6V0A2 and ATP6V0A3, did not significantly affect the levels of both RABGEF1 and TGF- β 1 (Supplementary Fig. 25C-F), suggesting that ATP6V0A1 may have distinct functions in regulating immune evasion compared to the other V0A subtypes.

Additionally, V-ATPases play vital roles in regulating cellular and vesicular pH, and the V-ATPase complexes with different V0A subtypes would exert different functions in regulating pH (**Journal of Biological Chemistry 2009, 284: 16400-16408**). Indeed, ATP6V0A1 suppression did not alter the cellular pH (Supplementary Fig. 26A), but enhanced the pH in vesicles (Supplementary Fig. 26B, C). Interestingly, ATP6V0A1 preferred to regulate the pH of RAB7+ vesicles rather than RAB7- vesicles (Supplementary Fig. 26B, D). We cannot exclude the possibility that the pH regulation of RAB7+ vesicles may be involved in ATP6V0A1-mediated cholesterol absorption and TGF- β 1 expression in the present study. In fact, ATP6V0A1 suppression can reduce 24-OHC production and TGF- β 1 expression in the HCT-8 cells treated with control vehicles but not in those treated with Baf-A1, an inhibitor of V-ATPase activity (**Supplementary Fig. 26E**), indicating that the acidification function of V-ATPase may involve in the pathway of ATP6V0A1 regulating 24-OHC and TGF- β 1. However, it is noteworthy that endosome acidification could be the result of endosome maturation (**J Cell Sci 2014, 127: 2697-2708**). While promoting endosome maturation, the RABGEF1 pathway can recruit VPS41 and RILP to endosomes and thus strengthen the linkage of V0-subcomplex and V1-subcomplex via the interaction of RILP and ATP6V1G1 (**J Cell Sci 2014, 127: 2697-2708**). Therefore, the process of ATP6V0A1 regulating endosome maturation via a RABGEF1-dependent pathway may not be dependent on the acidification activity of V-ATPases. We plan to further investigate the crosstalk between the RABGEF1 pathway, endosome maturation, and vesicle pH in the context of ATP6V0A1-induced cholesterol absorption.

Q5: *The data on Daclatasvir as a potential VoA1 inhibitor are not convincing. How does Dac binding affect VoA1 – does it suppress proton pumping? Does Dac alter the thermal stability of other V-ATPase subunits, in particular other VoA subunits? The authors show that Dac reduces Tgf- β levels in cells expressing VoA1 shRNA (depletion of the protein is not shown in the blot), and conclude that Dac acts on VoA1. This is a very indirect readout of VoA1 function – a more direct demonstration of VoA1 inhibition would be essential. Moreover, it would be important to demonstrate that Dac does not affect the growth of VoA1-depleted tumors.*

●**Response:**

Thanks for the constructive comments and suggestions. Firstly, Dac did not significantly bind to other V0A subunits (**Supplementary Fig. 23G**), suggesting the specificity of Dac binding to ATP6V0A1. Moreover, we conducted an additional experiment to investigate the role of Dac in treating ATP6V0A1-

depleted tumors in the revised manuscript. As shown in newly added **Fig. 10**, Dac treatment suppressed the tumor growth (Fig. 10E-G) and restored the activation of CD44⁺CD8⁺ T cells (Fig. 10H, I) in control MC38 tumors, but not in ATP6V0A1-suppressing MC38 tumors. These findings demonstrated that Dac enables the effective suppression of CRC by specifically targeting ATP6V0A1. Additionally, Fig. 10 A-D shows that Dac treatment does not significantly change the protein level of ATP6V0A1 but has similar effects with ATP6V0A1 knockdown on altering RABGEF1/24-OHC/TGF- β 1 signaling. These results suggest that Dac binding may have effects by inhibiting the functions of ATP6V0A1 protein. We will further investigate the mechanisms of how Dac binding affects the role of ATP6V0A1 in regulating immune evasion in future studies.

Reviewer: 4

General comments

Q1: *In supplementary fig.1 to supplementary fig.3, the authors divide TCGA dataset into lipid metabolism high group and low group, and ATP6V0A1 was selected among the V-ATPase subunits highly expressed in the lipid metabolism high group as the most inversely correlated with immune score. However, several subunits other than ATP6V0A1, for instance ATP6V0C, ATP6V0D1 and ATP6V0E1, show opposite trend in the immune score despite their increased expression in the lipid metabolism high group. Moreover, although ATP6V0A1 in CRCs mainly affects CD8⁺ T cell function in this study, all immune cell populations are considered as the immune score. Therefore, it is necessary to clarify how ATP6V0A1 differs in expression and function compared to other V-ATPase subunits in colorectal cancer cells, and why ATP6V0A1 is more important in lipid-mediated immunosuppression than the other subunits.*

●Response:

Thanks for the insightful comments. (1) Based on the analysis in Supplementary Figs.1-3, we aimed to select a V-ATPase subunit that is important to exogenous lipid-induced immune evasion. Previous study has reported that high levels of exogenous lipids caused by high-fat diet (HFD) can increase the overall level of lipid metabolism in tumors (**Cell. 2020 Dec 23;183(7):1848-1866**). Therefore, we investigated the potential correlation between the levels of different V-ATPase subunits, overall lipid metabolism score, and immune score. Among the 11 V-ATPase subunits correlated with high lipid metabolism, ATP6V0A1 showed the strongest inverse correlation with immune activity (**Supplementary Figs. 1-2**). Moreover, high lipid metabolism was positively associated with decreased immune activity only in the CRCs with

high ATP6V0A1 expression, but not in those with low ATP6V0A1 expression (**Supplementary Fig. 4**). More importantly, as shown in newly added **Fig. 1**, high-fat diet promoted tumor growth and suppressed anti-tumor immunity in control MC38 tumors, and ATP6V0A1 depletion in tumor cells blocked these changes induced by HFD. These data suggest the important roles of ATP6V0A1 in the exogenous lipid-suppressed anti-tumor immunity of CRCs. We thus aimed to investigate the role of ATP6V0A1 in regulating anti-tumor immunity through exogenous lipids in the present study. Different V-ATPase subunits may play different roles in regulating immune escape. ATP6V0C, ATP6V0D1 and ATP6V0E1 are positively correlated with immune activity despite their increased expression in the high lipid metabolism group, indicating that exogenous lipids may not regulate immune evasion by upregulating these subunits. Moreover, ATP6V0A1 showed the strongest inverse correlation with immune activity among the V-ATPase subunits. Thus, we primarily focused on the roles of ATP6V0A1 in regulating anti-tumor immunity through lipid metabolism in the present study. We will investigate the expression of other subunits and their function in regulating anti-tumor immunity in CRC in the future. The regulating network in CRC cells for the expression of different V-ATPase subunits, including V0A1, also needs to be explored in the future. (2) ATP6V0A1 may have distinct functions in regulating RABGEF1 signaling and TGF- β 1-mediated immune suppression compared to other subunits, since the suppression of other subunits, such as ATP6V0A2 and ATP6V0A3, did not significantly affect the levels of both RABGEF1 and TGF- β 1 (**Supplementary Fig. 25C-F**). RABGEF1 signaling is important for exogenous lipid-induced immune suppression. Therefore, the difference in regulating RABGEF1 signaling between ATP6V0A1 and other subunits may explain why ATP6V0A1 is more important in lipid-mediated immunosuppression than the other subunits. We will further figure out this issue in the future. (3) During the initial stage of our present study, the mechanism by which the V-ATPase subunits regulate immunity remained unknown. Therefore, in Supplementary Figs 1-4, the overall immune score based on the analysis of different immune cells was used to indicate the immune activities within TME.

Q2: *In addition, the flowcytometry (FCM) analysis in Figure 2 shows not including live dead, CD45, or CD3 staining, therefore, target cells are not properly gated and the reliability of the subsequent analysis cannot be ensured.*

●**Response:**

Thanks for the suggestion. In fact, in the previous version of the manuscript, we used the FCAS strategy shown in Fig. 2e to analyze the data of Fig. 2g and Fig. 2h, and used another strategy including FVS510 (Fixable Viability Stain 510; BD Biosciences) live/dead and CD45 staining for the other analyses of tumor-infiltrated CD8⁺ T cells. The later strategy was shown in Fig. 3D in the revised manuscript, and used to repeat the analyses in Fig. 2g and Fig. 2h of the

previous version of the manuscript. The new data are shown in Fig. 3F and Fig. 3G, respectively. A similar conclusion can be drawn from these new data.

Specific points

Q1: *Quantitative evaluation should be added in all Western blotting.*

●**Response:**

Thanks for the suggestion. Quantitative evaluation has been added for the Western blotting data in the functional experiments throughout the manuscript (Pls see the quantitative data in the revised Supplementary Fig. 10A-B, Supplementary Fig. 13, Supplementary Fig. 15B, Supplementary Fig. 16, Supplementary Fig. 20B-C, Fig. 10B, and Supplementary Fig. 25B, D, F).

Q2: *Numbers on the Y-axis should start from zero (Fig.3e, 3f, 4d, 4e, 4f, 4g, 6a, 6b, 6c etc).*

●**Response:**

In accordance to reviewer's suggestion, the Y-axis numbers have been adjusted to begin from zero in Fig. 4E, 4G, 5D, 5H, S14B, 5G, 7B, 7C, and S19D. These figures correspond to Figs 3e, 3f, 4d, 4e, 4f, 4g, 6a, 6b, and 6c in the previous version of the manuscript.

Q3: *In supplementary fig.4, bulk MC38 tumors include a lot of tumor-infiltrated immune cells and stroma cells. The increased expression of ATP6V0A1 by HFD is not necessarily derived from tumor cells.*

●**Response:**

Thanks for the comment. The data in Fig. S4 of the previous version was analyzed based on the RNA sequencing data from the GSE157994 database. As described in the database (<https://www.ncbi.nlm.nih.gov/geo/query/acc.cgi>; **Accession NO: GSE157994**), the RNA sequencing data we used is analyzed in the GFP-labeled MC38 cells isolated from MC38 tumors. In addition, we provided key experimental data to show that ATP6V0A1 is required for HFD-induced immunosuppression in CRC (**Fig. 1**). These data directly support our hypothesis that ATP6V0A1 is crucial for exogenous lipid-induced immune evasion. To keep concise data presentation in our manuscript, we thus removed the data corresponding to Fig. S4 from the previous version.

Q4: *In supplementary fig.5, the authors evaluate the expression of ATP6V0A1 by immunohistochemistry (IHC) using specimens from CRC patients. Is high expression of ATP6V0A1 in IHC staining associated with low CD8+T cell infiltration?*

●**Response:**

Thanks for the question. Although the CD8⁺T cell infiltration seems to be obviously increased in MC38 cells by *Atp6v0a1* knockdown as shown in the scRNA sequencing data, the FCM data with higher accuracy shows that CD8⁺T cells increase only very slightly (~1.1 folds) in ATP6V0A1-suppressing tumors (**Reply letter Figure 4A**). Moreover, ATP6V0A1 interference also does not significantly change the percentage of memory CD8⁺T cells in MC38 tumors (**Reply letter Figure 4B**). The evaluation of comprehensive scRNA sequencing and flow cytometry analysis suggested that tumor cell-derived ATP6V0A1 mainly regulates immune evasion by suppressing the activities of memory CD8⁺T cells (**Supplementary Fig. 9E-F and Fig. 3; Reply letter Figure 4A-B**) rather than suppressing the infiltration of CD8⁺T cells. Therefore, we did not analyze the correlation of ATP6V0A1 expression with CD8⁺ T cell infiltration. Moreover, due to the small size of the tissue section, the tissue microarray (TMA) is not suitable for evaluating the activity of CD8⁺T cells in the TME. In the revised manuscript, we conducted additional experiment to analyze the correlation between ATP6V0A1 expression and the effectiveness of memory CD8⁺ T cells using paraffin-embedded tumor tissue sections from CRC patients. (**Supplementary Fig. 22 and Fig. 8**). The results showed that ATP6V0A1 is inversely correlated with the effectiveness of memory CD8⁺ T cell.

Q5: In supplementary fig.10, single cell RNA seq is powerful to identify the specific immune cell subset. However, the authors should be careful when using scRNAseq for quantitative evaluation. FCM is advantageous for quantitative evaluation, which also available for evaluation at protein level. If there is a marker that characterizes memory like T-2 subset, it is necessary to confirm these cells by FCM or IHC and whether the number of T cells expressing that marker is increasing in the tumor.

●**Response:**

Thanks for the constructive comments and suggestions. As described in the manuscript (**Line10-18 in Page 10**), memory like T-2 subset had higher effectiveness than the other memory like T subset (**Supplementary Fig. 8D**). While memory like T-2 subset was increased in ATP6V0A1-suppressing tumors, the total memory-like T cells was not significantly altered (**Supplementary Fig. 9B**). Moreover, suppression of tumor cell-derived ATP6V0A1 significantly increased the expression levels of effective factors in all memory-like T cells (**Supplementary Fig. 9F**). These data suggested that ATP6V0A1 suppresses the effectiveness of memory CD8⁺ T cells. CD44 is a classic marker of murine memory CD8⁺ T cells, whereas CD45RO is a classic marker of human memory CD8⁺ T cells. We thus used FCM assay to validate the involvement of ATP6V0A1 in suppressing the effectiveness of CD44⁺CD8⁺ T cells in murine CRC model (**Fig. 3F, G, and I**), as well as CD45RO⁺CD8⁺ T cells in human CRC model (**Fig. 3H**). Consistent with the scRNA seq data, the FCM analysis

showed that ATP6V0A1 depletion increases the effectiveness of memory CD8⁺ T cells. Moreover, to provide more reliable clinical evidence supporting the aforementioned conclusion, we used IF assay to investigate the correlation between ATP6V0A1 expression and CD45RO⁺CD8⁺ T cell effectiveness based on the paraffin-embedded tumor sections from CRC patients (**Supplementary Fig. 22 and Fig. 8**). Consistent with the scRNA seq analysis of human specimens (**Supplementary Fig. 21**), the IF data showed that ATP6V0A1 is inversely correlated with memory CD8⁺ T cell effectiveness.

Q6: *Representative FCM images are necessary for Fig.2i and Fig.3f.*

●Response:

Thanks for the suggestion. The data of newly revised Fig. 3E (corresponding to Fig. 2i in the previous manuscript) was analyzed on the tumor infiltrated CD8⁺T cells, which is similar to that of Fig. 3F, 3G, 3H, and 3I. To keep concise data presentation in our manuscript, the analysis strategy for Fig. 3E is integrated into Fig. 3D. Moreover, the representative FCM images for Fig.4G (corresponding to Fig. 3f in the previous manuscript) were provided as Supplementary Fig. S11.

Q7: *In Fig.3b, CD8+ cells were isolated from tumors. Their purity should be shown.*

●Response:

Thanks for the suggestion. The CD8⁺ T cells used for the analysis in newly revised Fig. 4B (corresponding to Fig. 3b in the previous version) were sorted from tumors with a FACS method, and the population selected for sorting has been shown in revised Fig. 4B.

REVIEWER COMMENTS

Reviewer #1 (Remarks to the Author):

The authors have provided new experimental results that support their overall concept. However, there are still several concerns that need to be addressed.

1) Response to question #3 is still confusing. Are patient groups shown Fig. 8D and Fig. 8E different? More than 80 % of patients with ATPV0A1<high> group survived after 60 month independent of CTL activities, while only 40% of patients with ATPV0A1<high> group survived after 60 months. If "ATP6V0A1 protein levels were inversely correlated with the overall survival of CRC patients (line 366 in new version of manuscript)", ATP6V0A1<low> CRCs should have better survival than ATP6V0A1<high> CRCs, even with insufficient infiltration of CTL, but this does not seem to be the case. Why? And why does CTL invasion not correlate with ATP6V0A1 expression in tumor cells?

2) Response to question #9 is not sufficient. Authors should knockdown Tgfβ1 in tumor cells and investigate their growth in vitro/in vivo.

3) Experimental results should be shown in the results section in general. It is thus suggested that Supplementary Figs. 25 and 26 should be described in the results section.

4) In the previous version of manuscript, the authors did not show factor(s) that potentially upregulate ATP6V0A1 in tumor cells. If the authors have any thoughts on this issue, they should address that in the discussion section.

5) In Fig. 2E, the authors calculate the significance between shNTC and shv0a1-1 and that between Shv01-1 and shv01-2 but the latter should be the significance between shNTC and shv0a1-2.

Reviewer #2 (Remarks to the Author):

The authors have responded to my comments and suggestions.

Reviewer #3 (Remarks to the Author):

The reviewers have satisfactorily addressed my comments.

Reviewer #4 (Remarks to the Author):

The additional revise experiments have much improved the content of this study, however, some of the experimental data are still indispensable. A couple of comments (red color) are reinserted in the point-by-point response letter. Please find the attached file.

A point-by-point response to the Reviewers' comments and suggestions

Reviewer: 1

Thank you very much for the positive and constructive comments and suggestions to improve our manuscript. Below is a point-by-point response to the comments and suggestions.

Q1: *Response to question #3 is still confusing. Are patient groups shown Fig. 8D and Fig. 8E different? More than 80 % of patients with ATPV0A1<high> group survived after 60 month independent of CTL activities, while only 40% of patients with ATPV0A1<high> group survived after 60 months. If “ATP6V0A1 protein levels were inversely correlated with the overall survival of CRC patients (line 366 in new version of manuscript)”, ATP6V0A1<low> CRCs should have better survival than ATP6V0A1<high> CRCs, even with insufficient infiltration of CTL, but this does not seem to be the case. Why? And why does CTL invasion not correlate with ATP6V0A1 expression in tumor cells?*

●Response:

Thanks to the reviewer for the insightful comments. The specimens used for Fig. 8D and Fig. 8E are from different patient cohorts, and the analysis in these two figures is based on different methods. Fig. 8D is analyzed by the IHC method on tissue microarray (TMA; Shanghai OUTDO BIOTECH Co., Ltd.), and Fig. 8E is analyzed based on the bulk mRNA sequencing data which are provided in the Tumor Immune Dysfunction and Exclusion (TIDE) analysis platform. By showing Fig. 8E in the previous manuscript, we aimed to show that higher CTL levels improve survival only in ATP6V0A1<low> CRC, while no significant alteration is observed in ATP6V0A1<high> CRC and thus provide the clinical evidence that high levels of ATP6V0A1 may impair the anti-tumor activity of CD8+T cells in CRCs. However, we agree with the reviewer that Fig.8E should provide consistent information with Fig. 8D about the correlation between ATP6V0A1 and patient survival in all samples. The TIDE analysis platform includes the sequencing data from different CRC datasets. These datasets are designed to be analyzed separately. Previous Fig. 8E was analyzed based on the GSE12945 dataset, including 62 CRC samples. In the **new Fig. 8E**, we used another CRC dataset with more CRC samples (GSE38832; 122 samples). Consistent with the previous Fig. 8E, the new Fig. 8E also showed that higher CTL levels improve survival only in ATP6V0A1<low> CRC (HR=0.1573, p<0.0001), while no significant alteration is observed in ATP6V0A1<high> CRC (HR=0.5796, p=0.3423). As shown in the **new Fig. 8E**, the overall survival in ATP6V0A1<low> CRC is similar to that in ATP6V0A1<high> CRC when the cytotoxic T lymphocytes (CTL) is insufficient (CTL bottom) but is better than that in ATP6V0A1<high> CRC when the CTL level is high (CTL Top). Therefore, the overall survival in all ATP6V0A1<low>

CRC could be better than that in all ATP6V0A1<high> CRC. Among different cohorts in TIDE, it is common that a higher CTL level improves survival in ATP6V0A1<low> CRC but not in ATP6V0A1<high> CRC. Still, the correlation of ATP6V0A1 with patient survival differs among various datasets, likely due to the limited number of cases in those datasets. To provide more accurate results for the correlation of ATP6V0A1 mRNA level with patient survival in CRC, we analyze the correlation between ATP6V0A1 mRNA level and overall survival using the TCGA-based Kaplan-Meier Plotter dataset, including 1061 CRC patients. The result shows that the ATP6V0A1 mRNA level is inversely correlative with the overall survival in these 1061 CRC patients (Reply letter Fig. 1). More importantly, we use IHC method to analyze the correlation of ATP6V0A1 protein level with patient survival based on CRC TMA samples, and the result shows the inverse correlation between ATP6V0A1 protein level and patient survival (**Fig. 8D**). These data suggested that ATP6V0A1 levels are inversely correlated with the overall survival of CRC patients.

It is of note that as the TIDE analysis, the CRC samples were divided into ATP6V0A1<high> and ATP6V0A1<low> groups, and then the samples in each group were divided into CTL-sufficient and CTL-insufficient cases with median cutoff. Therefore, the CTL level for CTL-sufficient or CTL-insufficient cases in the ATP6V0A1<high> group could differ from that in the ATP6V0A1<low> group. It could not be very scientific to compare the survival of CTL-sufficient or CTL-insufficient CRC between ATP6V0A1<high> and ATP6V0A1<low> groups. Similarly, the correlation between ATP6V0A1 and CTL level could not be analyzed based on Fig. 8E. The experimental data in this study showed that ATP6V0A1 promotes CRC development mainly by suppressing the effectiveness of memory CD8+T cells (**Figs. 2, 3, 4, 9**). Therefore, we focused on investigating the correlation between ATP6V0A1 and the activity of memory CD8+ T cells, and the result showed the inverse correlation between them (**new Fig. S24 and Fig. 8I**). It is of note that we have increased the sample number in the analysis of Fig. 8H-I to improve the data reliability.

Moreover, if required, Fig. 8E could be removed from the present data or replaced with Reply letter Fig. 1. Fig. 8E aims to provide clinical evidence that high levels of ATP6V0A1 may impair the anti-tumor activity of CD8+T cells in CRCs. However, we believe that removing Fig.8E will not significantly impact the conclusion of this study for the following reasons: (1) This study aims to determine that ATP6V0A1 promotes CRC development by suppressing the effectiveness of memory CD8+T cells. Compared with Fig. 8E, new Fig. S24 and Fig. 8I provided more accurate and direct clinical evidence to support the inverse correlation between ATP6V0A1 and memory CD8+ T-cell effectiveness on the paraffin-embedded tumor sections by using immunofluorescence (IF) assay. In addition, we have shown the positive correlation between memory CD8+ T-cell effectiveness and CRC survival (**new Fig. S23K**). (2) Compared with Fig. 8E, Reply letter Fig. 1 provided more accurate evidence for the correlation between ATP6V0A1 **mRNA** level and CRC survival, as it analyzed

a significantly larger number of samples. In addition, we have provided clinical evidence for the correlation between ATP6V0A1 **protein** level and CRC survival in Fig. 8D.

Huang *et al* Reply letter Figure 1

Q2: Response to question #9 is not sufficient. Authors should knockdown *Tgfb1* in tumor cells and investigate their growth *in vitro/in vivo*.

•**Response:**

Thanks for the kind suggestion. *Tgfb1*/TGFB1 was knocked down in MC38/HCT-8 cells, and the CCK8 assay was used to analyze the effects of *Tgfb1*/TGFB1 on the growth of these cells. As shown in **new Supplementary Fig. 12F-G** and described in **lines 251-252**, the suppression of *Tgfb1*/TGFB1 in MC38/HCT-8 cells does not significantly affect the growth of these cells. It is worth noting that the reduced level of *Tgfb1* by si*Tgfb1* is comparable to that by *Atp6v0a1* knockdown (**Fig. 4E**). Moreover, the data in this study have determined the critical roles of the TGF- β 1/SMAD3 pathway in ATP6V0A1 regulating memory CD8+T-cell effectiveness and tumor growth *in vivo* (**Fig. 4 and Supplementary Fig. 12A-E**). The suppression of memory CD8+T cells is essential to ATP6V0A1-induced tumor growth (**Figs. 2-3**). Therefore, it is suggested that ATP6V0A1-regulated TGF- β 1 suppresses the activities of CD44+CD8+T cells and thus promotes CRC development.

Note: The sequences of *Tgfb1*/TGFB1-targeted siRNAs are listed in Supplementary Table 3 (marked in red).

Q3: Experimental results should be shown in the results section in general. It is thus suggested that **Supplementary Figs. 25 and 26** should be described in the results section.

•**Response:**

In accordance with the reviewer's valuable suggestion, we have described the

data of previous Figs. S25 and S26 in the Result section “ATP6V0A1 facilitates the transportation of exogenous cholesterol to the ER via the RABGEF1-dependent endosome maturation pathway” (**Pages 18-20; Marked in red**). The data of the previous Fig. S26 are moved to the **new Fig. S21**. We described these data together with those regarding the roles of endosome maturation in ATP6V0A1-regulated TGF- β 1 expression (**Fig. 7G-H**). The data in the new Fig. S21 suggested the critical roles of endosome acidification in ATP6V0A1-regulated TGF- β 1 expression. Endosome acidification results from endosome maturation and further drives the endo-lysosomal traffic for cholesterol absorption. Therefore, by describing the data of New Fig. S21, together with those of Fig. 7G-H, we would strengthen the evidence to suggest that the RABGEF1-dependent endosome maturation and cholesterol absorption play critical roles in ATP6V0A1-regulated TGF- β 1 expression. Moreover, the data of the previous Fig. S25 are moved to the **new Fig. S22** and are described in the last paragraph of the section “ATP6V0A1 facilitates the transportation of exogenous cholesterol to the ER via the RABGEF1-dependent endosome maturation pathway”. In the earlier part of this section, we have determined that ATP6V0A1 facilitates 24-OHC-mediated TGF- β 1 expression through the cholesterol absorption driven by RABGEF1-dependent endosome maturation. Still, it is unclear whether the regulation of the RABGEF1/TGF- β 1 pathway by ATP6V0A1 relies on the changes in the expression level of V-ATPase complex. By answering this question with the new Fig. S22, we would make the structure of this section more complete.

Please note that we have removed descriptions that overlap entirely with the Results section from the Discussion section but retained the main framework of the corresponding discussion. The revisions in the Discussion section are marked in red (**Pages 26-27**).

Q4: In the previous version of manuscript, the authors did not show factor(s) that potentially upregulate ATP6V0A1 in tumor cells. If the authors have any thoughts on this issue, they should address that in the discussion section.

•**Response:**

Thanks for the suggestion. In the revised manuscript, we have added a brief discussion about the factors potentially upregulating ATP6V0A1 in CRC cells (**Lines 501-506**).

Q5: In Fig. 2E, the authors calculate the significance between shNTC and shv0a1-1 and that between Shv01-1 and shv01-2 but the latter should be the significance between shNTC and shv0a1-2.

•**Response:**

Thank you for pointing out the issue. Fig. 2E has been modified accordingly in

the revised manuscript. Although the growth of MC38 shv0a1-2 tumors in C57BL/6 Rag2^{-/-}Il2rg^{-/-} mice exhibited a statistically significant change compared to MC38 shNTC tumors, the reduction was very slight, resembling the trends observed in the in vitro model and the NOD/SCID mice model (**Fig. 2A, F, H**). In fact, the similar functions exhibited by “shv0a1-1” and “shv0a1-2” in all models of our present study suggest that ATP6V0A1 promotes CRC development by suppressing anti-tumor immunity.

Reviewer: 2

The authors have responded to my comments and suggestions.

●**Response:**

We appreciate very much your favorable reply.

Reviewer: 3

The reviewers have satisfactorily addressed my comments.

●**Response:**

We appreciate very much your favorable reply.

Reviewer: 4

Thank you very much for the positive and constructive comments and suggestions to improve our manuscript. Below is a point-by-point response to the comments and suggestions.

General comments

Q1-1: *In supplementary fig.1 to supplementary fig.3, the authors divide TCGA dataset into lipid metabolism high group and low group, and ATP6V0A1 was selected among the V-ATPase subunits highly expressed in the lipid metabolism high group as the most inversely correlated with immune score. However, several subunits other than ATP6V0A1, for instance ATP6V0C, ATP6V0D1 and ATP6V0E1, show opposite trend in the immune score despite their increased expression in the lipid metabolism high group. Moreover, although ATP6V0A1 in CRCs mainly affects CD8⁺ T cell function in this study, all immune cell populations are considered as the immune score. Therefore, it is necessary to clarify how ATP6V0A1 differs in expression and function compared to other V-ATPase subunits in colorectal cancer cells, and why*

ATP6V0A1 is more important in lipid-mediated immunosuppression than the other subunits.

Q1-2: Since the intent of my question was “how ATP6V0A1 differs in expression and function compared to other V-ATPase subunits in colorectal cancer cells, and why ATP6V0A1 is more important in lipid-mediated immunosuppression than the other subunits?”, I consider the newly added data as supplemental fig 25 is very important. Therefore, the authors should show that there is no change in ER-cholesterol in shV0A2 and shTcirg1 as compared to shNTC by using the same experimental method performed in Figure 5H.

●**Response:**

Thanks for the insightful comments and suggestions. As suggested, we have analyzed the level of ER cholesterol in the MC38 cells with *Atp6v0a2* or *Tcirg1* knockdown using the same experimental method performed in Figure 5H. The data in **new Supplementary Figures 22G and 22H** show that both knockdowns of *Atp6v0a2* and *Tcirg1* in MC38 cells do not significantly alter the level of ER-derived cholesterol (**Lines 376-380**).

Note: Since experimental results should be shown in the results section in general, we have integrated the data from the previous Supplementary Figure 25 into the result section “ATP6V0A1 facilitates the transportation of exogenous cholesterol to the ER via the RABGEF1-dependent endosome maturation pathway” (**the last paragraph of this section; Pages 19-20**). Moreover, the previous Supplementary Figure 25 is replaced by the new Supplementary Figure 22.

Q2-1: In addition, the flowcytometry (FCM) analysis in Figure 2 shows not including live dead, CD45, or CD3 staining, therefore, target cells are not properly gated and the reliability of the subsequent analysis cannot be ensured.

Q2-2: Properly addressed.

●**Response:**

Thanks for your favorable reply.

Specific points

Q1-1: Quantitative evaluation should be added in all Western blotting.

Q1-2: Properly addressed.

●**Response:**

Thanks for your favorable reply.

Q2-1: Numbers on the Y-axis should start from zero (Fig.3e, 3f, 4d, 4e, 4f, 4g, 6a, 6b, 6c etc).

Q2-2: New Figure 5D, 5G and 5H remain uncorrected.

●Response:

Thanks for the comment. As suggested, new Figures 5D, 5G, and 5H have been corrected as suggested in the revised manuscript.

Q3-1: In supplementary fig.4, bulk MC38 tumors include a lot of tumor-infiltrated immune cells and stroma cells. The increased expression of ATP6V0A1 by HFD is not necessarily derived from tumor cells.

Q3-2: Properly addressed.

●Response:

Thanks for your favorable reply.

Q4-1 : In supplementary fig.5, the authors evaluate the expression of ATP6V0A1 by immunohistochemistry (IHC) using specimens from CRC patients. Is high expression of ATP6V0A1 in IHC staining associated with low CD8+T cell infiltration?

Q4-2: In Figure 8A and 8B, IHC staining of ATPV0A1 is evaluated by IHC score. The same method should be applied for the newly added Figure 8H and 8I. Also, sample number is too small. It should be evaluated on at least 30-40 specimens.

●Response:

Thanks for the comments and suggestions. To detect the activation of memory CD8+ T cells, Figures 8H and 8I require staining for three markers in one sample section. Compared with IHC, IF could have a better simultaneous staining effect on three or more markers. Therefore, we used the IF staining for the analysis of memory CD8+ T-cell activation. To keep the method consistency in the analysis for the correlation of ATP6V0A1 with RABGEF1, TGF- β 1, and memory CD8+ T-cell activation, the expression levels of ATP6V0A1, RABGEF1, and TGF- β 1 were also assessed using the IF staining method. In fact, the IF staining method is more conducive to analyzing the co-localization of different markers within tissues (Fig. 8G). Moreover, it is of note that the specimens used in Figures 8A and 8B and those used in Figures 8F-8I are from different CRC cohorts.

Also, we agree with the reviewer that the previous sample number for Figs. 8H-I is small, and we have increased the sample number to 32 in the revised Figs. 8H-I to improve the data reliability.

Q5-1: *In supplementary fig.10, single cell RNA seq is powerful to identify the specific immune cell subset. However, the authors should be careful when using scRNAseq for quantitative evaluation. FCM is advantageous for quantitative evaluation, which is also available for evaluation at protein level. If there is a marker that characterizes memory like T-2 subset, it is necessary to confirm these cells by FCM or IHC and whether the number of T cells expressing that marker is increasing in the tumor.*

Q5-2: Properly addressed.

●Response:

Thanks for your favorable reply.

Q6-1: *Representative FCM images are necessary for Fig.2i and Fig.3f.*

Q6-2: Properly addressed.

●Response:

Thanks for your favorable reply.

Q7: *In Fig.3b, CD8+ cells were isolated from tumors. Their purity should be shown.*

Q7-2: Properly addressed.

●Response:

Thanks for your favorable reply.

REVIEWERS' COMMENTS

Reviewer #1 (Remarks to the Author):

The authors have addressed comments raised by reviewers and I have no more comments.

Reviewer #4 (Remarks to the Author):

The reviewers have satisfactorily addressed my comments.